# Fine-scale fluctuations of PM$_1$, PM$_{2.5}$, PM$_{10}$ and SO$_2$ concentrations caused by a prolonged volcanic eruption (Fagradalsfjall 2021, Iceland)

Rachel C. W. Whitty[1], Evgenia Ilyinskaya[1], Melissa A. Pfeffer[2], Ragnar H. Thrastarson[2], Þorsteinn Johannsson[3], Sara Barsotti[2], Tjarda J. Roberts[4, 5], Guðni M. Gilbert[2, 9], Tryggvi Hjörvar[2], Anja Schmidt[6, 7, 8], Daniela Fecht[10], Grétar G. Sæmundsson[11]

[1]COMET, Institute of Geophysics and Tectonics, School of Earth and Environment, University of Leeds, Leeds, LS2 9JT, United Kingdom

[2]Icelandic Meteorological Office, 150 Reykjavík, Iceland

[3]The Environment Agency of Iceland, 108 Reykjavík, Iceland

[4]CNRS UMR7328, Laboratoire de Physique et de Chimie de l'Environnement et de l'Espace, Universite d'Orleans, Orleans, 45071, France

[5]LMD/IPSL, ENS, Université PSL, École Polytechnique, Institute Polytechnique de Paris, Sorbonne Université, CNRS, F-75005 Paris, France

[6]Yusuf Hamed Department of Chemistry, University of Cambridge, Cambridge, CB2 1EW, United Kingdom

[7]Institute of Atmospheric Physics (IPA), German Aerospace Centre (DLR), 82234 Oberpfaffenhofen, Germany

[8]Meteorological Institute, Ludwig Maximilian University of Munich, 80333 Munich, Germany

[9]Nox Medical, 150 Reykjavík, Iceland

[10]MRC Centre for Environment and Health, School of Public Health, Imperial College London, London, W12 0BZ, United Kingdom

[11]Department of Research and Analysis, Icelandic Tourist Board, 101 Reykjavík, Iceland

*Correspondence to*: Evgenia Ilyinskaya (e.ilyinskaya@leeds.ac.uk)

## Abstract

The 2021 Fagradalsfjall eruption marked the first in a series of ongoing eruptions in a densely populated region of Iceland (> 260,000 residents within 50 km distance). This eruption was monitored by an exceptionally dense regulatory air quality network, providing a unique opportunity to examine fine-scale dispersion patterns of volcanic air pollutants (SO$_2$, PM$_1$, PM$_{2.5}$, PM$_{10}$) in populated areas.

Despite its relatively small size, the eruption led to statistically-significant increases in PM and SO$_2$ concentrations at distances of at least 300 km. Peak daily-mean concentrations of PM$_1$ (measured in the capital area, 25-35 km distance from the source)

rose from 5–6 µg/m³ to 18–20 µg/m³, and the proportion of $PM_1$ within $PM_{10}$ increased by ~50%. In areas with low background pollution, average $PM_{10}$ and $PM_{2.5}$ levels increased by ~50% but in places with high background sources, the eruption's impact was not detectable. These findings suggest that ash-poor eruptions are a major source of $PM_1$ in Iceland and potentially in other regions exposed to volcanic emissions.

Air quality guidelines for $PM_1$ and $SO_2$ were exceeded more frequently during the eruption than under background conditions. This suggests the potential for an increase in adverse health effects. Moreover, pollutant concentrations exhibited strong fine-scale temporal (≤1 hour) and spatial (<1 km) variability. This suggests disparities in population exposures to volcanic air pollution, even from relatively distal sources, and underscores the importance of a dense monitoring network and effective public communication.

## 1 Introduction

Airborne volcanic emissions pose both acute and chronic health hazards that can affect populations across large geographic areas (Stewart et al., 2021, and references within). Globally, over one billion people are estimated to live within 100 km of an active volcano (Freire et al., 2019), a distance within which they might be exposed to volcanic air pollution (Stewart et al., 2021). The number of potentially exposed people is growing, for example, due to building expansion into previously uninhabited areas near volcanoes. In this study, we examine the impacts of volcanic emissions on air quality in populated areas using high-resolution, high-quality observational data. We focus on the 2021 Fagradalsfjall fissure eruption on the Reykjanes peninsula as a case study. Fissure eruptions are one of the most common types of volcanic activity that affects air quality. Recent examples include the Kīlauea volcano in Hawai'i (with tens of episodes since 1983), Cumbre Vieja on La Palma in 2021, and the Reykjanes peninsula in Iceland (11 eruptions since 2021). Fissure eruptions have low explosivity and produce negligible ash but release prodigious amounts of gases and aerosol particulate matter close to ground level. Even small fissure eruptions can cause severe air pollution episodes (Whitty et al., 2020).

Fine-scale spatial variability in air pollutant concentrations—characterized by steep gradients over distances of just a few kilometres or less—is currently one of the most active areas of research within the broader field of air pollution (Apte and Manchanda, 2024). In urban areas, these fine-scale variations contribute to disparities in air quality, population exposure, and associated physical, mental, and social well-being (Apte and Manchanda, 2024, and references within). The 2021 Fagradalsfjall eruption provided a novel opportunity to investigate the fine-scale variability of volcanic air pollution in urban settings, as it was monitored by an exceptionally dense regulatory air quality network. Here, we use the term 'regulatory' to describe an air quality monitoring network operated by a national agency, employing certified commercial instrumentation with regulated setup and calibration protocols. These networks provide high-accuracy, high-precision measurements with high temporal resolution, but typically with low spatial resolution due to the high costs of installation (typically > € 100,000) and maintenance (typically > € 100,000 per annum). For example, Germany has approximately one regulatory station per ~250,000 people, with a similar density in the United States (Apte and Manchanda, 2024). In many volcanic regions, regulatory air quality monitoring

is either absent or very sparse (Felton et al., 2019). Prior to our study, the best-observed case studies of volcanic air pollution came from Kīlauea volcano in Hawaii (in particular, its large fissure eruption in 2018), and the large Holuhraun fissure eruption 2014-2015 in Iceland (Crawford et al., 2021; Gíslason et al., 2015; Ilyinskaya et al., 2017; Schmidt et al., 2015; Whitty et al., 2020). These events were monitored by relatively few and distant regulatory stations—approximately 90 km from the eruption site at Holuhraun and about 40 km at Kīlauea. In contrast, the 2021 Fagradalsfjall eruption occurred in Iceland's most densely

populated region and in response, national authorities made a strategic decision early on to expand the regulatory network, ensuring that nearly every community was covered by at least one station. During the eruption, 27 regulatory stations were operational across Iceland, with 14 located within 40 km of the eruption site. Some stations were positioned less than 1 km apart, enabling unprecedented spatial resolution in observing volcanic air pollution.

Regulatory air quality networks can be supplemented by so-called lower-cost sensors (LCS), which are typically small in size

(a few centimetres) and cost approximately € 200. An active body of research on the expanding use of LCS highlights their potential to enhance the relatively sparse regulatory networks (reviewed in Apte and Manchanda, 2024; and Sokhi et al., 2022). For example, during a two-week campaign in 2018, the regulatory air quality network on Hawai'i Island was augmented with 16 LCS. This denser network significantly changed the estimates of population exposure to volcanic air pollution (Crawford et al., 2021). Despite their advantages in affordability and portability, LCS have notable limitations, including relatively poor

accuracy and precision compared to regulatory-grade instruments, and a lack of standardised protocols for installation and maintenance. In our study, LCS were deployed to establish a rapid-response monitoring network directly at the eruption site, aimed at mitigating exposure hazards for the approximately 300,000 visitors who came to view the eruption. We present and discuss the use of LCS in a crisis mitigation context, which has broader relevance for other high-concentration, rapid-onset air pollution events, such as wildfires.

## 1.1 Volcanic air pollutants and associated health impacts

Much of the existing knowledge on the health impacts of volcanic gases and aerosols comes from epidemiological and public health investigations of the eruptions at Holuhraun in Iceland and Kīlauea in Hawaii. The Holuhraun eruption was associated with increased healthcare utilisation for respiratory conditions in the country's capital area, located approximately 250 km from the eruption site (Carlsen et al., 2021a, b). These findings are consistent with observations from Kīlauea on Hawaii, which

have been based on more qualitative health assessments and questionnaire-based surveys (Horwell et al., 2023; Longo, 2009; Longo et al., 2008; Tam et al., 2016). Volcanic emissions contain a wide array of chemical species, many of which are hazardous to human health (Stewart et al., 2021). In this study, we focus on sulfur dioxide gas ($SO_2$) and three particulate matter (PM) size fractions— $PM_1$, $PM_{2.5}$, $PM_{10}$—which refer to particles with aerodynamic diameters less than 1 μm, 2.5 μm, and 10 μm, respectively. These pollutants are typically elevated both near the eruption source and at considerable distances

downwind reviewed in Stewart et al. (2021). Throughout this work, we use the term 'volcanic emissions' to refer collectively to $SO_2$ and PM, unless otherwise specified.

Sulfur dioxide is abundant in volcanic emissions and a key air pollutant in volcanic areas (Crawford et al., 2021; Gíslason et al., 2015; Ilyinskaya et al., 2017; Schmidt et al., 2015; Whitty et al., 2020). Laboratory studies have shown that individuals with asthma are particularly sensitive to even relatively low concentrations of $SO_2$ (below 500 µg/m³), and air quality thresholds are typically established to protect this vulnerable group (US EPA National Center for Environmental Assessment, 2008). Epidemiological studies in volcanic regions further indicate that young children (defined as ≤ 4 years old) and the elderly (≥ 64 years old) are more susceptible to adverse health effects from above-threshold $SO_2$ exposure compared to the general adult population (Carlsen et al., 2021b). This study provides an unprecedented spatial resolution of $SO_2$ exposure in a densely populated, modern society affected by this pollutant. In recent decades, the number of regulatory air quality stations monitoring $SO_2$ has declined across much of the Global North, largely due to reductions in anthropogenic emissions, particularly from coal combustion. To our knowledge, Iceland currently maintains the highest number and spatial density of regulatory $SO_2$ monitoring stations worldwide.

Volcanic emissions are extremely rich in PM, comprising both primary particles emitted directly from the source (including ash) and secondary particles formed through post-emission processes, such as sulfur gas-to-particle conversion. Some eruptions (e.g. at Kīlauea, Cumbre Vieja, and several recent Reykjanes episodes) ignite significant wildfires, which are also a source of PM. All three PM size fractions reported in this study— $PM_1$, $PM_{2.5}$, $PM_{10}$—are known to be significantly elevated near volcanic sources. In fissure eruptions, $PM_1$ is typically the dominant size fraction at-source (Ilyinskaya et al., 2012, 2017; Mather et al., 2003). Exposure to PM air pollution, from natural and anthropogenic sources, has been linked to a wide range of adverse health outcomes, including cardiovascular and respiratory diseases, and lung cancer (Brauer et al., 2024, and references within). Health impacts have been observed even at low concentrations, with children and the elderly particularly vulnerable. The size of PM plays a critical role in determining health impacts. $PM_{2.5}$ has long been associated with worse health outcomes compared to $PM_{10}$ (Janssen et al., 2013; Mcdonnell et al., 2000), and the importance of $PM_1$ is now a key focus in air pollution and health research. Multiple epidemiological studies from China have found $PM_1$ exposure to be more strongly correlated with negative health outcomes than $PM_{2.5}$ (Gan et al., 2025; Guo et al., 2022; Yang et al., 2018; Zhang et al., 2020). In Europe, epidemiological research on $PM_1$ health impacts is still in its early stages (Tomášková et al., 2024), largely due to a lack of high-quality observational data on $PM_1$ concentrations and exposure. This study reports on the first three years of regulatory-grade $PM_1$ measurements in Iceland (2020-2022) and represents the first regulatory-grade time series of $PM_1$ from a volcanic source.

In volcanic emissions, concentrations of both $SO_2$ and PM in various size fractions are consistently elevated, but their relative proportions vary depending on several factors, including distance from the source, plume age, and the rate of gas-to-particle conversion. Existing evidence suggests that this variability in plume composition may influence the associated health outcomes in distinct ways. An epidemiological study in Iceland comparing $SO_2$-dominated plumes with PM-dominated plumes found that the latter was associated with a greater increase in the dispensation of asthma medication and reported cases of respiratory infections (Carlsen et al., 2021a). In contrast, statistically significant increases in healthcare utilization for chronic obstructive pulmonary disease (COPD) were observed only in association with exposure to $SO_2$-dominated plumes (Carlsen et al., 2021a).

Our study contributes a dataset on different types of volcanic air pollutants with a higher spatial resolution than has previously been possible. This offers a foundation for future epidemiological research into the health impacts of recent and ongoing eruptions in Iceland.

### 1.2 Fagradalsfjall 2021 eruption

The 2021 Fagradalsfjall event (19 March - 19 September 2021) was the first volcanic eruption on the Reykjanes peninsula in nearly 800 years. This region is the most densely populated area of Iceland, with over 260,000 people—around 70% of the national population—residing within 50 km of the eruption site. The eruption site was 9 km from the town of Grindavík and approximately 25 km from the capital area of Reykjavík (Fig. 1). Although the eruption took place in an uninhabited area, it attracted an estimated 300,000 visitors who observed the event at close range.

The eruption was a basaltic fissure eruption with an effusive and mildly explosive style, dominated by lava fountaining and lava flows (Barsotti et al., 2023). While relatively small in size—emitting a total of ~0.3–0.9 Mt of $SO_2$ and covering an area of 4.82 km² with lava (Barsotti et al., 2023; Pfeffer et al., 2024)—its proximity to urban areas and the high number of visitors likely resulted in greater population exposure to volcanic air pollution than any previous eruption in Iceland.

This eruption is considered to mark the onset of a new period of frequent eruptions on the Reykjanes peninsula. Such periods,
locally referred to as the 'Reykjanes Fires', have occurred roughly every 1000 years, each lasting for decades to centuries. The last period of Reykjanes Fires ended with an eruption in 1240 CE (Sigurgeirsson and Einarsson, 2019). Since the 2021 eruption, eleven further eruptions have occurred on the Reykjanes peninsula: two within the Fagradalsfjall volcanic system (August 2022 and July 2023), and nine within the adjacent Reykjanes-Svartsengi system (December 2023 to August 2025). The 2021 eruption did not trigger significant wildfires; however, several subsequent episodes have caused extensive fires (primarily of
vegetation but also some urban structures), warranting a dedicated investigation into their effects on air quality and related health outcomes. Volcanic unrest continues at the time of writing, and based on the eruption history of the Reykjanes peninsula, further eruptions may occur repeatedly over the coming decades or centuries.

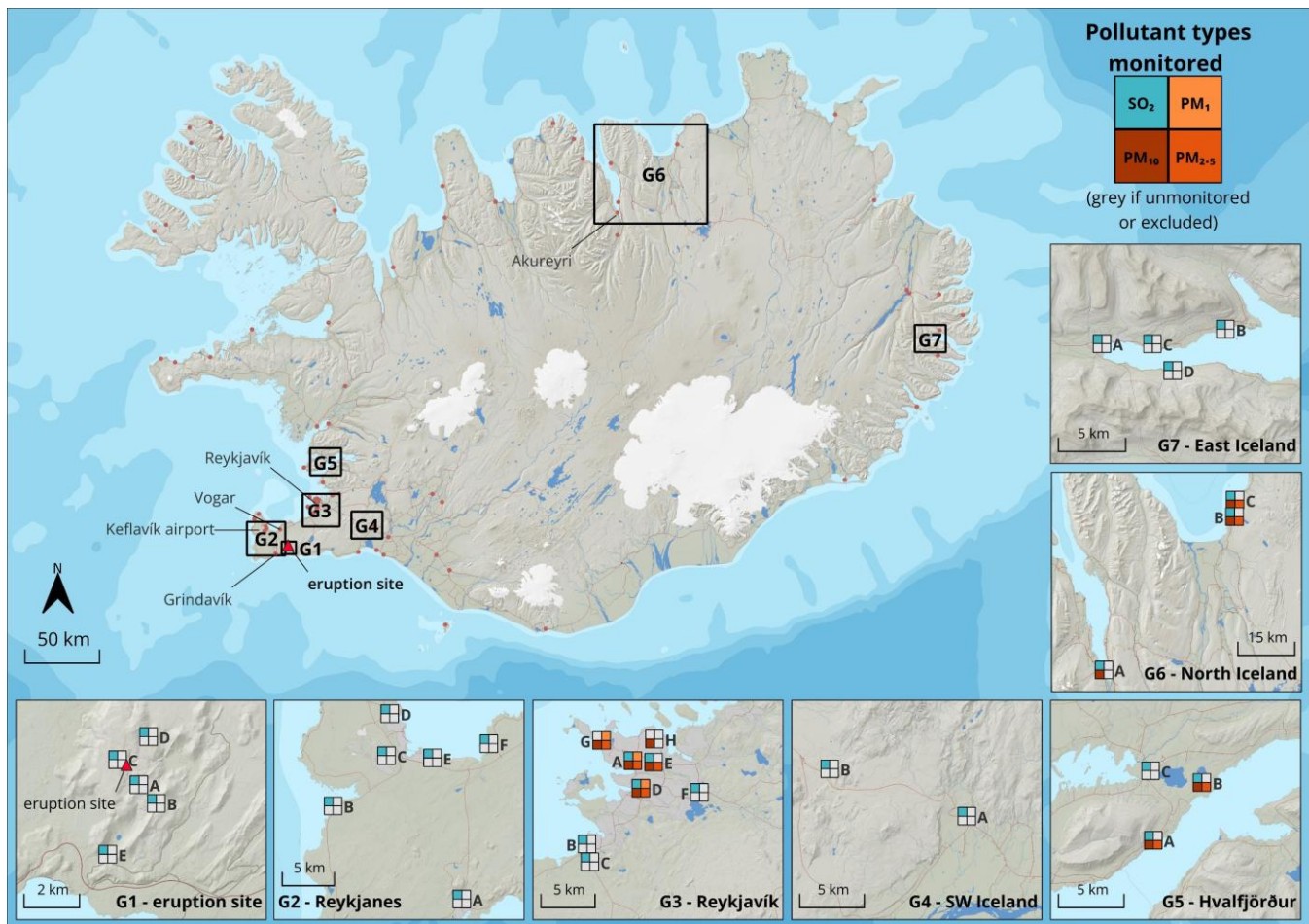

**Figure 1: Map of Iceland showing the eruption site and air quality monitoring stations. Red circles on the main map show the location of populated areas, including the capital area Reykjavík which is represented with a comparatively larger circle. The stations were organised in seven geographic clusters (each shown on the enlarged insets). G1 - Eruption site (0-4 km from the eruption site). G2 - Reykjanes peninsula (9-20 km). G3 - Reykjavík capital area (25-35 km). G4 - Southwest Iceland (45-55 km). G5 - Hvalfjörður (50-55 km). G6 - North Iceland (A and B ~280 km; C and D ~330 km). G7 - East Iceland (~400 km). The map shows the air pollutant species monitored at each station (SO₂, PM₁₀, PM₂.₅, PM₁). Areas G2-G7 were monitored with regulatory stations, while G1 was monitored using lower-cost eruption response sensors. Source and copyright of basemap and cartographic elements: Icelandic Met Office & Icelandic Institute of Natural History.**

## 2 Methods

Data were collected by two types of instrument networks:

1. A regulatory municipal air quality (AQ) network, managed by the Environmental Agency of Iceland (EAI), which measured $SO_2$ and particulate matter (PM) in different size fractions.

2. An eruption-response lower-cost sensor (LCS) network measuring $SO_2$ only, operated by the Icelandic Meteorological Office (IMO).

## 2.1 Regulatory municipal network

The regulatory network monitors air quality across Iceland in accordance with national legal mandates and complies with Icelandic Directive (ID) regulations. Most of the monitoring stations are located in populated areas and measure a variety of air pollutants. Here, we analysed $SO_2$ and PM in the $PM_1$, $PM_{2.5}$, and $PM_{10}$ size fractions, which are the most important volcanic air pollutants with respect to human health in downwind populated areas (Stewart et al., 2021). Detection of $SO_2$ is based on pulsed ultraviolet fluorescence, and detection of PM is based on light scattering photometry and beta attenuation. The detection 175 limits for the majority of the stations in this study were reported to be ~1-3 µg/m³ $SO_2$ and < 5 µg/m³ $PM_{10}$. Station-specific instrument details, detection and resolution limits, and operational durations are in Supplementary Table S1. Figure 1 shows the location of the stations and the air pollutants species measured at each site.

## 2.2 Eruption site sensors

At the eruption site (0.6-3 km from the active craters), the Icelandic Meteorological Office (IMO) installed a network of five 180 commercially available $SO_2$ LCS (Fig. A1) between April and July 2021 to monitor air quality. PM was not monitored with this network due to cost-benefit considerations. Two LCS sensor brands were used, Alphasense $SO_2$-B4 and Crowcon XGuard. The sensor specifications and operational durations are detailed in Table S1. Figure 1 shows the location of the eruption-response $SO_2$ sensor network. Stations A, B, and E were in close proximity to the public footpaths, while stations C and D were further afield to the north and northwest of the eruption site. The main purpose of the eruption-response network was to 185 alert visitors when $SO_2$ levels were elevated and therefore potentially unhealthy. The measurements from the sensor network were publicly available in real-time on the EAI air quality monitoring website (airquality.is). The eruption site was staffed by members of the rescue services and/or rangers, who carried handheld $SO_2$ LCS to supplement the installed network. When any of the LCS reported $SO_2$ concentrations as elevated (potentially-above 350 µg/m³) visitors were urged to relocate to areas with cleaner air. During the course of the 2021 eruption and subsequent events (2022–2025), $SO_2$ measurements from the LCS 190 stations were also used by the IMO to produce hazard maps around the active and potential eruption sites, with hazard zones defined by the distances at which elevated $SO_2$ was detected (Icelandic Meteorological office, 2025).

The LCS were used to alert people to elevated $SO_2$ levels and were not used to report accurate $SO_2$ concentrations. This was because LCS are known to be significantly less accurate than regulatory instruments (Crilley et al., 2018; Whitty et al., 2022, 2020). Whitty et al. (2022) assessed the performance of $SO_2$ LCS specifically in volcanic environments (same or comparable 195 sensor models to those used here) and found that they were frequently subject to interferences restricting their capability to monitor $SO_2$ in low concentrations. The sensor accuracy identified in the field study by Whitty et al. (2022) was significantly poorer than the detection limits reported by the manufacturer.

The sensors used in this study were not calibrated or co-located with higher-grade instruments during the field deployment as this network was set up *ad hoc* as part of an eruption crisis response by the IMO. The crisis was two-fold: the eruption itself, 200 and the unprecedented crowding of people who wanted to view the eruption at very close quarters. Furthermore, the 2021

eruption occurred during national and international COVID-19 lockdowns, which reduced the capacity for field-based research and operations.

The absence of a regulatory-grade field calibration significantly limits the accuracy of LCS dataset, particularly at lower concentration levels. To partially mitigate this, two LCS units were co-located at station G1-B between 6 and 22 June 2021 to quantify inter-sensor uncertainty. The co-located sensors were of two types used in this study: Crowcon XGuard (deployed at G1-A throughout the monitoring period and at G1-B until 22 June) and Alphasense $SO_2$-B4 (deployed at G1-B from 22 June and at G1-C, D, and E for the entire period). The measured concentrations showed a strong linear correlation ($r^2 = 0.70$), but Alphasense reported lower values relative to Crowcon, with a correlation coefficient of 0.38 (Fig. A2). This coefficient was used to estimate the measurement uncertainty for the two sensor types, represented here as error bars on relevant figures. While the colocation experiment was useful for identifying uncertainty between sensor brands, it did not quantify variability among sensors of the same brand.

Given the calibration and co-location limitations, we do not report quantitative $SO_2$ concentrations from the LCS network. Instead, the data are presented as a qualitative indicator of whether concentrations were likely elevated—defined as exceeding 350 µg m$^{-3}$ hourly mean—within the uncertainty of the sensors. This threshold is approximately two orders of magnitude above the manufacturer-reported detection limit, making it reasonable to assume that such levels were detectable. However, these values should be interpreted only as indicative; 'elevated levels' do not represent confirmed air quality exceedances.

**2.3 Data processing**

$SO_2$ measurements were downloaded from 24 regulatory stations and 5 eruption site sensors, and $PM_{10}$, $PM_{2.5}$ and $PM_1$ were downloaded from 12, 11 and 3 regulatory stations, respectively. Data from the regulatory stations were quality-checked and, where needed, re-calibrated by the EAI. Where the operational duration was sufficiently long, we obtained $SO_2$ and PM measurements for both the eruption period and the non-eruptive background period.

We excluded from the analysis any regulatory stations that had data missing for more than 4 months of the eruption period (>70%). Further details on exclusion of individual stations are in Table S1. These criteria excluded $PM_{10}$ and $PM_{2.5}$ from two stations (G3-B, G3-C); and $PM_{10}$ from one station (G3-H). Data points that were below instrument detection limits were set to 0 µg/m$^3$ in our analysis. See Table S1 for the instrument detection limits of each instrument.

The eruption period was defined as 19 March 2021 20:00 – 19 September 2021 00:00 UTC in agreement with Barsotti et al., (2023). The background period was defined differently for $SO_2$ and PM. For $SO_2$, the background period was defined as 19/03/2020 00:00 - 19/03/2021 19:00 UTC, i.e. one full calendar year before the eruption. Outside of volcanic eruption periods, $SO_2$ concentrations in Iceland are generally low with little variability due to the absence of other sources, as shown by previous work (Carlsen et al., 2021a; Ilyinskaya et al., 2017), and subsequently confirmed by this study. The only exception is in the vicinity of aluminium smelters where relatively small pollution episodes occur periodically. A one-year long period was therefore considered as representative of the background $SO_2$ fluctuations. We checked our background dataset against a

previously published study in Iceland that used the same methods (Ilyinskaya et al., 2017) and found no statistically significant difference.

PM background concentrations in Iceland are much higher and more variable than those of $SO_2$. PM frequently reaches high levels in urban and rural areas, with significant seasonal variations (Carlsen and Thorsteinsson, 2021; Dagsson-Waldhauserova et al., 2014); the causes of this variability are discussed in the Results and Discussion. To account for this variability, we downloaded PM data for as many non-eruptive years as records existed, and analysed only the period 19 March 20:00 – 19 September 00:00 UTC in each year, i.e. the period corresponding to the calendar dates of the 2021 eruption. From here on, we refer to this period as 'annual period'. The annual periods in 2010, 2011, 2014, and 2015 were partially or entirely excluded from the non-eruptive background analysis due to eruptions in other Icelandic volcanic systems (Eyjafjallajökull 2010, Grímsvötn 2011, Holuhraun 2014-2015) and associated post-eruptive emissions and/or ash resuspension events. The annual period of 2022, i.e. the year following the 2021 eruption, was partially included in the background analysis: measurements between 19 March 2022 and 1 August 2022 were included, but measurements from 2 August 2022 onwards were excluded because another eruptive episode started in the Fagradalsfjall volcanic system. Since August 2022 there have been ten more eruptions in the same area at intervals of weeks-to-months, and therefore we have not included more recent non-eruptive background data. Although the 2022 annual period is only partially complete, it was particularly important for statistical analysis of $PM_1$ as operational measurements of this pollutant began only in 2020. The number of available background annual periods for $PM_{10}$ and $PM_{2.5}$ varied depending on when each station was set up, ranging from 1 to 12 (Table S1).

The importance of non-volcanic sources of PM in Iceland meant that PM concentrations during the eruption period may have been elevated independently of volcanic activity. To identify the volcanic contribution to PM levels, we processed the data following a similar approach to Ilyinskaya et al. (2017). PM data were filtered to include only periods when $SO_2$ concentrations exceeded the non-eruptive background average; these periods are hereafter referred to as 'plume-present days'. Stations G3-G and G3-H did not monitor $SO_2$ and were filtered using $SO_2$ data from stations located within 2 km distance (G3-A and G3-E, respectively). This plume-identification approach has inherent strengths and limitations. First, it is effective at sites with negligible non-volcanic $SO_2$ sources, which applies to most of the monitored locations in Iceland; however, its reliability decreases near aluminium smelters, which represented a minor yet locally important $SO_2$ source at stations G5-all, G6-C, and G7-all. Second, it may exclude periods when the volcanic plume was present with low $SO_2$ but elevated PM, as can occur when the plume is chemically mature (Ilyinskaya et al., 2017). Third, it cannot distinguish between days when PM is predominantly sourced from an eruption and days when volcanic PM is strongly mixed with another PM source, such as dust storms. To address these uncertainties, we present both filtered and unfiltered PM datasets and compare them in our discussion. Finally, we considered whether the year 2020 had lower $PM_{10}$ and $PM_{2.5}$ concentrations compared to other non-eruptive years due to COVID-19 societal restrictions and the extent to which this was likely to impact our results. The societal restrictions in Iceland were relatively light, for example, schools and nurseries remained open throughout. We found that the average 2020 $PM_{10}$ and $PM_{2.5}$ concentrations fell within the maximum-minimum range of the pre-pandemic years for all stations except at G3-E where $PM_{10}$ was 10% lower than minimum pre-pandemic annual average, and $PM_{2.5}$ was 12% lower; and at G5-A where

PM$_{2.5}$ was 25% lower (no difference in PM$_{10}$). G3-E is at a major traffic junction in central Reykjavík, and G5-A is on a major commuter route to the capital area. For PM$_1$, only one station was already operational before the COVID-19 pandemic (G3-A); PM$_1$ concentrations at this station were 20% higher in 2020 compared to 2022 (post-pandemic). We concluded that PM data from 2020 should be included in our analysis but we note the potential impact of pandemic restrictions.

## 2.4 Data analysis

We organised the air quality stations into geographic clusters to assess air quality by region. The geographic clusters were the immediate vicinity of the eruption site (G1, 0-4 km from the eruption site), the Reykjanes peninsula (G2, 9-20 km), the capital area of Reykjavík (G3, 25-35 km), Southwest Iceland (G4, 45-55 km), Hvalfjörður (G5, 50-55 km), North Iceland (G6-A ~280 km; G6-B and C ~330 km), and East Iceland (G7, ~400 km), Fig. 1. Appendix A Figs. A3-A9 show SO$_2$ time series data for each individual station in geographic clusters G1-G7, respectively. Appendix A Figs. A10-A12 show PM time series data for each individual station in geographic clusters G3, G5 and G6, respectively.

For each station that had data for both the eruption and background periods, two-sample t-tests were applied to test whether the differences in background and eruption averages were statistically significant for the different pollutant species. For the eruption period, analyses were conducted separately for the full eruption duration and for plume-present days.

In addition to time series analysis, we analysed the frequency and number of events where pollutant concentrations exceeded air quality thresholds. Air quality thresholds are pollutant concentrations averaged over a set time period (usually 60 minutes or 24 hours), which are considered to be acceptable in terms of what is robustly known about the effects of the pollutant on health. An air quality threshold exceedance is an event where the pollutant concentration is higher than that set out in the threshold. Evidence-based air quality thresholds have been defined for SO$_2$, PM$_{2.5}$ and PM$_{10}$, but not yet for PM$_1$, largely due to the paucity of regulatory-grade data on concentrations, dispersion and exposure (World Health Organization, 2021). For SO$_2$, most countries, including Iceland, use an hourly-mean threshold of 350 µg/m$^3$; and the threshold for the total number of exceedances in one year is 24 (Icelandic Directive, 2016). We used these thresholds for SO$_2$ in our study. The air quality thresholds for PM are based on 24-hour averages, as there is currently insufficient evidence base for hourly-mean thresholds. For PM$_{10}$ we used the Icelandic Directive (ID) and World Health Organisation (WHO) daily-mean threshold of 50 µg/m$^3$, and for PM$_{2.5}$ we used the WHO daily-mean threshold of 15 µg/m$^3$, as no ID threshold is defined. While there are currently no evidence-based air quality thresholds available for PM$_1$, some countries, including Iceland use selected values to help communicate the air pollutant concentrations and their trends to the public. The Environment Agency of Iceland (EAI) uses a 'yellow' threshold for PM$_1$ at 13 µg/m$^3$ to visualise data from the regulatory stations and this value was used here (termed 'EAI threshold').

To meaningfully compare the frequency of air quality threshold exceedance events for PM$_{10}$, PM$_{2.5}$ and PM$_1$ between the eruption and the non-eruptive background periods we normalised the number of exceedance events. This was done because the eruption covered only one annual period (see the definition of 'annual period' in 2.3) but the number of available background annual periods varied between stations depending on how long they have been operational, ranging between 1 and

12 periods. We normalised by dividing the total number of exceedance events at a given station by the number of annual periods at the same station. For example, for a station where the non-eruptive background was 6 annual periods the total number of exceedance events was divided by 6 to give a normalised annual number of exceedance events. The eruption covered one annual period and therefore did not require dividing. We refer to this as 'normalised number of exceedance events' in the
305 Results and Discussion. Table S1 contains summary statistics for all analysed pollutant means, maximum concentrations, number of air quality threshold exceedances, and number of background annual periods for PM data.

Three regulatory stations within geographic cluster G3 (Reykjavík capital area) measured all three PM size fractions ($PM_1$, $PM_{2.5}$ and $PM_{10}$), which allowed us to calculate the relative contribution of different size fractions to the total PM concentration. Since PM size fractions are cumulative, in that $PM_{10}$ contains all particles with diameters ≤10 µm, the size modes were
310 subtracted from one another to determine the relative concentrations of particles in the following categories: particles ≤1 µm in diameter, 1 - 2.5 µm in diameter and 2.5 - 10 µm in diameter. The comparison of size fractions between the eruption and the background was limited by the relatively short $PM_1$ time series and our results should be re-examined in the future when more non-eruptive measurements have been obtained.

**3 Results and discussion**

**3.1 Eruption-driven increase in $PM_1$ concentrations relative to $PM_{10}$ and $PM_{2.5}$**

Emerging studies of the links between $PM_1$ and health impacts in urban air pollution have shown that even small increases in the $PM_1$ proportion within $PM_{10}$ can be associated with increasingly worse outcomes; e.g. liver cancer mortalities in China were found to increase for every 1% increase in the proportion of $PM_1$ within $PM_{10}$ (Gan et al., 2025). Time series of $PM_1$, $PM_{2.5}$ and $PM_{10}$ concentrations were collected at three stations in the Reykjavík capital (G3-A, G3-D and G3-G, Fig. 1),
allowing us to compare the relative contributions of the three size fractions in this area (25-35 km distance from the eruption site). All three stations exhibited low $SO_2$ concentrations during non-eruptive periods (both in mean values and variability), providing high confidence in detecting plume-present days (126 days at G3-A and G3-G, and 78 days at G3-D, out of 184 eruption days). When we considered the whole eruption period, all three stations showed a measurable increase in the average $PM_1$ mass proportion relative to $PM_{10}$ and $PM_{2.5}$ (Fig. 2). The proportion of $PM_1$ mass within $PM_{10}$ increased from the average
of 16-24% in the background (one standard deviation ±7-13%) to 24-32% during the eruption (±16-19%); and within $PM_{2.5}$ from approximately 47% in the background to ~60% during the eruption period. When considering only plume-present days (Fig. 2), the proportional increase in $PM_1$ was even more pronounced—accounting for 27–36% of $PM_{10}$—compared to background conditions, further highlighting the dominant influence of the volcanic source.

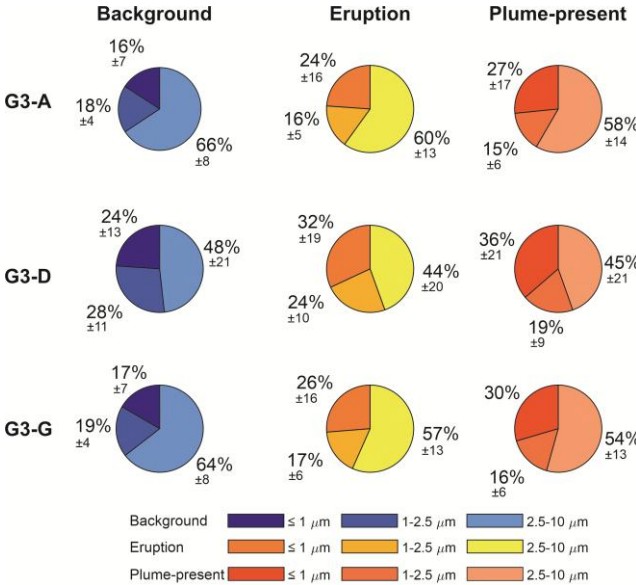

**Figure 2: The relative contributions of three PM size fractions within PM$_{10}$ (expressed as mass%) during the non-eruptive background, during the whole eruption period ('Eruption'), and on plume-present days only (see main text for the definition of 'plume-present'). The size fractions shown are: PM ≤ 1 μm, PM 1–2.5 μm, and PM 2.5–10 μm in diameter. The %mass is the mean ± 1σ standard deviation. G3-A, G3-D and G3-E were the stations in Iceland where all three size fractions were measured, all located within Reykjavík capital area.**

These are novel findings showing that volcanic plumes contribute a higher proportion of PM$_1$ relative to both PM$_{10}$ and PM$_{2.5}$ when sampled at a distal location from the source (25-35 km in this study). When sampled at the active vent, volcanic plumes from basaltic fissure eruptions have been previously shown to contain a large amount of PM$_1$, but also a substantial proportion of coarse PM (> 2.5 μm) (Ilyinskaya et al., 2017; Martin et al., 2011; Mason et al., 2021). At the vent, the composition of the fine and coarse size modes is typically very different: the finer fraction is primarily formed through the conversion of SO$_2$ gas into sulphate particles, whereas the coarser fraction consists of fragmented silicate material (i.e. ash), which may be present in small concentrations even in ash-poor fissure eruptions (Ilyinskaya et al., 2021; Mason et al., 2021). The conversion of SO$_2$ gas to sulphate particles continues for hours to days after emission, generating new fine particles over time (Green et al., 2019; Pattantyus et al., 2018). In contrast, ash particles are not replenished in the plume after emission and are progressively removed through deposition. This may explain the elevated concentrations of particles in the finer size fractions observed downwind of the eruption site relative, to the coarser size fractions.  These findings have implications for public health hazards, as volcanic plumes most commonly affect populated areas located tens to hundreds of kilometres from the eruption site.

### 3.2 Significant increases in average and peak pollutant levels

Most areas of Iceland, up to 400 km from the eruption site, recorded statistically significant increases in average and/or peak SO$_2$ and PM$_1$ concentrations during the eruption compared to the background period.

Figure 3 and Table 1 present $SO_2$ concentrations (hourly-means in µg/m³), measured by regulatory stations across Iceland. During the non-eruptive background period, $SO_2$ concentrations at the majority of the monitored locations were low (long term average of hourly-means generally <2 µg/m³), which is in agreement with previous studies (Ilyinskaya et al., 2017). Stations near aluminium smelters (G5-all, G6-C, and G7-all) had higher long-term average values and periodically recorded short-lived escalations in $SO_2$ hourly-mean concentrations of several tens to hundreds µg/m³ during the background period (Fig. 3, Table 1 and Table S1). The average $SO_2$ concentrations were higher during the eruption at all of the regulatory stations that had data from both before and during the eruption ($n = 16$), and the increase was statistically significant (p <0.05) at 15 out of the 16 stations (with the exception of G7-D near a smelter). The absolute increase in average $SO_2$ concentrations between the background and eruption period was relatively low, on the order of a few µg/m³ (Fig. 3 and Table 1). For example, the average concentration across the Reykjavík capital increased from 0.32 µg/m³ in the background to 4.1 µg/m³ during the eruption.

The eruption period was also associated with substantial increases in peak $SO_2$ concentrations and number of air quality exceedance events across the populated areas. Figure 3 and Table 1 compare the background and eruption periods in terms of peak $SO_2$ concentrations and the number of exceedance events relative to the Icelandic Directive (ID) air quality threshold of 350 µg/m³ hourly-mean. During the non-eruptive background period, $SO_2$ concentrations remained below the ID threshold at all 16 stations that were in operation. In contrast, during the eruption, 15 of the 24 stations recorded exceedances, with individual stations reporting between 0 and 31 events, generally highest near the eruption site. Two communities on the Reykjanes peninsula (G2) also exceeded the threshold for total exceedances in a one-year period (Fig. 3).

We attribute the combination of a relatively low absolute increase in average $SO_2$ concentrations and a large increase in peak concentrations to a combination of the dynamic nature of the eruption emissions (Barsotti et al., 2023; Pfeffer et al., 2024) and highly variable local meteorological conditions (wind rose for the eruption site in Fig. A13). These factors likely resulted in the volcanic plume being intermittently advected into populated areas, rather than acting as a continuous source of pollution.

**Table 1: SO₂ concentrations (hourly-mean, µg/m³) in populated areas around Iceland during both the non-eruptive background and the Fagradalsfjall 2021 eruption. 'Average' is the long-term mean of all stations within a geographic area ± 1σ standard deviation. 'Peak' is the maximum hourly-mean recorded by an individual station within the geographic area. 'ID exceedances' denotes the maximum number of times SO₂ concentrations (at any single station within a geographic area) exceeded the Icelandic Directive (ID) air quality threshold of 350 µg/m³.**

| Geographic area | N of stations | Distance from eruption site (km) | $SO_2$ hourly-mean (µg/m³) | | | | ID exceedances (max $n$) | |
| --- | --- | --- | --- | --- | --- | --- | --- | --- |
| | | | Background average ± standard deviation (1σ) | Eruption average ± standard deviation (1σ) | Background peak | Eruption peak | Background | Eruption |
| Reykjanes peninsula (G2) | 6 | 9-20 | 0.13±0.45 | 4.8±44 | 7.7 | 2400 | 0 | 31 |
| Reykjavík capital (G3) | 6 | 25-35 | 0.32±1.8 | 4.1±21 | 57 | 750 | 0 | 9 |
| South Iceland (G4) | 2 | 45-55 | No data | 6.1±44 | No data | 2400 | No data | 18 |
| Hvalfjörður (G5) | 3 | 50-55 | 3.9±16 | 8.2±28 | 210 | 860 | 0 | 6 |
| North Iceland (G6) | 3 | 280-330 | 0.41±1.6 | 1.7±6.3 | 9.1 at 280 km; 62 at 330 km | 250 at 280 km; 48 at 330 km | 0 | 0 |
| East Iceland (G7) | 4 | 400 | 1.7±4.1 | 2.1±4.9 | 69 | 79 | 0 | 0 |

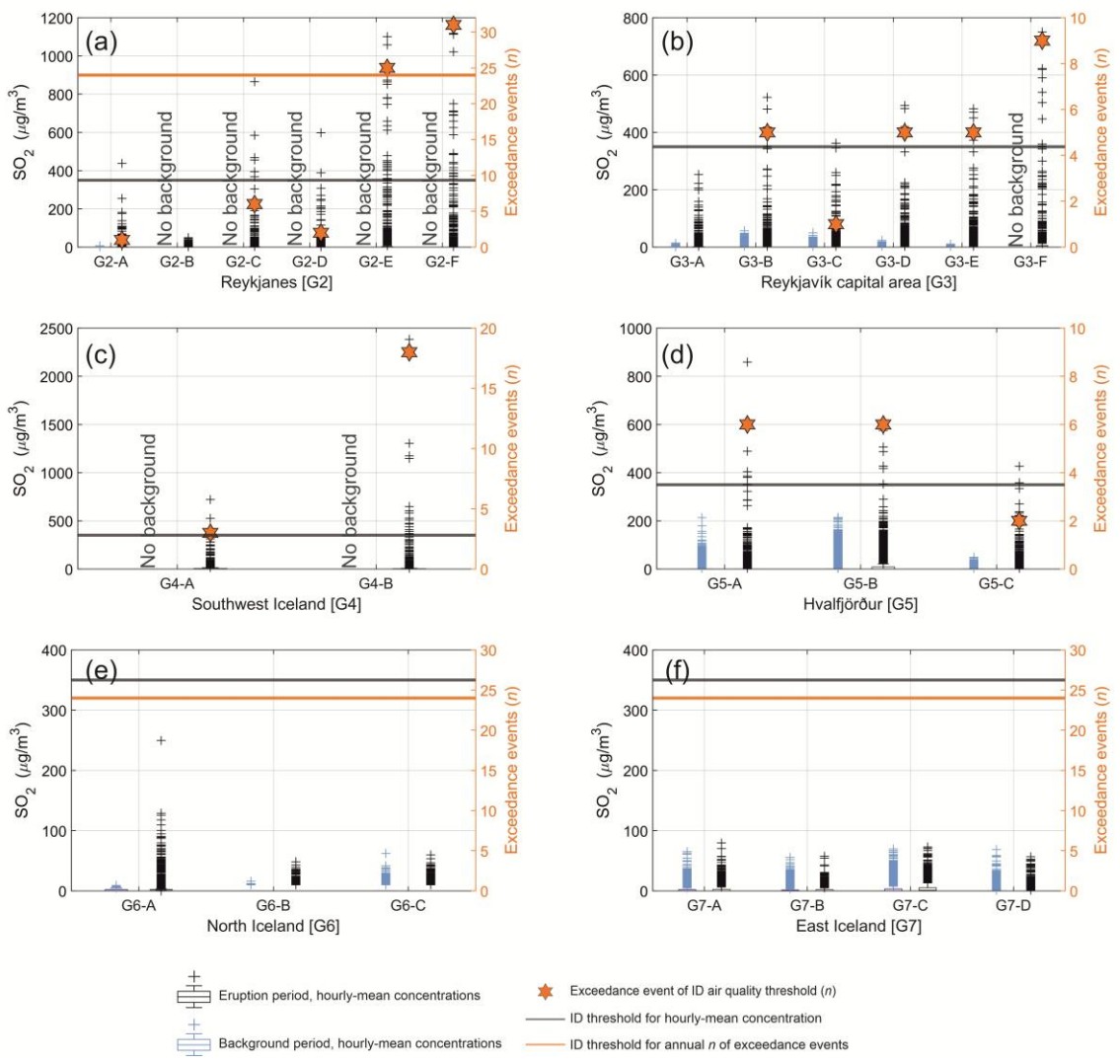

**Figure 3: SO₂ hourly-mean concentrations (μg/m³) and number of Icelandic Directive (ID) threshold exceedance events, measured by 24 regulatory-grade stations in populated areas in Iceland shown as six geographic clusters G2-G7 (panels a-f). Pre-eruptive background data are shown for stations that were operational before the eruption began. The data are presented as box-and-whisker plots: boxes represent the interquartile range (IQR), the whiskers extend to +/–2.7σ from the mean, and crosses represent very high values (statistical outliers beyond +/–2.7σ from the mean). Note that the IQR is very low in most cases due to the negligible SO₂ concentrations in the local background; as a result, most of the SO₂ pollution episodes are statistical outliers. The ID air quality threshold of 350 μg/m³ hourly-mean is indicated by a black horizontal line in all panels. Orange stars represent the number of times this threshold was exceeded at each station ('Exceedance events'). The annual limit for cumulative hourly exceedance events is 24, shown by an orange horizontal line. Stations with orange stars above the orange line exceeded the annual threshold. Time series plots for each station are available in Appendix A.**

Figs. 4–6 and Table 2 show daily mean $PM_{10}$, $PM_{2.5}$ and $PM_1$ concentrations measured in the three regions with regulatory-grade monitoring (G3, G5, G6). Using above-background $SO_2$ as a proxy for plume presence, we identified 126 likely plume-affected days in the Reykjavík capital area (G3), 145 in Hvalfjörður (G5), and 40 in North Iceland (G6). Confidence is high for G3 due to the absence of local $SO_2$ sources, but lower for G5 and G6 because of nearby aluminium smelters; thus, plume-day counts for these areas should be considered maxima.

Some of the highest $PM_{10}$ and $PM_{2.5}$ peaks in Reykjavík capital area (G3) during the eruption occurred on non-plume days (Fig. 4), notably in the periods 24–29 May and 3–4 June 2021. These two events accounted for most threshold exceedances—for example, five of seven for $PM_{10}$ and four of six for $PM_{2.5}$ at station G3-A—and were recorded across all G3 stations, suggesting a diffuse distal source. The dominant non-volcanic PM source in Iceland is natural dust from highland deserts, with dust storms occurring frequently throughout the year with significant regional and seasonal variability (Butwin et al., 2019; Dagsson-Waldhauserova et al., 2014; Nakashima and Dagsson-Waldhauserová, 2019). We used back-trajectory analysis (HYSPLIT) and crowd-sourced observations to confirm that the $PM_{10}$ and $PM_{2.5}$ peaks in Reykjavík on 24–29 May and 3–4 June were consistent with dust storms (Fig. A14).

When focusing on plume-present days, the frequency of $PM_{10}$ and $PM_{2.5}$ exceedances in Reykjavík capital was comparable to or lower than background levels, indicating that ash-poor fissure eruptions are significant PM sources but not exceptionally high compared to other sources. In contrast, $PM_1$ peaks were strongly associated with plume days (Fig. 4), particularly during the time periods 2–6 July and 18–19 July 2021 (Figs. 7–8). We have high confidence in a volcanic origin of these events, supported by concurrent $SO_2$ peaks (up to 250 µg m$^{-3}$ hourly mean) and strong $SO_2$– $PM_1$ correlation (Figs. 7–8). $PM_1$ exceedances never exceeded the EAI threshold (13 µg m$^{-3}$) during background periods, but during the eruption, exceedances occurred at all $PM_1$-monitoring stations, with up to four exceedances at G3-D on 19 July (Fig. 4; Table 2).

Presence of the volcanic plume was associated with a small but statistically significant increase in average $PM_{10}$, $PM_{2.5}$ and $PM_1$ concentrations in multiple locations. Average $PM_1$ concentrations were significantly higher at all monitored stations on plume-present days and throughout the eruption. $PM_{2.5}$ and $PM_{10}$ averages were significantly higher at approximately half of the Reykjavík stations (G3) during plume-present days (Fig. 4). At these stations, $PM_{10}$ increased from ~9 µg m$^{-3}$ (background) to 12–14 µg m$^{-3}$ during the eruption; $PM_{2.5}$ rose from ~3 µg m$^{-3}$ to ~5 µg m$^{-3}$; and $PM_1$ from 1.3–1.5 µg m$^{-3}$ to ~3 µg m$^{-3}$. Stations with significant increases in mean $PM_{10}$ and $PM_{2.5}$ had cleaner backgrounds (peak daily means <90 µg m$^{-3}$ for $PM_{10}$ and <20 µg m$^{-3}$ for $PM_{2.5}$), whereas stations without significant increases had peak daily means ≥160 µg m$^{-3}$ $PM_{10}$ and ≥40 µg m$^{-3}$ $PM_{2.5}$. The higher-background stations were generally near roads with heavy traffic, suggesting that local sources—particularly traffic—were more influential for mean $PM_{10}$ and $PM_{2.5}$ levels than the distal eruption.

Further afield, in Hvalfjörður (G5) and North Iceland (G6), all stations showed significantly higher $PM_{2.5}$ and $PM_{10}$ concentrations on plume-present days compared to background (Figs. 5–6), though plume-day identification in these areas had lower confidence compared for G3 due to a higher $SO_2$ background.

The unequivocal eruption-related increase in both average and peak $PM_1$ concentrations indicates that volcanic fissure eruptions are among the most important—if not the dominant—sources of $PM_1$ in Iceland. Figure 9 and Table A1 compare $PM_1/PM_{10}$ ratios in Reykjavík under three scenarios: (1) volcanic plume presence, (2) two major Icelandic dust storms causing the highest PM pollution in 2021 (24–29 May and 3–4 June), and (3) representative eruption-free background periods. This comparison suggests distinct 'fingerprint' ratios for the different PM sources: volcanic plume periods show the highest $PM_1/PM_{10}$ ratios (mean range 0.3–0.9), dust storms the lowest (mean range during storm peaks 0.04–0.05, mean range during the whole storm 0.1–0.3), and background conditions intermediate (mean ~0.2). These ratios may aid source attribution for PM episodes in Reykjavík and potentially other populated areas, especially when meteorological or visual observations are inconclusive. One limitation of this analysis is that $PM_1$ measurements were only available in Reykjavík; whether volcanic $PM_1$ dominates in more distal communities remains to be investigated when high-quality datasets become available. Furthermore, data from winter eruptions are needed to assess seasonal variability in $PM_1$ sources. This analysis focused only on summer conditions due to the timing of the 2021 eruption. In urban areas, non-volcanic PM peaks are typically higher in winter, driven by tarmac erosion from studded tires (Carlsen and Thorsteinsson, 2021) and there are extreme spikes during New Year's fireworks and bonfires. Finally, we note that our study period included only two dust storms which cased elevated PM concentrations in Reykjavík. Different $PM_1/PM_{10}$ ratios (~0.4–0.5) were reported for two dust storms affecting Reykjavík in 2015 (Dagsson-Waldhauserova et al., 2016), suggesting variability among these events and the need for further research.

The statistically significant increase in average $PM_{2.5}$ and $PM_{10}$ levels observed at least up to 300 km from the eruption site is remarkable, given the eruption's relatively small size and the prominence of non-volcanic PM sources in Iceland.

Historically, larger Icelandic fissure eruptions (>1 km$^3$ of erupted magma) have caused volcanic air pollution episodes far beyond Iceland—across mainland Europe during the 2014–2015 Holuhraun eruption (Schmidt et al., 2015; Twigg et al., 2016) and potentially even farther during the 1783–1784 Laki eruption (Grattan, 1998; Trigo et al., 2009). Simulations indicate that associated health impacts in Europe could have been substantial (Heaviside et al., 2021; Schmidt et al., 2011; Sonnek et al., 2017). During the recent Reykjanes eruptions (2021–2025), elevated volcanic $SO_2$ was detected at ground level by UK regulatory-grade stations on at least one occasion, in May 2024, exceeding previously documented levels at this distance (UKCEH, 2024). This suggests that PM concentrations may also have been elevated beyond Iceland during these events. Assessing the impacts of recent eruptions on air quality and public health in European and potentially more distant communities is therefore an important priority for future research.

**Table 2:** PM$_{10}$, PM$_{2.5}$ and PM$_1$ concentrations (µg/m³, 24-h mean) in populated areas around Iceland during both the non-eruptive background ('BG'), the whole eruption period ('Eruption'), and on 'plume present' days only (see Methods for the definition of plume-present days). 'Average' refers to the long-term mean of 24-hour values of all stations within a geographic area ± 1σ standard deviation. 'Peak' is the maximum 24 h-mean recorded by an individual station within the geographic area. 'AQ exceedances' denotes the maximum number of times PM concentrations (at any single station within a geographic area) exceeded the following thresholds: PM$_{10}$ - 50 µg/m³; PM$_{2.5}$ - 15 µg/m³; PM$_1$ - 13 µg/m³.

| Geographic area | n of stations (PM$_{10}$, PM$_{2.5}$, PM$_1$) | Distance from eruption site (km) | PM$_{10}$ Average (24-h mean ± 1σ, µg/m³) BG | PM$_{10}$ Average Eruption / Plume present | PM$_{10}$ Peak (24-h mean, µg/m³) BG | PM$_{10}$ Peak Eruption / Plume present | PM$_{10}$ AQ exceedances (max n) BG | PM$_{10}$ AQ exceedances Eruption / Plume present | PM$_{2.5}$ Average (24-h mean ± 1σ, µg/m³) BG | PM$_{2.5}$ Average Eruption / Plume present | PM$_{2.5}$ Peak (24-h mean, µg/m³) BG | PM$_{2.5}$ Peak Eruption / Plume present | PM$_{2.5}$ AQ exceedances (max n) BG | PM$_{2.5}$ AQ exceedances Eruption / Plume present | PM$_1$ Average (24-h mean ± 1σ, µg/m³) BG | PM$_1$ Average Eruption / Plume present | PM$_1$ Peak (24-h mean, µg/m³) BG | PM$_1$ Peak Eruption / Plume present | PM$_1$ AQ exceedances (max n) BG | PM$_1$ AQ exceedances Eruption / Plume present |
|---|---|---|---|---|---|---|---|---|---|---|---|---|---|---|---|---|---|---|---|---|
| Reykjavík capital (G3) | 5, 4, 3 | 25-35 | 15±11 | 14±14 | 170 | 140 | 2.9 | 5 | 6.6±6.8 | 5.7±6.2 | 87 | 48 | 15 | 22 | 1.4±0.94 | 2.8±2.6 | 6.3 | 20 | 0 | 4 |
| | | | | 14±10 | | 110 | | 3 | | 6.3±6.3 | | 48 | | 20 | | 3.4±3.0 | | 20 | | 4 |
| Hvalfjörður (G5) | 3 | 50-55 | 5.6±5.7 | 7.3±7.8 | 58 | 59 | 0.25 | 2 | 2.1±3.4 | 3.9±5.3 | 34 | 31 | 1 | 8 | No data | | | | | |
| | | | | 8.3±7.7 | | 55 | | 1 | | 4.7±5.5 | | 31 | | 7 | | | | | | |
| North Iceland (G6) | 3 | 280-330 | 7.7±10 | 8.9±11 | 100 | 79 | 7.7 | 7 | 0.53±1.9 | 0.71±2.2 | 13 | 16 | 0 | 1 | No data | | | | | |
| | | | | 14±12 | | 79 | | 7 | | 3.0±4.0 | | 16 | | 1 | | | | | | |

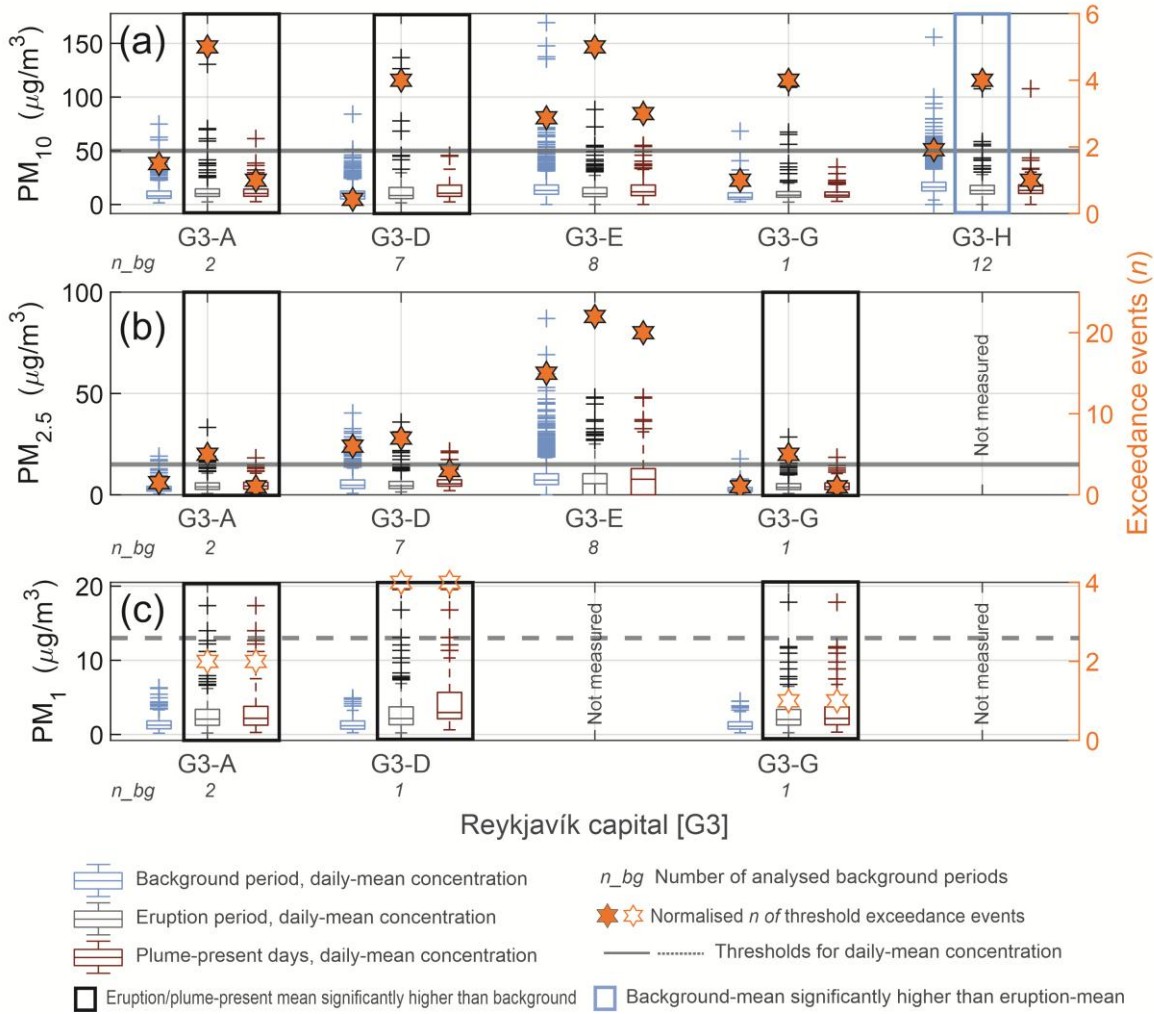

**Figure 4: Daily-mean concentrations of (a) PM$_{10}$, (b) PM$_{2.5}$, and (c) PM$_1$ (µg/m$^3$) measured in the Reykjavík capital area during the non-eruptive background, during the whole eruption period, and on plume-present days only (see Methods for the definition of plume-present days). The data are presented as box-and-whisker plots, where boxes represent the interquartile range (IQR), the whiskers extend to +/–2.7σ from the mean, and crosses represent very high values (statistical outliers beyond +/–2.7σ from the mean). The median is shown with a horizontal line within each box. The value _n_bg_ shown on the x-axis indicates the number of background annual periods available for each station (see Methods for the definition of a background annual period). Stations where the average concentration during the eruption period and/or on the plume-present days was significantly higher (p <0.05) than during the background are highlighted with a black box. The one station where the average concentration during the eruption period was significantly lower than during the background is highlighted with a blue box (G3-H). The absence of a box indicates no significant difference. Black solid line shows the Icelandic Directive (ID) air quality thresholds for PM$_{10}$ = 50 µg/m$^3$ and PM$_{2.5}$ = 15 µg/m$^3$ (24-hour mean). Dashed black line shows the Environmental Agency of Iceland (EAI) threshold for PM$_1$ = 13 µg/m$^3$(24-hour mean), a locally used threshold that is not internationally standardized. Stars with solid orange fill represent the normalised number of times PM$_{10}$ and PM$_{2.5}$ concentrations at each station exceeded the ID thresholds. Non-filled stars indicate the number of times PM$_1$ concentrations exceeded the EAI threshold. The number of the exceedance events is normalized to the length of the measurement period—refer to the main text for details on the normalization method. Time series plots for each station are available in Appendix A.**

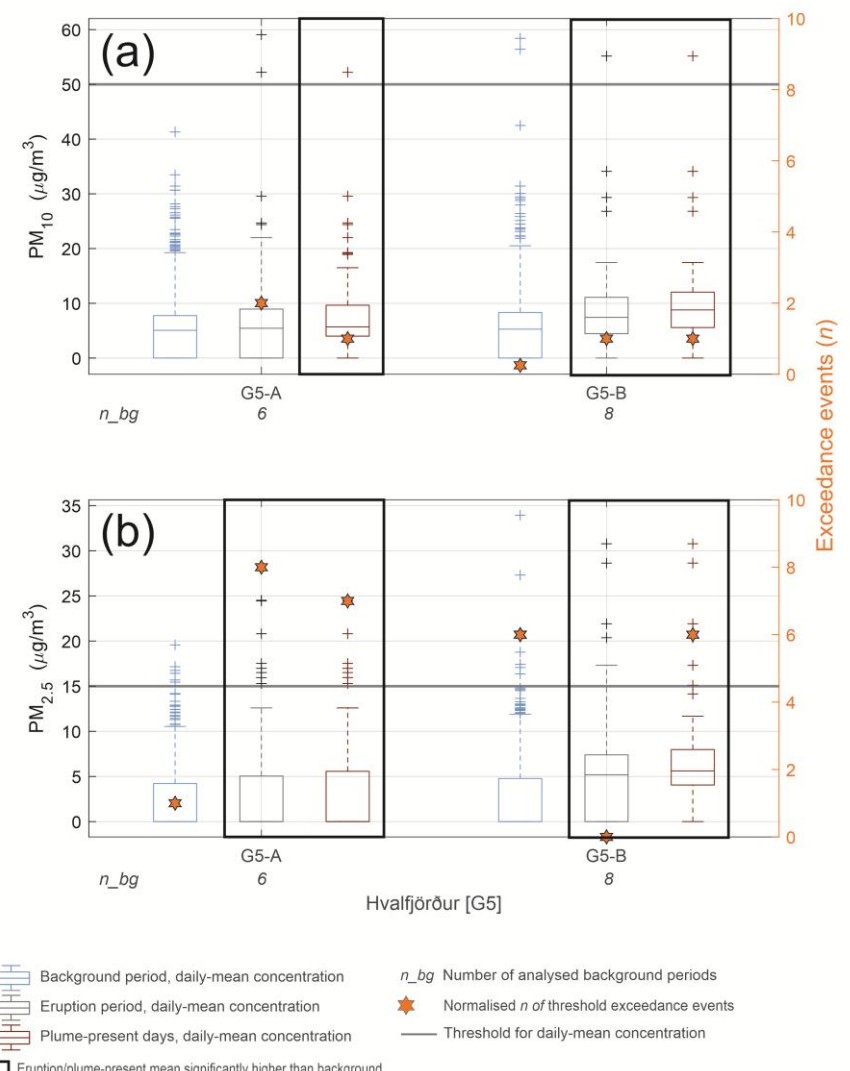

**Figure 5: Daily-mean concentrations of (a) PM$_{10}$, and (b) PM$_{2.5}$ (µg/m$^3$), measured in the Hvalfjörður area during the non-eruptive background, during the whole eruption period, and on plume-present days only (see Methods for the identification of plume-present days). The data are presented as box-and-whisker plots, where boxes represent the interquartile range (IQR), the whiskers extend to +/–2.7σ from the mean, and crosses represent very high values (statistical outliers beyond +/–2.7σ from the mean). The median is shown as a horizontal line within each box; if the median line is absent, the value is zero. The value _n_bg_ shown on the x-axis indicates the number of background annual periods available for each station (see Methods for the definition of a background annual period). Stations where the average concentration during the eruption period and/or the plume-present days was significantly higher (p <0.05) than during the background are highlighted with a black box. The absence of a box indicates no significant difference. Black solid line shows the Icelandic Directive (ID) air quality thresholds for PM$_{10}$ = 50 µg/m$^3$ and PM$_{2.5}$ = 15 µg/m$^3$ (24-hour mean). Stars represent the normalised number of times PM$_{10}$ and PM$_{2.5}$ concentrations at each station exceeded the ID thresholds. The number of the exceedance events is normalized to the length of the measurement period—refer to the main text for details on the normalization method. Time series plots for each station are available in Appendix A.**

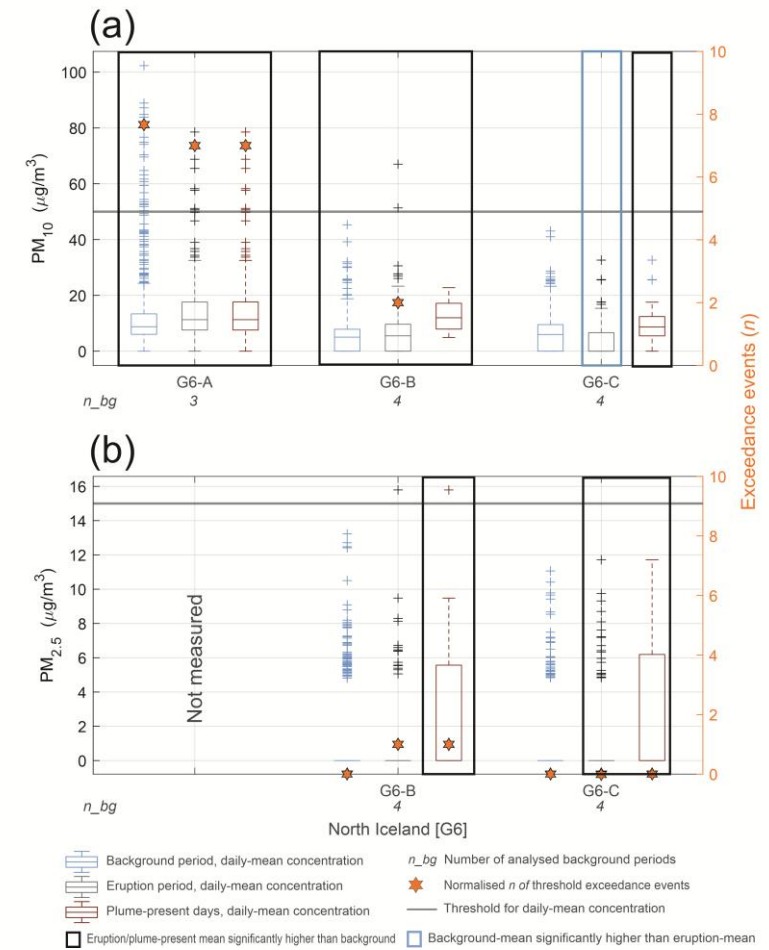

**Figure 6: Daily-mean concentrations of (a) $PM_{10}$, and (b) $PM_{2.5}$ (µg/m³), measured in North Iceland during the non-eruptive background, during the whole eruption period, and on plume-present days only (see Methods for the identification of plume-present days). The data are presented as box-and-whisker plots, where boxes represent the interquartile range (IQR); if a box is missing the 25th and 75th percentiles have the same value. The whiskers extend to +/–2.7σ from the mean, and crosses represent very high values (statistical outliers beyond +/–2.7σ from the mean). The median is shown as a horizontal line within each box; if the median line is absent, the value is zero. The value *n_bg* shown on the x-axis indicates the number of background annual periods available for each station (see Methods for the definition of a background annual period). Stations where the average concentration during the eruption period and/or the plume-present days was significantly higher (p <0.05) than during the background are highlighted with a black box. The one station where the average concentration during the whole eruption was significantly lower than during the background is highlighted with a blue box (G6-C). The absence of a box indicates no significant difference. Black solid line shows the Icelandic Directive (ID) air quality thresholds for $PM_{10}$ = 50 µg/m³ and $PM_{2.5}$ = 15 µg/m³ (24-hour mean). Stars represent the normalised number of times $PM_{10}$ and $PM_{2.5}$ concentrations at each station exceeded the ID thresholds. The number of the exceedance events is normalized to the length of the measurement period—refer to the main text for details on the normalization method. Time series plots for each station are available in Appendix A.**

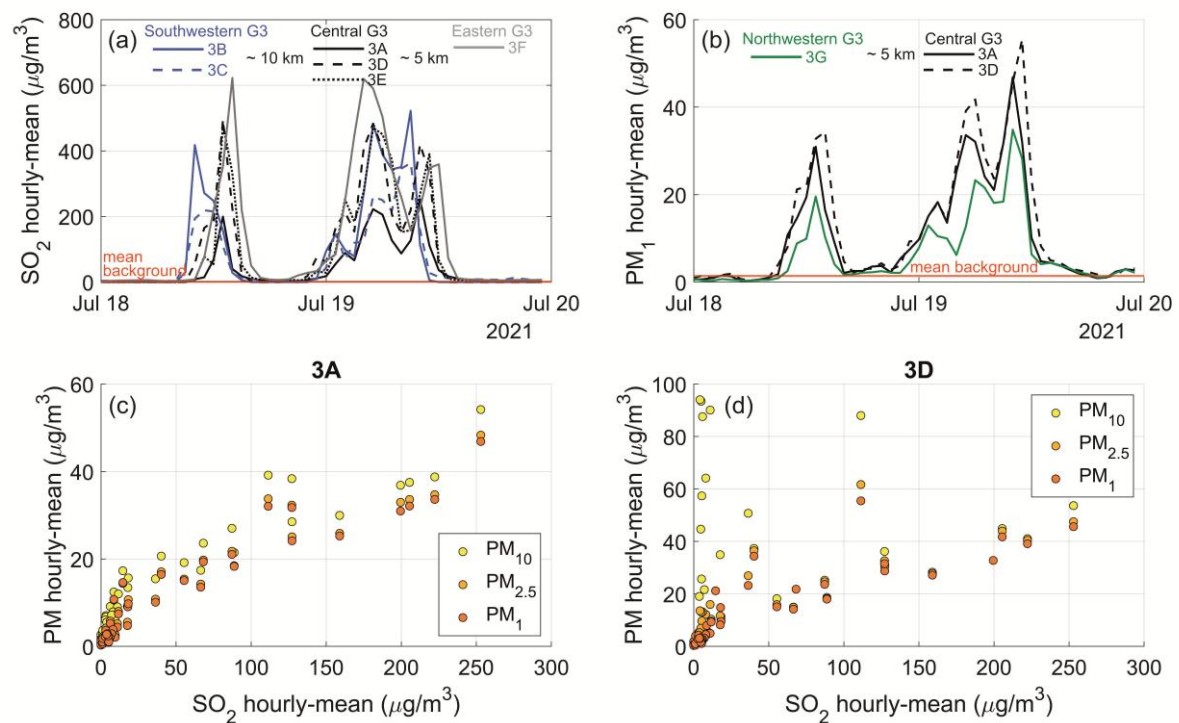

**Figure 7: SO₂ and PM concentrations (µg/m³, hourly-mean) during a 'fresh' volcanic plume advection episode in the Reykjavík capital area (G3) on 18–19 July 2021. Stations G3-A to G3-F are regulatory monitoring sites, and the figure indicates their respective locations within Reykjavík (southwestern, central, eastern, and northwestern), along with approximate distances between them. Panel (a): SO₂ hourly-mean time series. Panel (b): PM₁ hourly-mean time series. Panel (c): Scatter plot of concentrations of SO₂ and PM₁₀, PM₂.₅ and PM₁ at station 3A, which measured all four pollutants. Panel (d): Scatter plot of concentrations of SO₂ and PM₁₀, PM₂.₅ and PM₁ at station 3D, which measured all four pollutants.**

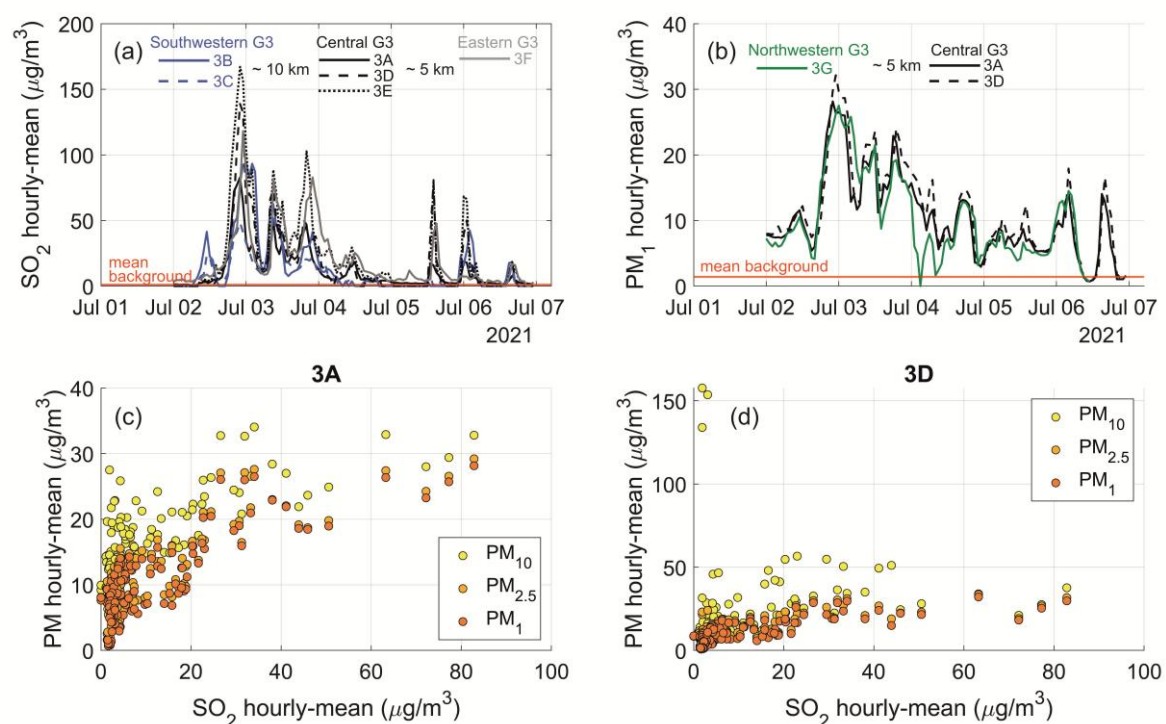

Figure 8: SO₂ and PM concentrations (µg/m3) during a 'mature' volcanic plume advection episode in Reykjavík capital area (G3) 2-6 July 2021. 3A to 3F are names of regulatory stations and the figure indicates their respective locations within Reykjavík (southwestern, central, eastern, and northwestern) and the approximate distance between them. Panel (a): SO₂ hourly-means time series. Panel (b): PM₁ hourly-means time series. Panel (c): Scatter plot between concentrations of SO₂ and PM₁₀, PM₂.₅ and PM₁ at station 3A, which measured all of these pollutants. Panel (d): Scatter plot between concentrations of SO₂ and PM₁₀, PM₂.₅ and PM₁ at station 3D, which measured all of these pollutants.

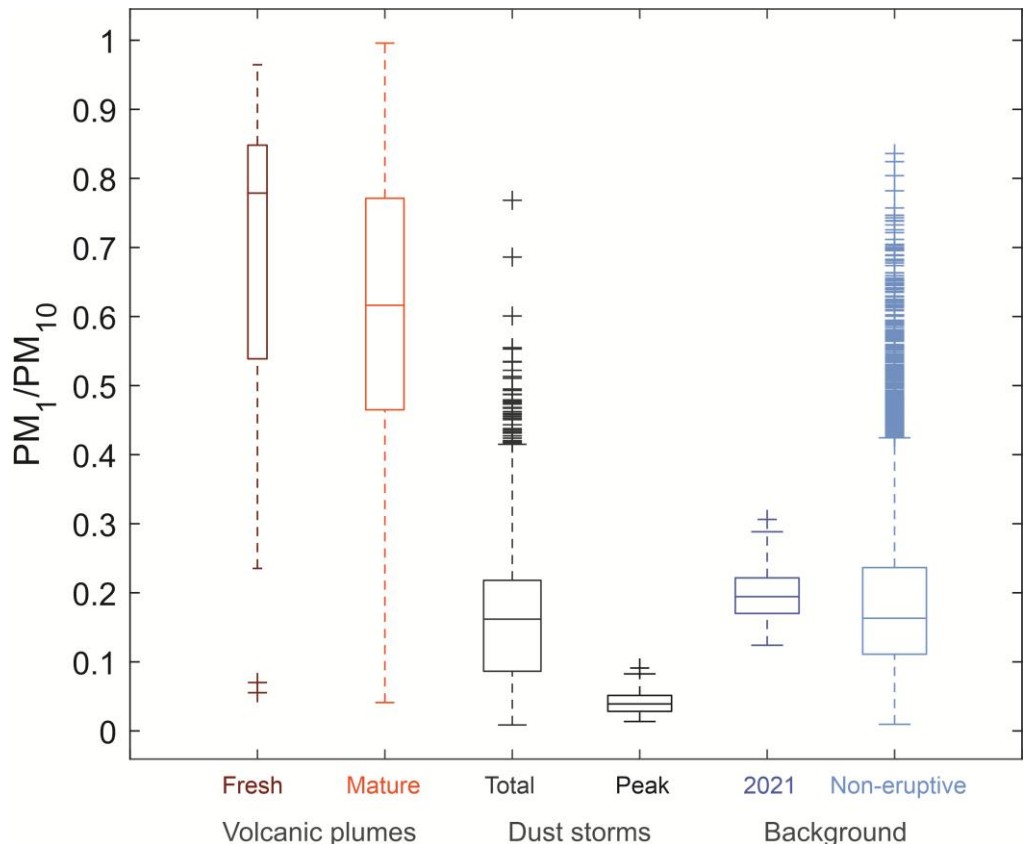

**Figure 9: Variability in PM$_1$/PM$_{10}$ concentration ratios associated with different pollution sources in the Reykjavík capital area.** Data represent hourly-means from stations measuring both size fractions (G3-A, G3-D, G3-G) and are shown as box-and-whisker plots: boxes indicate the interquartile range (IQR), whiskers extend to ±2.7σ from the mean, and crosses mark statistical outliers beyond this range. The median is shown as a horizontal line within each box. 'Volcanic plumes': periods during the 2021 eruption when the plume was advected toward Reykjavík (for definitions of 'fresh' and 'mature' plumes see Section 3.3). Data include one prolonged fresh plume event (>24 h) and three discrete mature plume events, as mature plumes exhibit greater variability in PM size ratios (Ilyinskaya et al., 2017). 'Dust storms': two Icelandic highland desert storms (~200 km source distance) affecting Reykjavík in 2021; 'total' refers to the full duration of dust storm events with PM above background (PM$_{10}$ > 10 µg m$^{-3}$), while 'peak' includes only hours with highly elevated PM (PM$_{10}$ > 50 µg m$^{-3}$). 'Background': representative summer conditions; '2021' refers to eruption-period without volcanic plume influence; 'Non-eruptive' covers summer periods in 2020 and 2022. Table A1 provides the event timings and mean ratios for PM$_1$/PM$_{10}$, PM$_1$/PM$_{2.5}$ and PM$_{2.5}$/PM$_{10}$.

## 3.3 Fine-scale temporal and spatial variability in $SO_2$ and $PM_1$ peaks

The dense regulatory monitoring network located 9–35 km from the eruption site (clusters G2 and G3, Fig. 1) revealed fine-scale variability in $SO_2$ concentrations at these relatively distal locations. Five out of six stations on the Reykjanes peninsula (monitoring $SO_2$ only) were positioned north and northwest of the eruption site, within the most common wind direction (wind rose in Fig. A13). Despite being only 3–16 km apart, two of these stations—G2-E and G2-F—recorded 25 and 31 hourly $SO_2$ exceedance events, respectively, while G2-B, G2-C, and G2-D recorded between 0 and 6 events (Fig. 3). To ensure this pattern was not an artifact of staggered station deployment, we recalculated exceedance events starting from 7 May 2021, the date by which all G2 stations were operational. The results remained consistent: G2-E and G2-F recorded 7 and 26 events, respectively, while G2-B, G2-C, and G2-D recorded between 0 and 6 events. The spatio-temporal difference between the two 'high-exceedance' stations—G2-E and G2-F, located within 5 km of each other—is also noteworthy. During the first seven weeks of the eruption (19 March – 7 May 2021), G2-E recorded 18 of its 25 total exceedance events, while G2-F recorded only 5 of its 31.

Figure 10 illustrates one such episode of fine-scale variability in $SO_2$ concentrations between the G2 stations on Reykjanes peninsula (28–30 May 2021). During this event, the volcanic pollution cloud 'migrated' between the closely spaced stations G2-C, G2-D, and G2-E (separated by ~2 km). The plume first reached G2-C, then shifted to G2-D and G2-E, with G2-D recording nearly twice the peak concentration of G2-E. This demonstrates that the edges of the volcanic pollution cloud at ground level were sharply defined.

Stations in the Reykjavík capital (area G3), located 25–35 km from the eruption site and within <1–10 km of one another, also recorded fine-scale variability in pollutant concentrations—even at this relatively large distance from the source. The most significant volcanic plume advection episode in this area occurred on 18–19 July 2021, during which the G3 stations cumulatively recorded 21 $SO_2$ hourly mean air quality exceedance events—out of the 23 total exceedances recorded throughout the entire eruption. This episode revealed pronounced spatio-temporal variability in volcanic pollutant concentrations. Figure 7 illustrates the variation in $SO_2$ and $PM_1$ abundances during this episode, shown as time series (Figs. 7a–7b) and as concentration ratios (Figs. 7c–7d). This discussion focuses on $PM_1$ rather than $PM_{2.5}$ and $PM_{10}$ because $PM_1$ was more pronounced in the volcanic air pollution, as discussed in the previous sections. Both $SO_2$ and $PM_1$ were significantly elevated above background levels at all G3 stations during the advection episode. Stations G3-A and G3-E, located within 1 km of each other, showed notable differences: G3-E recorded a maximum $SO_2$ concentration of 480 µg/m³ and five exceedance events, while G3-A recorded a peak of 250 µg/m³ and no exceedances (Figs. 3 and 7a). Similar fine-scale differences were observed in $PM_1$: for example, G3-D recorded up to twice the $PM_1$ hourly mean concentrations of G3-G during the same episode (Fig. 7b). The relative proportions of $SO_2$ and $PM_1$ during this episode also varied strongly between the two stations that measured both pollutants (G3-A and G3-D). The peak hourly mean $SO_2$ concentration differed by nearly a factor of two between the stations (Fig. 7a), whereas peak $PM_1$ hourly means differed by no more than 20% (Fig. 7b). During the advection episode, both pollutants exhibited three principal concentration peaks. The first peak, on 18 July at 13:00, corresponded to the highest

SO$_2$ concentration recorded at station G3-D. The final peak, on 19 July at 23:00, marked the highest PM$_1$ concentration at the same station (Figs. 7a–7b). Topographic elevation differences are unlikely to explain this spatial variability, as most G3 stations are located between 10 and 40 m above sea level (a.s.l.), with G3-F at 85 m a.s.l. One potential contributing factor could be the channelling or downwash of air currents by urban buildings—a process that may be particularly relevant in central Reykjavík. This warrants further investigation, such as through fine-scale dispersion modelling, but is beyond the scope of this study due to the challenges with accurately simulating relatively small volcanic plumes.

Supplementary Figures S1 and S2 show animations of the simulated dispersion of volcanic SO$_2$ at ground level during the two pollution episodes discussed in this section, 28-30 May and 18-19 July 2021. The simulations were produced by a dispersion model used operationally for volcanic air quality advisories during the eruption by the Icelandic Meteorological Office (IMO) (Barsotti, 2020; Pfeffer et al., 2024). As discussed by Pfeffer et al. (2024), the model had a reasonable skill in predicting the general plume direction but relatively low accuracy in simulating ground-level SO$_2$ concentrations for the 2021 eruption (Pfeffer et al., 2024).The model results are included here for qualitative purposes—as a binary yes/no indicator of potential plume presence at ground level. The sharp ground-level movement and boundaries of the plume during the 28–30 May episode were captured reasonably well by the model (Supplementary Figure S1), but the larger episode on 18-19 July was not reproduced by the model. This highlights the challenges of accurately simulating ground-level dispersion of volcanic emissions from eruptions like Fagradalsfjall 2021, as well as other small but highly dynamic natural and anthropogenic sources (Barsotti, 2020; Pfeffer et al., 2024; Sokhi et al., 2022). High-resolution observational datasets, including those presented here, can support improvements in dispersion model performance.

We also examined fluctuations in SO$_2$ and PM$_1$ during an advection episode of a chemically mature volcanic plume—locally known as *móða* (or *vog* in English, meaning volcanic smog)—in the Reykjavík capital area between 2 and 6 July 2021 (Fig. 8). A chemically mature plume has undergone significant gas-to-particle conversion of sulfur in the atmosphere and, as shown by Ilyinskaya et al. (2017), may be advected into populated areas several days after the initial emission. Compared to a fresh plume (Figs. 7c–7d), the mature plume (Figs. 8c–8d) is characterized by a higher PM/SO$_2$ ratio, with SO$_2$ elevated above background levels to a variable degree—sometimes only slightly (Ilyinskaya et al., 2017). Conditions that typically facilitate the formation and accumulation of *móða* include low wind speeds, high humidity, and intense solar radiation. Based on these factors, the 2–6 July episode was identified by the IMO as *móða* at the time of the event, and a public air quality advisory was issued. Figures 8c–8d show that during the *móða* episode, PM$_1$ was frequently elevated without a correspondingly high increase in SO$_2$. While SO$_2$ peaks were well-defined, PM$_1$ remained consistently elevated above background levels throughout the entire episode, with less prominent individual concentration peaks. This suggests that PM$_1$ may ground more persistently than SO$_2$—an observation that could be tested in future studies using high-resolution dispersion modelling near the surface.

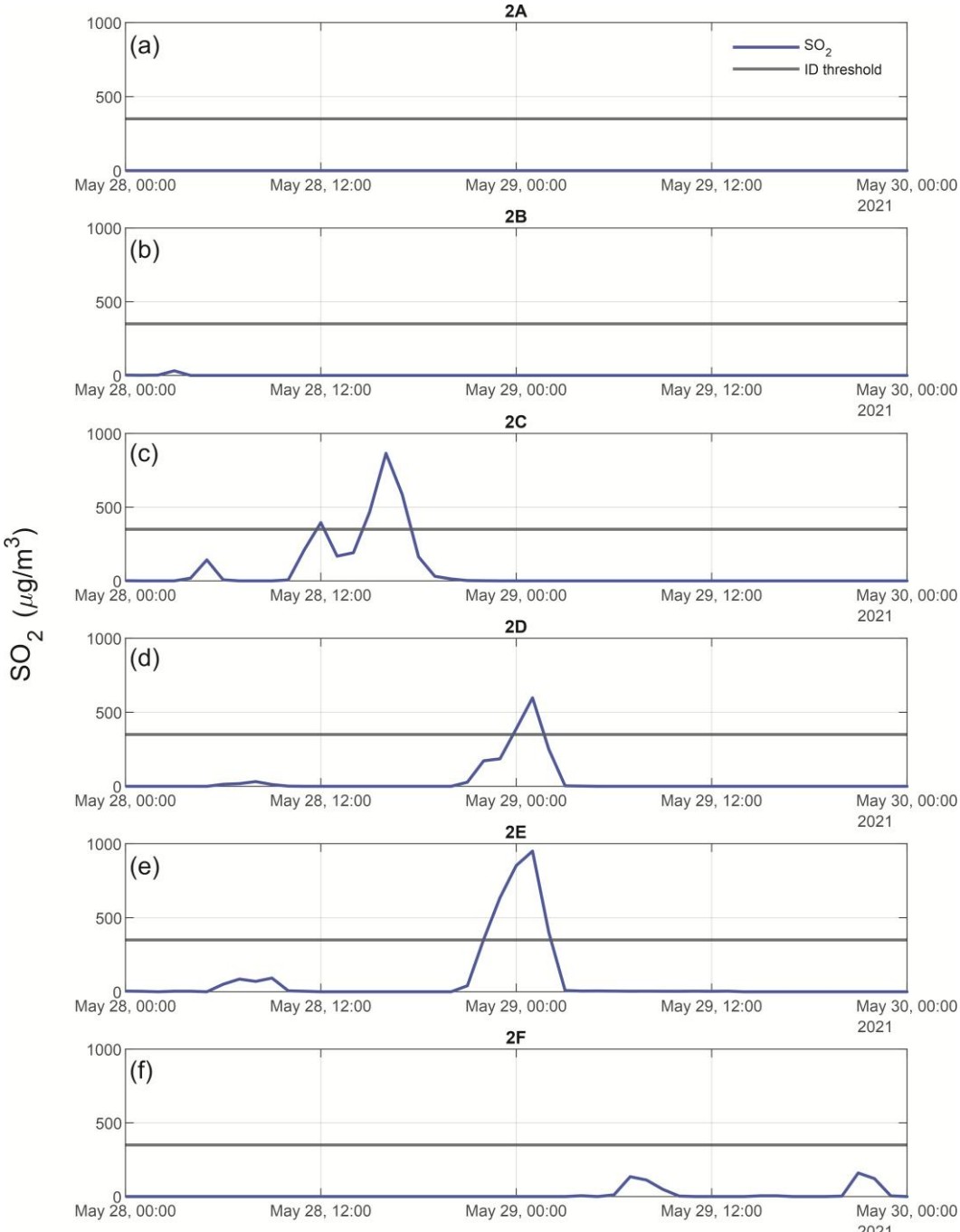

**Figure 10: Spatial and temporal variability in SO₂ concentrations (µg/m³, hourly-mean) between monitoring stations on the Reykjanes peninsula (G2) during 28–30 May 2021. The Icelandic Directive (ID) air quality threshold for hourly SO₂ concentrations (350 µg/m³) is indicated by a black horizontal line. Panel (a): Station G2-A. Panel (b): Station G2-B. Panel (c): Station G2-C. Panel (d): Station G2-D. Panel (e): Station G2-E. Panel (f): Station G2-F.**

## 3.4 Estimates of population exposure and implications for health impacts

### 3.4.1 Exposure of residents

We assessed the frequency of exposure to $SO_2$ concentrations above the ID air quality threshold (350 µg/m³ hourly-mean) in populated areas G1, G2 and G3 using the data from the regulatory-grade network. Based on available evidence in volcanic areas, exceedances of this threshold are associated with adverse health effects (Carlsen et al., 2021a, b). The exceedance of the $SO_2$ air quality threshold was also a proxy for exposure to elevated PM concentration, since the volcanic pollution episodes contained elevated levels of $SO_2$, $PM_1$ and $PM_{2.5}$—and to a lesser extent, $PM_{10}$ (Figs. 7 and 8).

Population data for Iceland in the year 2020 were obtained from Statistics Iceland (2022) and were considered representative for 2021. Data were collected at the municipal level and included both total population and age-specific demographics. Municipality-level population datasets are relatively easy to obtain and are therefore frequently used in population exposure analyses (Caplin et al., 2019), but there are limitations to the resolution due to significant fine-scale spatial variations such as reported in this study.

In 2020, Iceland had a population of 369,000. Of this total, 6% were aged ≤ 4 years and 15% were aged ≥ 65 years—age groups which have been shown to be more vulnerable to volcanic air pollution (Carlsen et al., 2021b, a). A total of 263,000 people—equivalent to 71% of the national population—resided within 50 km of the Fagradalsfjall eruption site, where most $SO_2$ air quality threshold exceedances occurred. Figure 11 presents municipality-level population data for this area, including total population and density, the number and density of individuals in vulnerable age groups, the locations of hospitals, and

the number of ID air quality threshold exceedances recorded at monitoring stations.

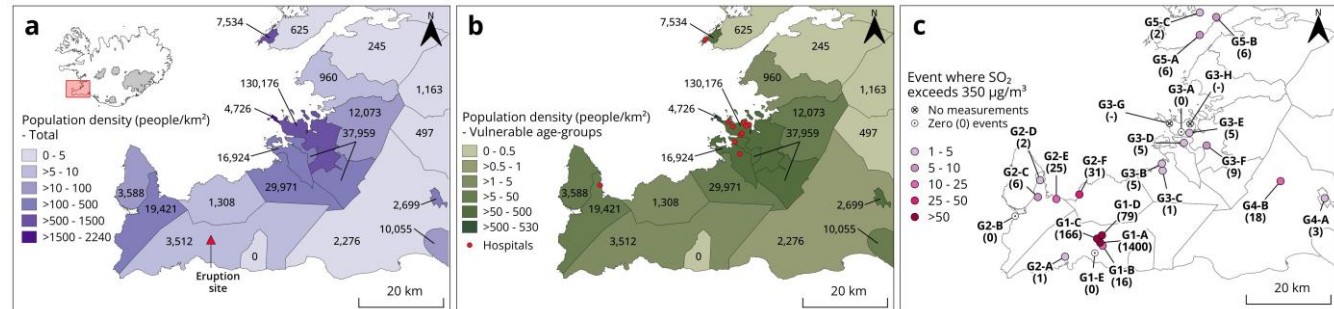

**Figure 11: Potential exposure of the residents in the densely populated southwestern part of Iceland, including the Reykjavík capital area (G3) to above-threshold $SO_2$ concentrations. Population data are from Statistics Iceland for 2020. Panel (a): The number of residents and the population density at the municipality level. The number of residents is shown for each municipality, and the colour scale represents the population density (*n* of people/km² in each municipality). Panel (b): Potentially vulnerable age groups (≤ 4 years and ≥ 65 years of age). The number of people in the vulnerable age groups is shown for each municipality, and the colour scale represents the population density (*n* of people/km² in each municipality). The map also shows the location of hospitals. Panel (c): Number of times when the $SO_2$ concentrations exceeded the ID air quality threshold of 350 µg/m³ hourly-mean during the eruption period as measured by the regulatory stations in areas G1, G2 and G3. Source and copyright of basemap and cartographic elements: Icelandic Met Office & Icelandic Institute of Natural History**

The Reykjavík capital area had approximately 210,000 residents (60% of the total population), a high density of individuals in the potentially more vulnerable age groups, and a large number of hospitals (area G3 on Fig. 11). Air quality stations in this

densely populated capital area recorded between 0 and 9 threshold exceedance events during the eruption period. Fine-scale spatial differences in ground-level pollutant concentrations (discussed in Section 3.3) may have played a critical role in determining people's exposure. For example, one of the largest hospitals in the country was located approximately equidistant (~2 km) from stations G3-A and G3-E, which recorded 0 and 5 $SO_2$ exceedance events, respectively. As a result, it remains unknown how frequently individuals at the hospital were exposed to above-threshold $SO_2$ levels. Similarly, the hospital closest to the eruption site—located about 20 km away—was situated between two monitoring stations, G2-D and G2-E, which recorded markedly different numbers of exceedance events: 2 and 25, respectively (Fig. 11). These examples highlight the importance of spatial resolution in air quality monitoring for accurately assessing population exposure.

The most frequent exposure to potentially unhealthy $SO_2$ levels occurred predominantly within a 20 km radius of the eruption site, particularly in municipalities on the Reykjanes peninsula (Fig. 11). In this area (G2), up to 31 exceedance events were recorded—surpassing the annual threshold of 24 exceedances ($n = 24$). However, exposure estimates based solely on place of residence may not fully capture individual exposure, especially for working adults who commute. For example, station G2-A in the township of Grindavík recorded only one exceedance event, yet many residents worked at Keflavík Airport, where higher $SO_2$ levels were observed (five exceedance events at station G2-C). Conversely, residents in the town of Vogar (station G2-E, 25 exceedance events) who may have commuted to the Reykjavík capital area—where fewer exceedances were recorded (0–9 events)—may have experienced lower actual exposure than estimated based on residence alone. In contrast, exposure estimates for children are likely more accurate, as most attend schools within walking distance or a short commute from home. The same applies to long-term hospital inpatients, whose exposure is closely tied to the location of the healthcare facility.

From a nationwide public health perspective, it was fortunate that volcanic pollutants were predominantly transported to the north and northwest of the eruption site. This atmospheric transport pattern likely mitigated the frequency of $SO_2$ pollution episodes in the densely populated capital area, situated to the northeast of the eruption site. Supplementary Figure S3 illustrates the total probability of above-threshold $SO_2$ concentrations at ground level during the eruption, as simulated by the IMO dispersion model (Pfeffer et al., 2024). As outlined in Section 3.3, these simulations are used here solely to provide a qualitative indication of the broad plume direction at ground level. The modelled dispersion patterns are consistent with observational data, indicating that the plume most frequently grounded to the north and northwest of the eruption site, and more rarely in the capital area (Fig. S3).

Based on the available evidence, it is possible that the 2021 eruption may have led to adverse health impacts among exposed populations. Epidemiological studies by Carlsen et al. (2021a, b) on the 2014–2015 Holuhraun eruption demonstrated a measurable increase in healthcare utilisation for respiratory conditions in the Reykjavík capital area, associated with the presence of the volcanic plume. Exposure to above-threshold $SO_2$ concentrations was linked to approximately 20% increase in asthma medication dispensations and primary care visits. During the Fagradalsfjall eruption, $SO_2$ concentrations in populated areas reached levels broadly comparable to those observed during the larger but more distal Holuhraun eruption. Holuhraun emissions led to 33 exceedances of the $SO_2$ air quality threshold in Reykjavík, with hourly-mean concentrations peaking at 1400 µg/m³ (Ilyinskaya et al., 2017). In comparison, the Fagradalsfjall eruption caused 31 exceedances, with a

maximum of 2400 µg/m³ $SO_2$ recorded in the community of Vogar (station G2-F). Up to 18 $SO_2$ threshold exceedances were also recorded in areas within approximately 50 km of the eruption site (areas G1–G5). All areas that recorded above-threshold pollutant concentrations may have experienced adverse health effects.

Although the monitored regions in North and East Iceland (areas G6 and G7) did not register threshold exceedances, potential adverse health impacts in these areas cannot be ruled out. As reported by Carlsen et al. (2021b), even relatively small above-background increases in $SO_2$ levels during Holuhraun were associated with small but statistically significant rises in healthcare usage—approximately a 1% increase per 10 µg/m³ $SO_2$—suggesting the absence of a safe lower threshold.

Given the limited number and scope of health impact studies on previous volcanic eruptions, the potential health implications

discussed here should be further investigated through dedicated epidemiological and/or clinical studies focused specifically on the Fagradalsfjall event. Moreover, existing health studies from volcanic regions have primarily concentrated on short-term exposure (hourly and daily), with a gap in research of potential long-term effects. Since the 2021 eruption, 11 additional eruptions of similar style and in the same geographic area have occurred. Although each event has been relatively short-lived—ranging from several days to several months—their cumulative impact on public health may be chronic as well as acute, and

thus warrants comprehensive investigation.

Carlsen et al. (2021a) found that when volcanic air pollution events from the Holuhraun eruption were successfully forecast and public advisories were issued, the associated negative health impacts were reduced compared to events that were not forecast. In Iceland, residential buildings are predominantly well-insulated concrete structures with double-glazed windows, offering substantial protection from outdoor air pollution. However, under normal conditions, windows are kept open for

ventilation, facilitated by the availability of inexpensive geothermal heating. Additionally, it is common practice for infants to nap outdoors in prams, and for school-aged children to spend breaks outside. Public advisories included simple, easily implemented measures such as keeping windows closed and minimizing outdoor exposure for vulnerable individuals. Given that such basic societal actions have been shown to be effective, it is likely that further improvements in pollution detection—particularly enhancements in spatial resolution—and more effective communication strategies could provide additional

protection to the population.

### 3.4.2 Exposure of eruption site visitors

An interesting aspect of the eruption was that it was generally considered a very positive event by the Icelandic public (Ilyinskaya et al., 2024), and even though it took place in an uninhabited location the site became akin to a densely populated area due to the extremely high number of visitors. The mountainous area had no infrastructure before the eruption and was

700 only accessible by rough mountain tracks. It was unsuitable for an installation of a regulatory air quality network but there were serious concerns about the hazard posed to the visitors by potentially very high $SO_2$ concentrations. In response, national and local authorities undertook significant efforts to mitigate hazards associated with both volcanic activity and general outdoor hazards. A network of three footpaths was established, originating from designated parking areas (Fig. 12). These footpaths were modified multiple times throughout the eruption as the lava field expanded and optimal viewing locations shifted (Barsotti

et al., 2023). In this study, we evaluate the deployment of eruption-response LCS as a means to minimize exposure to hazardous SO$_2$ levels.

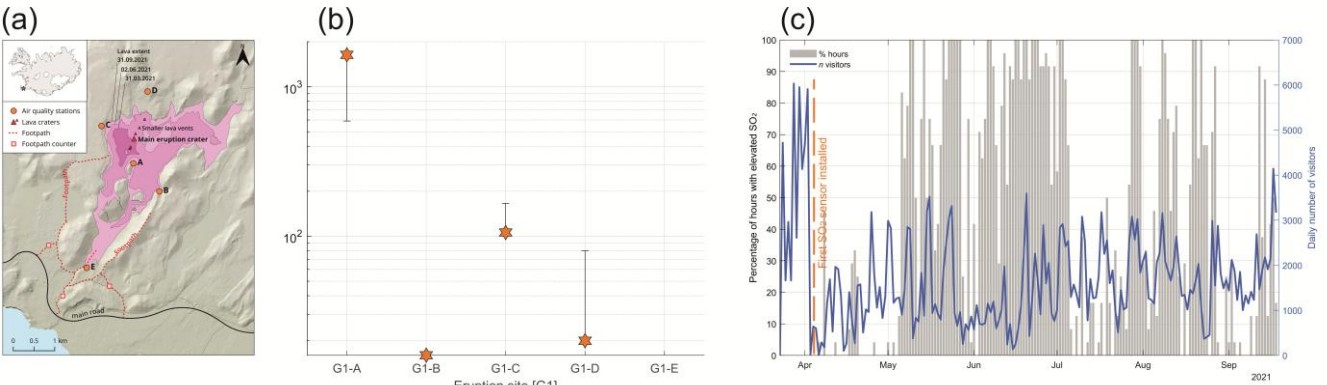

Figure 12: Visitor numbers and potential exposure to elevated SO$_2$ at the Fagradalsfjall eruption site between 24 March and 18 September 2021. Panel (a): Topographic map of the eruption site showing crater locations, the evolving lava field extent, five LCS stations (A–E), primary visitor footpaths, and footpath visitor counters. Panel (b): Total hours with elevated SO$_2$ concentrations recorded at each LCS station. Error bars indicate measurement uncertainty; the y-axis is logarithmic. Panel (c): Daily visitor counts (n of people) and daily percentage of time with elevated SO$_2$ (elevated hours/24 × 100). Grey bars show the daily max–min range across the five LCS stations. The LCS data should be interpreted only as indicative; 'elevated SO$_2$' levels do not represent confirmed air quality exceedances. Source and copyright of basemap and cartographic elements: Icelandic Met Office & Icelandic Institute of Natural History.

Automated footpath counters were installed by the Icelandic Tourist Board on 24 March 2021, with one device placed on each of the main footpaths leading to the eruption site and designated viewpoints (Fig. 12). These counters (PYRO-Box by Eco Counter) have a reported accuracy of 95% and a sensing range of 4 meters. The visitor numbers presented here represent a minimum estimate. While the majority of visitors used the established footpath network, some individuals may have walked outside the detection range of the counters and were therefore not recorded. Additionally, visitors arriving via helicopter sightseeing tours, children being carried, and individuals with authorized vehicle access (e.g., scientists and rescue personnel) were not included in the count. The visitor data also lacked demographic information, preventing any assessment of exposure among more vulnerable age groups. In addition, there is no data on whether people visited the eruption multiple times and were therefore potentially cumulatively more exposed. During the visitor-counting period (24 March to 18 September 2021), the eruption site was visited by approximately 300,000 people, averaging 1,600 visitors per day (Fig. 12). The highest visitor numbers occurred in the early weeks of the eruption, coinciding with the Easter holiday period, with a daily average of 3,300 visitors and a peak of 6,000 on 28 March.

The five eruption-response LCS were strategically deployed along the main footpaths (Fig. 12a) to ensure proximity to visitors. Figure 12b shows the number of times at each LCS station that hourly-mean SO$_2$ was recorded as elevated (see Section 2.2 for definition of 'elevated' and the sensor uncertainty). There was high variability between the stations, and therefore high variability in the potential exposure of the visitors to elevated SO$_2$ depending on where they were. Station G1-A, located closest

to the active craters, recorded elevated $SO_2$ between 600 and 1600 times. Stations G1-B, G1-C, and G1-D recorded elevated $SO_2$ between 20 and 110 times, while G1-E did not register any highly elevated periods. Stations G1-C and G1-D were more frequently located downwind of the active vents, as supported by the wind rose diagram in Fig. A13. Additionally, based on visual observations during this eruption and similar fissure eruptions, a volcanic plume can occasionally collapse and spread laterally. This leads to extremely high concentrations of $SO_2$ even at locations in close vicinity of but upwind of the volcanic vent.

During the course of the 2021 eruption and subsequent events (2022–2025), $SO_2$ measurements from the LCS stations were used by the IMO to produce hazard maps around the active and potential eruption sites, with hazard zones defined by the distances at which elevated $SO_2$ was detected (Icelandic Meteorological office, 2025). Visitors were clearly advised to remain upwind of the active craters and lava field. The site was staffed by members of the rescue services and/or rangers, who carried handheld $SO_2$ LCS to supplement the semi-permanent sensor network. When $SO_2$ concentrations became elevated, and therefore potentially unhealthy, visitors were urged to relocate to areas with cleaner air. Although no formal health impact studies have been published to date, anecdotal reports in the Icelandic media suggest that only a small number of individuals sought medical attention after visiting the eruption site, citing symptoms related to gas exposure. This likely represents a very small proportion of the total visitor population. Instances of exposure to unhealthy $SO_2$ levels may have occurred for several reasons: not all visitors were in proximity to a sensor during their visit, and rapid shifts in wind direction or changes in eruption dynamics occasionally transported $SO_2$ into areas that had previously been unaffected.

In conclusion, the deployment of the LCS network at the eruption site for the purposes of alerting people to potentially-high $SO_2$ concentrations was likely valuable given the high frequency of elevated $SO_2$ concentrations and the large number of visitors in a confined area. However, the absence of regulatory-grade calibration prevented any quantitative assessment of individual exposure to hazardous pollutants. To obtain high-quality datasets with LCS, regular and frequent field calibration against regulatory instruments is essential. However, such calibration is typically feasible only during short-term campaigns at reasonably accessible locations. In this crisis-response scenario, the challenging terrain and limited accessibility of the eruption site precluded field calibration. The primary concerns associated with uncalibrated LCS in emergency contexts are false negatives—where the sensor underreports concentrations that exceed health thresholds—and false positives—where the sensor overreports concentrations that are actually below threshold. False negatives pose a problem by failing to alert individuals to hazardous conditions, while repeated false positives may undermine public trust and reduce compliance with safety advisories.

## 4 Conclusions

The 2021 eruption of Fagradalsfjall marked the onset of a prolonged eruptive phase on the Reykjanes peninsula, with 11 subsequent eruptions occurring through to the time of writing, and continued volcanic unrest. Our findings demonstrate that even a relatively small volcanic event, such as the 2021 eruption, can lead to significant air pollution of $SO_2$ and PM. Due to

its proximity to densely populated areas, the Fagradalsfjall eruption caused elevated pollutant concentrations, and air quality threshold exceedances comparable to those observed during the much larger Holuhraun eruption of 2014–2015. These results suggest that the Fagradalsfjall eruption generated sufficient air pollution that it may have triggered negative health responses, which should be investigated retrospectively or during future events. Moreover, the high frequency of eruptions, and eruption-ignited wildfires in this region since 2021 raises the possibility of chronic exposure, which should also be examined,

particularly given that the ongoing Reykjanes Fires eruptions may continue for several generations.

We showed that even the exceptionally dense, reference-grade air quality monitoring networks in the densely populated part of Iceland (Reykjavík capital and the Reykjanes peninsula) were insufficient to fully capture the fine-scale spatial variability of volcanic air pollution episodes. We recommend augmenting existing networks with well-calibrated low-cost sensors (LCS) to enhance spatial coverage, particularly in sensitive locations such as schools and hospitals, where vulnerable populations

may be at greater risk. Previous studies on the Holuhraun eruption have demonstrated that public advisories on volcanic air pollution can serve as effective health protection measures. Therefore, improving the spatial resolution of air quality monitoring may further enhance public health outcomes by enabling more targeted and timely advice.

Understanding the volcanic air pollution in a uniquely Icelandic event like the Reykjanes Fires has important implications for how we manage and prepare for other eruptions globally. The fine-scale temporal and spatial variability in pollution dispersion

identified in this study highlights the need for further investigation—not only in future Icelandic eruptions but also in other regions exposed to volcanic activity. Enhanced understanding of these dynamics can inform more effective monitoring strategies and public health responses worldwide.

## Appendix A.

**Table A1: Variability in PM$_1$/PM$_{10}$ concentration ratios associated with different pollution sources in the Reykjavík capital area measured by three stations (G3-A, G3-D, G3-G). 'Background': representative summer conditions; '2021' refers to eruption-period without volcanic plume influence; 'Non-eruptive' covers summer periods in 2020 and 2022. 'Volcanic plumes': periods during the 2021 eruption when the plume was advected toward Reykjavík (definitions of 'fresh' and 'mature' plumes in main text, Section 3.3). Data include one prolonged fresh plume event (>24 h) and three discrete mature plume events, as mature plumes exhibit greater variability in PM size ratios. 'Dust storms': two Icelandic highland desert storms (~200 km source distance) affecting Reykjavík; 'total' refers to the full duration of dust storm events with PM above background (PM$_{10}$ > 10 μg m$^{-3}$), while 'peak' includes only hours with highly elevated PM (PM$_{10}$ > 50 μg m$^{-3}$). Station G3-G is listed first, as it is considered the most sensitive to the presence of volcanic plume due to its low background concentrations from local sources. Dates are in the format DD/MM/YYYY.**

| | Start date | Start time | End date | End time | G3-G PM$_1$/PM$_{10}$ | G3-A PM$_1$/PM$_{10}$ | G3-D PM$_1$/PM$_{10}$ | G3-G PM$_1$/PM$_{2.5}$ | G3-A PM$_1$/PM$_{2.5}$ | G3-D PM$_1$/PM$_{2.5}$ | G3-G PM$_{2.5}$/PM$_{10}$ | G3-A PM$_{2.5}$/PM$_{10}$ | G3-D PM$_{2.5}$/PM$_{10}$ |
|---|---|---|---|---|---|---|---|---|---|---|---|---|---|
| Background non-eruptive | 01/05/2020 | 00:00 | 01/09/2020 | 00:00 | 0.16 | 0.15 | 0.13 | 0.44 | 0.43 | 0.22 | 0.35 | 0.34 | 0.61 |
| Background 2021 | 01/04/2021 | 09:00 | 02/04/2021 | 10:00 | 0.17 | 0.19 | 0.24 | 0.41 | 0.43 | 0.45 | 0.42 | 0.43 | 0.54 |
| Fresh plume | 18/07/2021 | 10:00 | 19/07/2021 | 16:00 | 0.65 | 0.68 | 0.7 | 0.9 | 0.92 | 0.84 | 0.72 | 0.73 | 0.78 |
| Mature plume 1 | 28/04/2021 | 08:00 | 29/04/2021 | 20:00 | 0.43 | 0.29 | 0.49 | 0.8 | 0.73 | 0.8 | 0.53 | 0.39 | 0.6 |
| Mature plume 2 | 19/05/2021 | 14:00 | 21/05/2021 | 11:00 | 0.71 | 0.65 | 0.85 | 0.96 | 0.95 | 0.95 | 0.73 | 0.68 | 0.89 |
| Mature plume 3 | 01/07/2021 | 09:00 | 06/07/2021 | 08:00 | 0.67 | 0.59 | 0.65 | 0.91 | 0.88 | 0.84 | 0.74 | 0.66 | 0.74 |
| Desert dust 1 total | 24/05/2021 | 20:00 | 29/05/2021 | 21:00 | 0.11 | 0.11 | 0.14 | 0.31 | 0.3 | 0.3 | 0.33 | 0.32 | 0.39 |
| Desert dust 1 peak | | | | | 0.05 | 0.05 | 0.04 | 0.18 | 0.17 | 0.17 | 0.26 | 0.26 | 0.27 |
| Desert dust 2 total | 03/06/2021 | 09:00 | 04/06/2021 | 11:00 | 0.12 | 0.25 | 0.17 | 0.33 | 0.44 | 0.31 | 0.31 | 0.42 | 0.42 |
| Desert dust 1 peak | | | | | 0.05 | n/a | 0.04 | 0.18 | n/a | 0.16 | 0.27 | n/a | 0.27 |

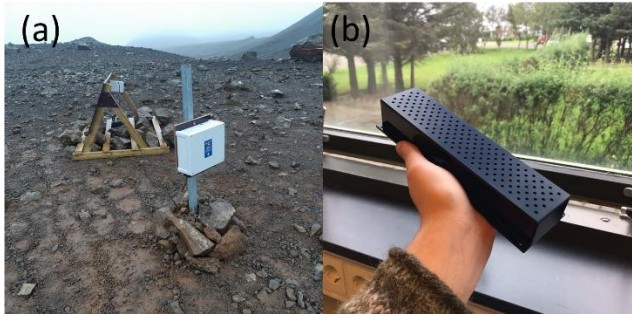

**Figure A1: Lower-cost sensors used for the Fagradalsfjall 2021 eruption. Panel (a) shows the instrument installed in the field. The station was powered by a solar panel (triangular trellis at the back of the photo). The air intake was underneath the instrument (the white box at the front of the image). Panel (b) shows the air intake of the sensor. The air intake was designed in-house at the IMO taking into account local conditions, in particular the weather and dust resuspension. The cover was custom-made from Plexiglass with the sensors recessed behind it to be protected from dust, precipitation, and other potentially damaging environmental factors.**

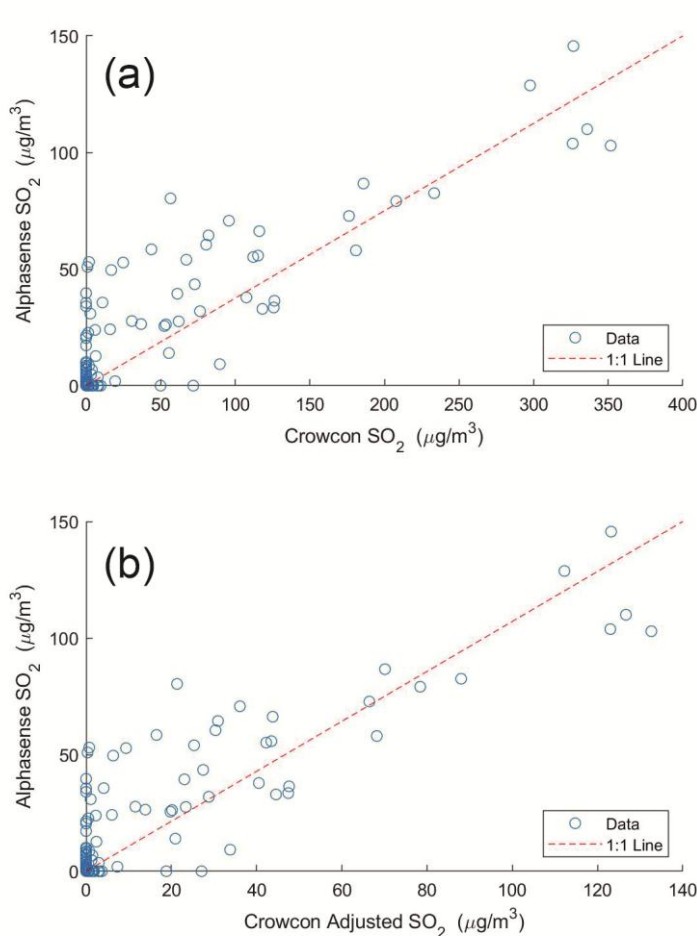

**Figure A2: SO₂ concentrations measured by two types of lower-cost sensors (LCS) used in this study—Alphasense SO₂-B4 and Crowcon XGuard—during a field colocation at the eruption site (6–22 June 2021). Measurements from the two sensors showed a strong linear correlation ($r^2 = 0.70$), but Alphasense reported lower values relative to Crowcon, with a correlation coefficient of 0.38. Panel (a) Correlation of raw data points from the two sensors. Panel (b) Correlation after Crowcon data were adjusted using the correlation coefficient**

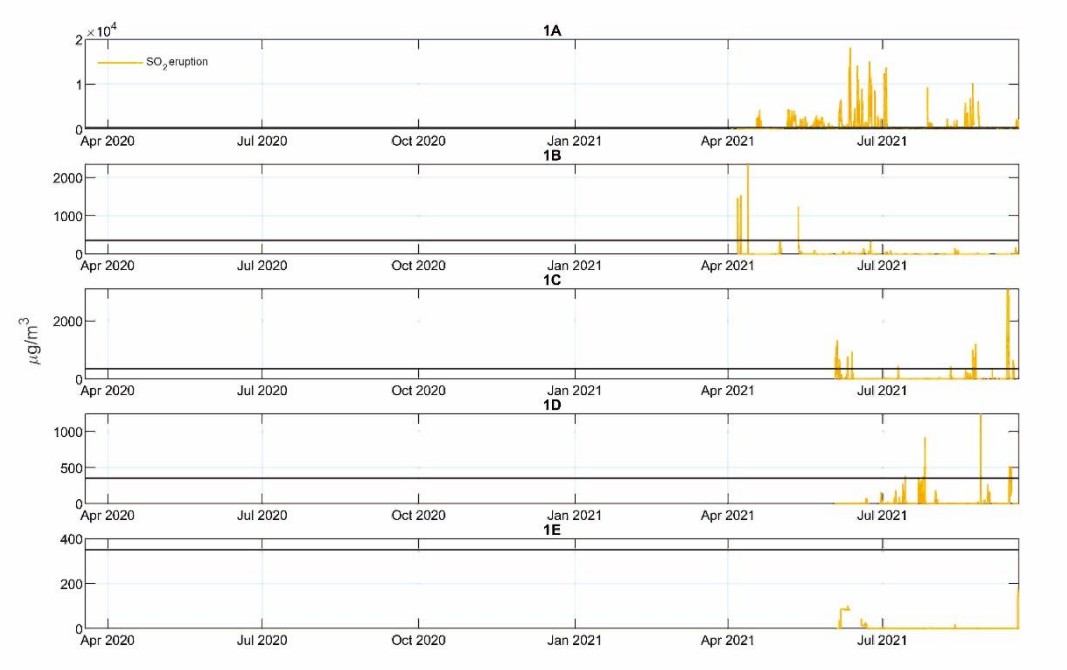

**Figure A3: Time series of hourly-mean concentrations SO₂ (µg/m³), measured by the eruption site stations (G1 A-E) during the 2021 eruption. The stations were not in operation before the eruption and therefore there are no data on pre-eruptive background. The ID air quality threshold of 350 µg/m³ hourly-mean is shown on all panels with a black horizontal line. Note that the eruption-site LCS have low accuracy and were only used in this study to indicate time periods that were over the ID threshold, the absolute concentration values were not included in the analysis.**

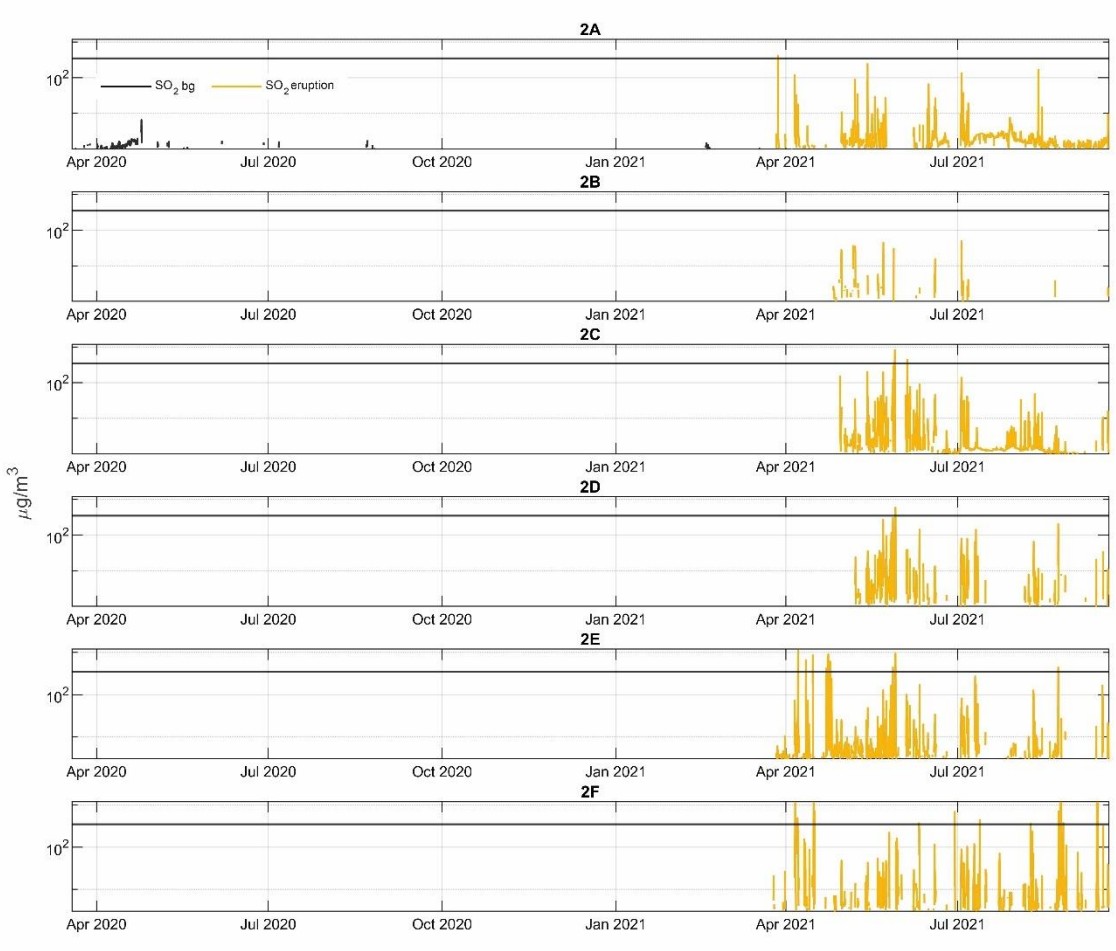

Figure A4: Time series of hourly-mean concentrations SO₂ (µg/m³), measured by Reykjanes peninsula regulatory air quality stations (G2 A-F) during the 2021 eruption and the non-eruptive background in 2020 (bg). The ID air quality threshold of 350 µg/m³ hourly-mean is shown on all panels with a black horizontal line. Please note the logarithmic y-axis scale.

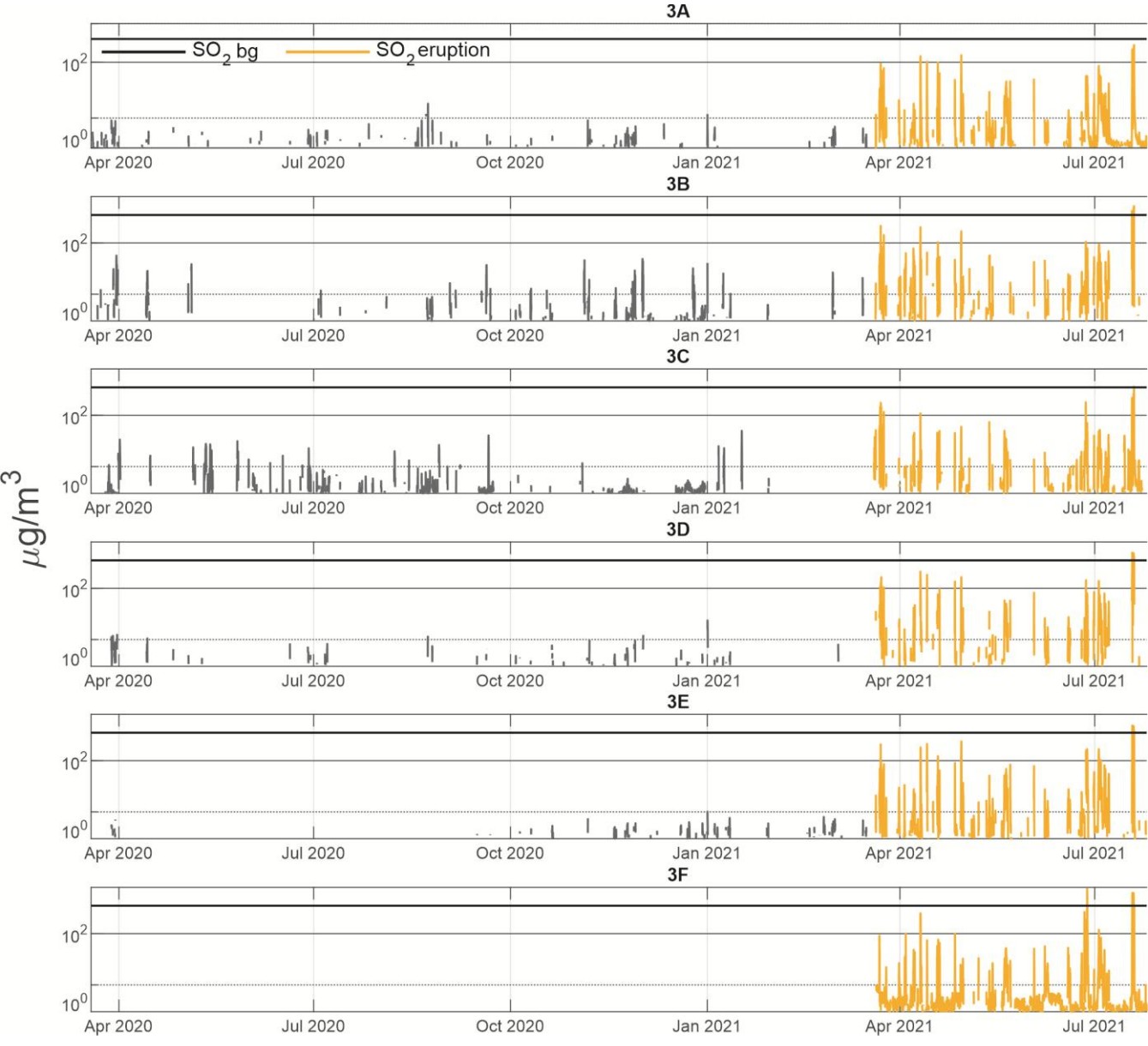

**Figure A5: Time series of hourly-mean concentrations SO₂ (μg/m³), measured by Reykjavík capital area regulatory air quality stations (G3 A-F) during the 2021 eruption and the non-eruptive background in 2020 (bg). The ID air quality threshold of 350 μg/m³ hourly-mean is shown on all panels with a black horizontal line. Please note the logarithmic y-axis scale.**

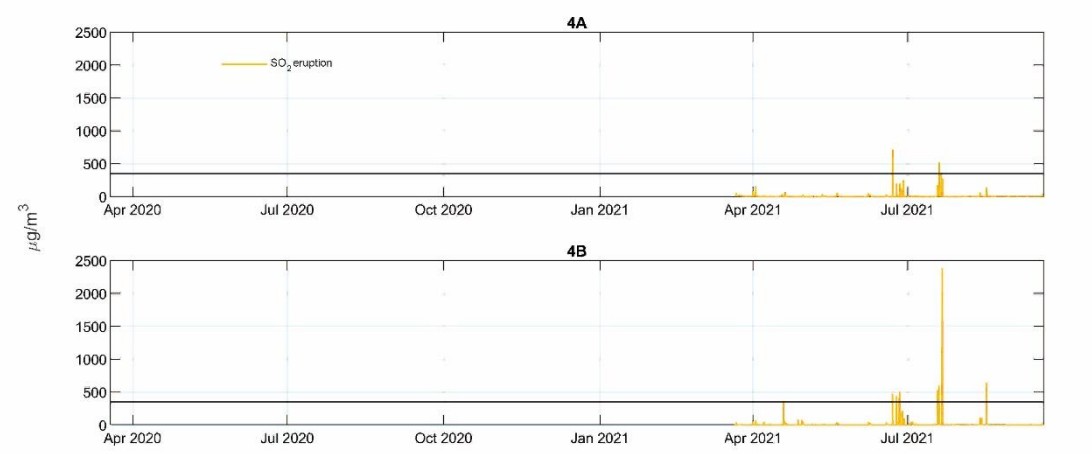

**Figure A6: Time series of hourly-mean concentrations SO₂ (µg/m³), measured in Southwest Iceland by regulatory air quality stations (G4 A-B) during the 2021 eruption. The stations were not in operation before the eruption and therefore there are no data on pre-eruptive background. The ID air quality threshold of 350 µg/m³ hourly-mean is shown on all panels with a black horizontal line.**

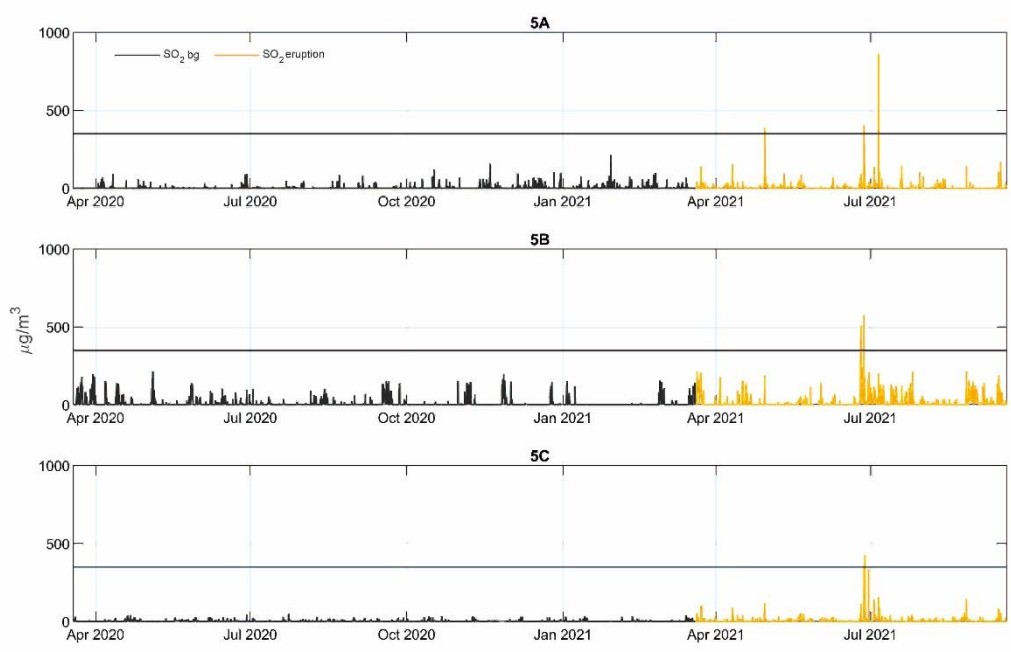

**Figure A7: Time series of hourly-mean concentrations SO$_2$ (µg/m$^3$), measured in Hvalfjörður area by regulatory air quality stations (G5 A-C) during the 2021 eruption and the non-eruptive background in 2020 (bg). The ID air quality threshold of 350 µg/m$^3$ hourly-mean is shown on all panels with a black horizontal line.**

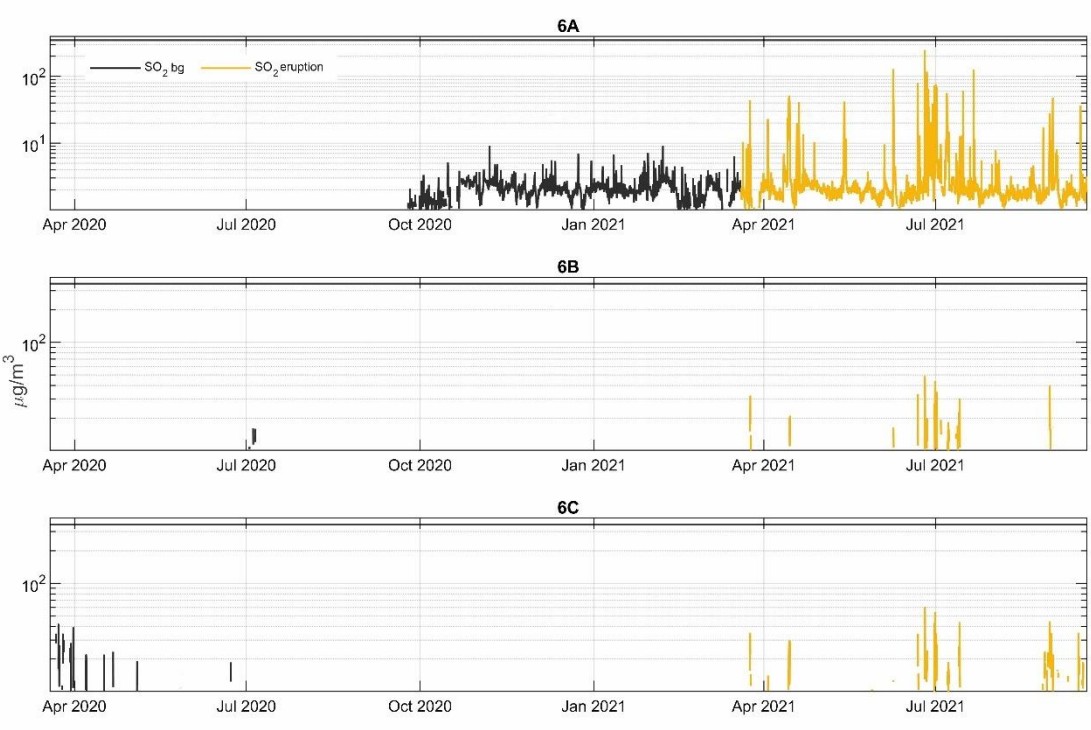

**Figure A8: Time series of hourly-mean concentrations SO$_2$ (μg/m$^3$), measured in North Iceland by regulatory air quality stations (G6 A-C) during the 2021 eruption and the non-eruptive background in 2020 (bg). The ID air quality threshold of 350 μg/m$^3$ hourly-mean is shown on all panels with a black horizontal line. Please note the logarithmic y-axis scale.**

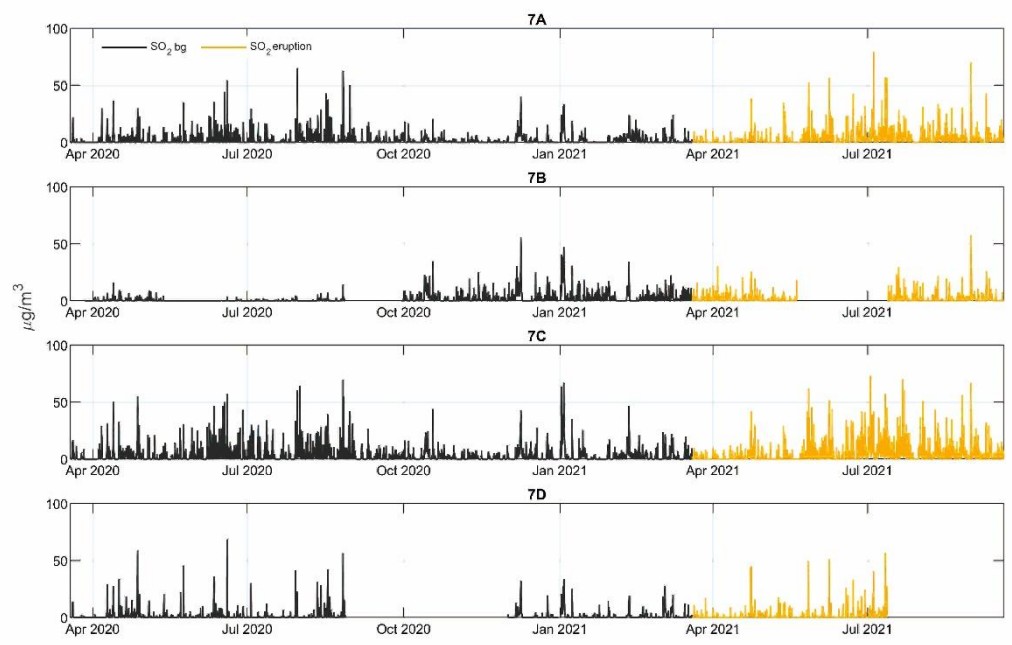

**Figure A9: Time series of hourly-mean concentrations SO₂ (µg/m³), measured in East Iceland by regulatory air quality stations (G7 A-D) during the 2021 eruption and the non-eruptive background in 2020 (bg). The ID air quality threshold of 350 µg/m³ hourly-mean is shown on all panels with a black horizontal line.**

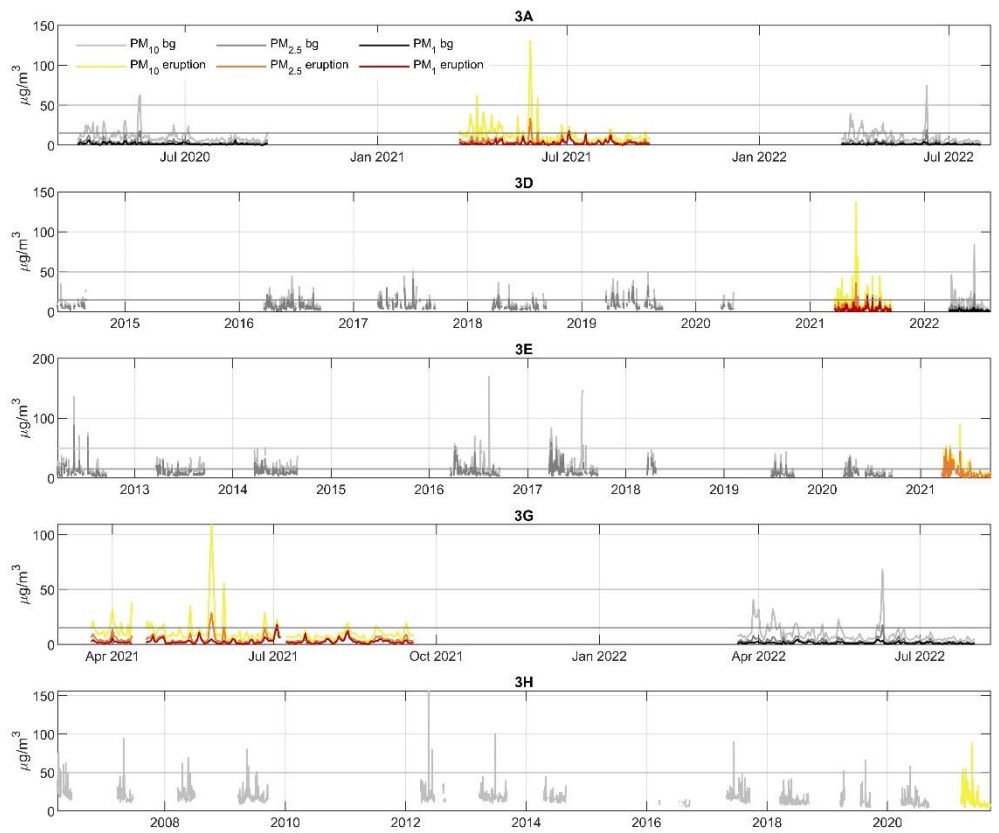

Figure A10: Time series of daily-mean concentrations of PM₁₀, PM₂.₅ and PM₁ (μg/m³) measured in Reykjavík capital area by regulatory air quality stations (G3 A, D, E, G, H) during the 2021 eruption and in the non-eruptive background (bg). The amount of non-eruptive background data varies between stations based on their installation date. The figures only include data for the period 19 March 20:00 – 19 September 00:00 UTC in each year, i.e. the period corresponding to the calendar dates and months of the 2021 eruption. See main text for the justification of this approach. The figures show the ID air quality thresholds for PM₁₀ and PM₂.₅ of 50 and 15 μg/m³ daily-mean, respectively as grey horizontal lines. For PM₁, air quality thresholds have not been determined.

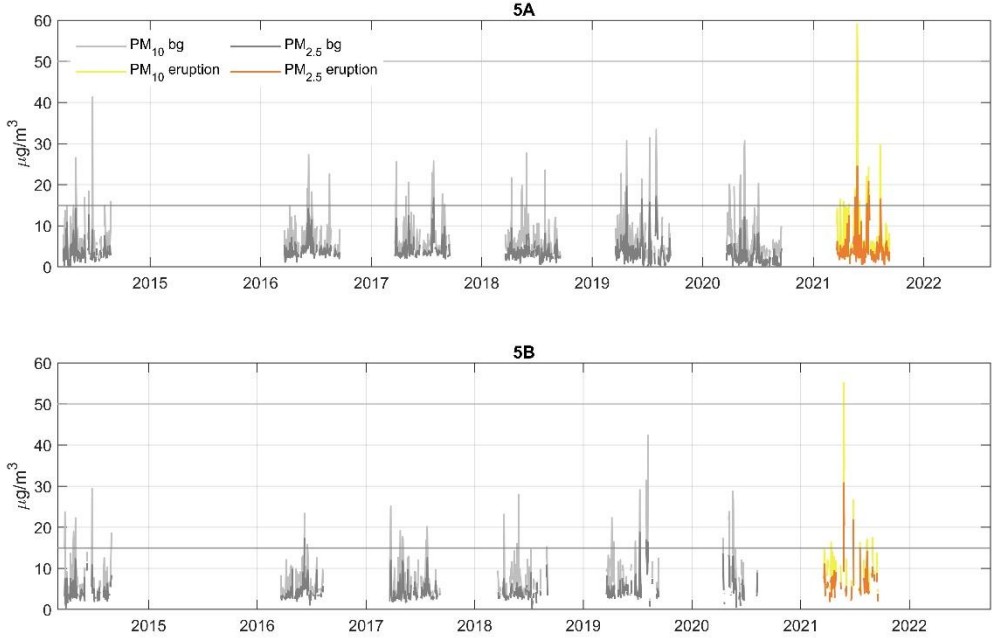

**Figure A11: Time series of daily-mean concentrations of PM$_{10}$, and PM$_{2.5}$ (µg/m$^3$) measured in Hvalfjörður area by regulatory air quality stations (G5 A, B) during the 2021 eruption and in the non-eruptive background (bg). PM$_1$ was not measured at these stations. The amount of non-eruptive background data varies between stations based on their installation date. The figures only include data for the period 19 March 20:00 – 19 September 00:00 UTC in each year, i.e. the period corresponding to the calendar dates and months of the 2021 eruption. See main text for the justification of this approach. The figures show the ID air quality thresholds for PM$_{10}$ and PM$_{2.5}$ of 50 and 15 µg/m$^3$ daily-mean, respectively as grey horizontal lines.**

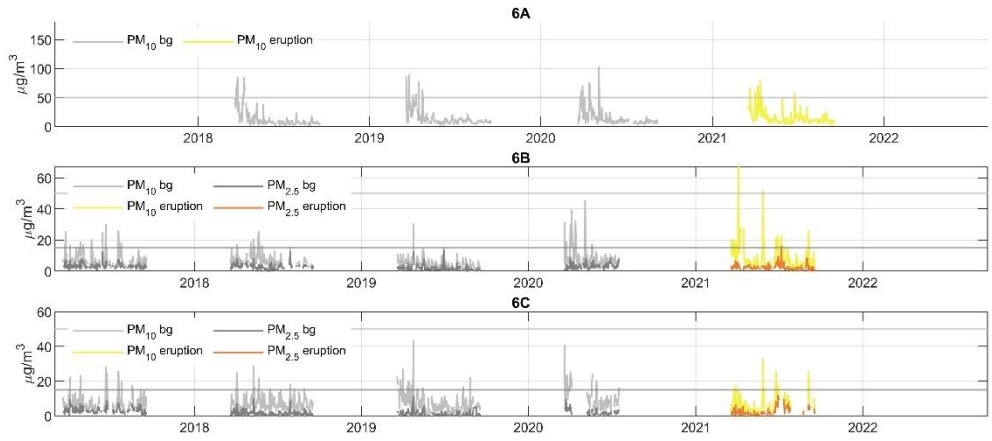

**Figure A12: Time series of daily-mean concentrations of PM$_{10}$, and PM$_{2.5}$ (µg/m$^3$) measured in North Iceland by regulatory air quality stations (G6 A, B, C) during the 2021 eruption and in the non-eruptive background (bg). PM$_1$ was not measured at these stations. The figures only include data for the period 19 March 20:00 – 19 September 00:00 UTC in each year, i.e. the period corresponding to the calendar dates and months of the 2021 eruption. See main text for the justification of this approach. The figures**
**show the ID air quality thresholds for PM$_{10}$ and PM$_{2.5}$ of 50 and 15 µg/m$^3$ daily-mean, respectively as grey horizontal lines.**

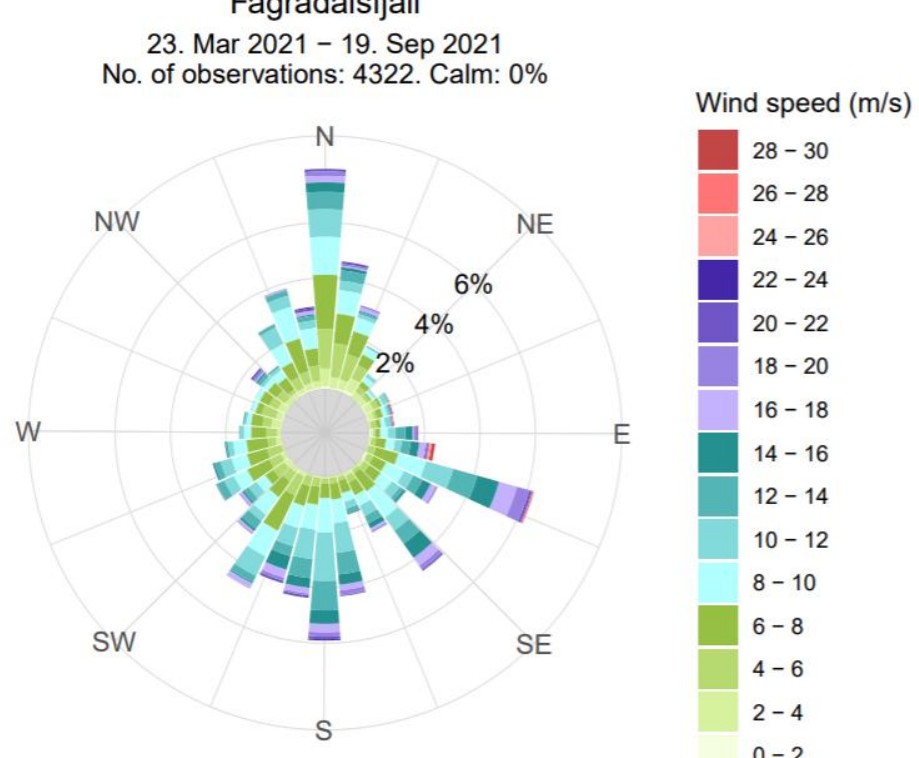

**Figure A13: Wind rose shows wind direction (wind coming from) and wind speed measured by Icelandic Meteorological Office weather station at the Fagradalsfjall eruption site 23 March – 19 September 2021.**

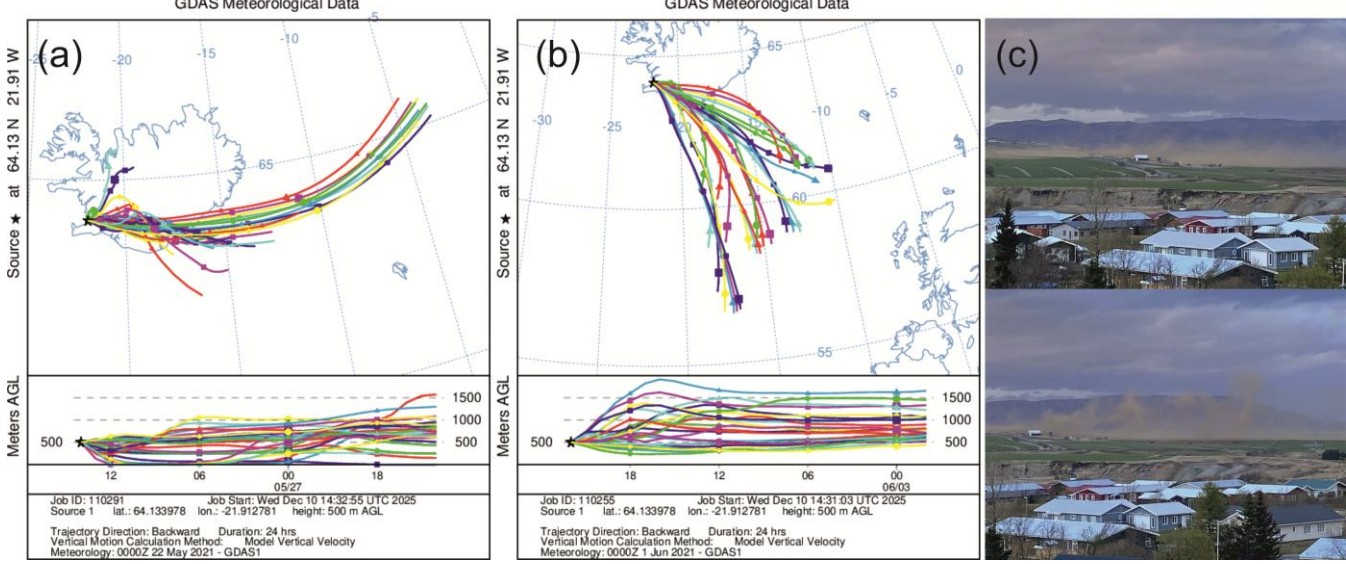

Figure A14: Evidence for two Icelandic highland storms affecting PM levels in Reykjavík capital area 24-29 May and 3-4 June 2021.
Panel (a) Ensemble back-trajectory analysis for the peak PM concentrations in Reykjavík on 27 May at 14:00 UTC/local time, calculated using the NOAA HYSPLIT model (Stein et al., 2015). Panel (b) Ensemble back-trajectory analysis for the peak PM concentrations in Reykjavík on 3 June at 22:00 UTC/local time, calculated using the NOAA HYSPLIT model (Stein et al., 2015). The back-trajectory analysis for both events is consistent with well-known Icelandic dust storm source areas (Dagsson-Waldhauserova et al., 2014). Panel (c) Two crowd-sourced photographs taken on 28 May 2021 near the source area identified by the
back-trajectory analysis, confirming the dust storm origin. No photographs were available for the 3–4 June event. Photo credit: Sigurður H. Magnússon, posted on *Dust Storms in Iceland* Facebook page (28 May 2021).

**Data availability**

Full dataset of measured $SO_2$, $PM_{10}$, $PM_{2.5}$ and $PM_1$ concentrations is openly available for download from the Environment
Agency of Iceland https://loftgaedi.is/en.

**Author contributions**

RCWW performed the original data analysis and drafted the original manuscript and figures. SB, MAP, TR and AS contributed to data interpretation and manuscript drafting. EI finalised the data analysis, the manuscript and produced Figs. 2-10 and Appendix A1-A12 and A14. RHTh produced Figs. 1, 11, and 12. TH, GMG and MAP designed, built and maintained IMO's
measurement and data systems. TJ contributed access to and information on the regulatory AQ network and quality-controlled data. DF and RHTh led on the ArcGIS ArcMap analysis methodology. GGS supplied the data from footpath counters. All coauthors contributed to draft review and editing.

**Competing interests**

The authors declare that they have no conflict of interest.

**Acknowledgements**

The authors would like to thank Kristín Björg Ólafsdóttir from the Icelandic Meteorological Office for analysis of wind data and creation of the wind rose in Fig. A13. Bogi Brynjar Björnsson at the Icelandic Meteorological Office is thanked for the preparation of Supplementary Figure S3. Microsoft Copilot (GPT-4) was used for English language editing, proofreading, and improving sentence clarity and structure; all content edited by Microsoft Copilot (GPT-4) was critically reviewed and approved by the authors.

**Financial support**

RCWW was funded by the Leeds-York Natural Environment Research Council (NERC) Doctoral Training Partnership (DTP) NE/L002574/1, in CASE partnership with the Icelandic Meteorological Office. TJR was funded by the ANR Projet de Recherche Collaborative VOLC-HAL-CLIM (Volcanic Halogens: from Deep Earth to Atmospheric Impacts) ANR-18-CE01-0018, and Labex Orléans Labex VOLTAIRE (VOLatils-Terre Atmosphère Interactions–Ressources et Environnement, ANR-10-LABX-100-0). EI acknowledges NERC Centre for Observation and Modelling of Earthquakes, Volcanoes and Tectonics (COMET+), a partnership between UK Universities and the British Geological Survey; NERC Highlight Topic V-PLUS NE/S00436X/1 and NERC Urgency Grant NE/Z000262/1 "Chemistry of emissions at lava-urban interfaces".

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
