# Peer review of "Fine-scale fluctuations of PM1, PM2.5, PM10 and SO2 concentrations caused by a prolonged volcanic eruption (Fagradalsfjall 2021, Iceland)"

_EGUsphere, 2025_

## Author Response (AR1)

We thank the reviewers and the editors for their time and comments that have significantly improved the manuscript. Below we list our responses to the reviewers' comments. The reviewers' comments are in blue font, and our responses are in normal font.

**Replies to Reviewer RC1**

*RC1: This paper appears to have been written by the corresponding author, and not have been edited by the native English-speaking co-authors. There are several English grammar errors, which are annoying to this reviewers. Reviewers should not be expected to edit the paper. And every author is required to agree to submission before a paper is submitted. Why did they not read it and correct the grammar? Languages like Russian and Chinese do not have articles, and so it is obvious when the paper is missing articles that the author did not learn English well enough to use them correctly.*

We acknowledge the comments provided. As noted by the Executive Editor, the remarks above were inappropriate. In accordance with the editor's guidance, we will not be responding to those specific comments.

We remain committed to constructive scientific dialogue and appreciate the opportunity to address the remaining points raised during the review process.

*RC1: This paper is about health effects of volcanic emissions, looking at $SO_2$ and PM.*

While we appreciate the reviewer's interest in the broader implications of our findings, we would like to respectfully clarify that it is somewhat misleading to characterise the main focus of our manuscript being "about health effects." Our study presents observational data on ground-level concentrations of particulate and gaseous air pollutants associated with volcanic activity. At no point do we present, or claim to present, data on health outcomes.

We acknowledge the importance of understanding health impacts; however, such assessments require clinical or epidemiological methodologies, which are beyond the scope of this study and the remit of ACP.

*RC1: But adverse health effects depend on exposure, which includes not just concentration of the pollutants but on how long a person is exposed. There was no discussion of this in the paper.*

We understand the reviewer's comment to suggest that the manuscript lacks a discussion of how long a person is exposed to volcanic air pollutants. We respectfully disagree with this interpretation.

Our analysis explicitly compares observed concentrations of $SO_2$ and particulate matter ($PM_1$, $PM_{2.5}$, $PM_{10}$) against both international and national air quality thresholds. These thresholds are inherently time-based: for example, the $SO_2$ threshold is based on a 60-minute averaging period, while the PM thresholds are based on 24-hour averages. By definition, these thresholds incorporate both concentration and duration of exposure.

Figures 3, 4, 5, and 6 present pollutant concentrations alongside the relevant exposure thresholds and show the total number of exceedances for each time-based standard. These figures, and their accompanying discussion in the manuscript, directly address the duration and frequency of exposure to above-threshold pollutant levels.

We hope this clarifies that the temporal dimension of exposure is indeed a central part of our analysis.

*RC1: How dangerous are these pollutants?  Is $SO_2$ or PM more dangerous? And I have never heard of $PM_1$ before.  Is it more or less dangerous than $PM_{2.5}$?*

We have restructured the Introduction to improve clarity and provide additional context. Specifically, we have added a new subsection titled 1.1. Volcanic Air Pollutants and Associated Health Impacts, which offers a more detailed overview of the respective health effects of $SO_2$ and particulate matter (PM), including $PM_1$, $PM_{2.5}$, and $PM_{10}$. This new section also addresses the limited but emerging evidence on the combined effects of $SO_2$ and PM in volcanic plumes, and highlights the growing body of research on the health impacts of $PM_1$, particularly in comparison to $PM_{2.5}$. We believe this addition strengthens the manuscript by better contextualizing the relevance of our observational data to public health.

Furthermore, we would like to respectfully point out that, as of the date the reviewer submitted their comments, there were 1,367 articles in ACP and ACPD on $PM_1$. Of these, 56 had been published since the beginning of 2025. This underscores that $PM_1$ is an active and growing area of research that is highly relevant to the scope of ACP.

*RC1: The paper needs a discussion of these issues, and the conclusions need to frame the results in terms of short or long-term health impacts.  Were the lifespans of visitors or Icelandic residents really affected by this eruption?*

We thank the reviewer for their helpful suggestion. In response, we have expanded the discussion of the potential health impacts on the Icelandic population in Section 3.5.1. This addition provides further context for the observed pollutant concentrations and highlights the relevance of our findings for public health considerations in the affected regions. We also added an explicit mention of the potential health impacts to the Conclusions.

**Revised text from section 3.5.1 'Exposure of residents':**

[revised manuscript text omitted]

*RC1: The paper analyzes concentrations of particles, but does not provide any modeling of the particles from the volcanic vents to the sensors. This could be done with various air pollution models, forced by actual meteorology, say from reanalysis, or downscaled with a local high-resolution model. This would explain the variations, and also provide a capability to predict the air pollution from future eruptions. Emissions from a fissure only get to populated places if the wind is blowing that way. Even without sophisticated models, the paper needs to explain the synoptic situation in the most polluted cases to show how the pollution gets to the people. The wind rose in Fig. B11 is not sufficient. Show the data. Provide time loops of satellite images, and show weather maps. Animations are allowed and encouraged as supplementary information.*

We agree that, ideally, our high-quality observational dataset could be complemented by meaningful dispersion modelling. Prior to writing this manuscript, we carefully considered whether to include in-depth dispersion modelling and ultimately decided against it. Below, we

outline the rationale for this decision and the steps we have taken in the revised manuscript to address the reviewer's suggestion as fully as possible. To enhance the manuscript, we have now included a discussion of dispersion modelling results and their limitations in the revised text. Additionally, we have added several maps and animations to the Supplementary Material to provide further context.

**Rationale for not including in-depth dispersion modelling in the main manuscript:**

The dispersion models that have been used for simulating the 2021 Fagradalsfjall eruption do not currently have sufficient skill to meaningfully complement our observational dataset. One of the main reasons for the relatively low level of skill is that eruptions like the 2021 event are small but highly dynamic emission sources. While the models can simulate the broad direction of the plume with approximately 80% accuracy, the fine-scale spatial (≤ a few kilometres) and temporal (≤ 6 hours) fluctuations in plume direction are not accurately reproduced, and the models particularly struggle to simulate the pollutant concentrations at ground level. For example, the most significant volcanic $SO_2$ pollution event in Reykjavík (18–19 July 2021, when ground-level $SO_2$ reached 750 μg/m$^3$) was not reproduced by the model. This limitation is now mentioned in the revised manuscript (the new text is copied below).

The most extensively used model for this eruption is 'CALPUFF', operated by the Icelandic Meteorological Office (IMO). CALPUFF has been used operationally by IMO since 2014 for forecasting volcanic $SO_2$ pollution and alerting the public to the presence of unhealthy concentrations. A recent evaluation of CALPUFF's performance during the 2021 eruption (Pfeffer et al., 2024; https://www.sciencedirect.com/science/article/pii/S0377027324000568#s0010) identified key challenges in simulating this eruption and explored improvements through varying source parameters. Furthermore, a workshop was held at IMO (organised by co-author S. Barsotti) to compare the performance of different dispersion models for this eruption. Models evaluated included CALPUFF (IMO), NAME (UK Met Office), and a WRF-based model (UK National Centre for Atmospheric Science). The consensus was that none of the models were able to simulate the dispersion of pollutants from this eruption with high skill, and that further work is needed to improve model performance. Given these limitations, we concluded that full dispersion modelling analysis is beyond the scope of this paper and warrants a full investigation of its own. Importantly, we believe that publishing the observational dataset first is essential to support future model development and validation.

We appreciate the reviewer's suggestion to include satellite imagery to illustrate the synoptic dispersion of the volcanic plume. While we agree that satellite data can be valuable for understanding the dispersion volcanic plumes, we have carefully assessed its applicability to this particular eruption and concluded that it does not meaningfully complement our high-resolution ground-based observations. Below, we outline the reasons for this decision.

The best satellite instrument for $SO_2$ detection available during the 2021 eruption was the TROPOMI sensor onboard Sentinel-5P, which provides daily global coverage and measures $SO_2$ as column-integrated Dobson Units (DU). The TROPOMI is optimised for detecting large-scale $SO_2$ plumes and has a spatial resolution of 5 × 12 km (with the longer axis oriented north–south), which is insufficient to resolve the fine-scale spatial variability captured by our ground-based network. The data quality is also compromised during periods of low solar angle, which is common in Iceland, particularly outside of summer months. Below is a Sentinel-5p image showing $SO_2$ on the 20 of April 2021. Note the area

over NE Iceland. This is noise in the data, probably due to the low solar angle affecting the data capture.

Another limitation for using satellite data to explain our ground-level observations is that the TROPOMI detects $SO_2$ throughout the atmospheric column and cannot determine whether the plume is present at ground level ('grounding'). In a comparative analysis conducted by the lead author as part of her PhD thesis (Whitty, 2022, University of Leeds), it was found that on days when the satellite detected the volcanic plume, it was grounding in only ~36% of cases.

In summary, while satellite data can provide useful synoptic context for large eruptions, the resolution, coverage, and vertical sensitivity of the available satellite observations during the 2021 Fagradalsfjall eruption are not sufficient to contribute meaningfully to our high-resolution ground-based dataset.

[Figure]

Figure: Sentinel-5p image of Iceland from 20 of April 2021 showing $SO_2$ in Dobson Units (DU).

**Revised text discussing plume dispersion simulations and their limitations:**

In section 3.4 ("Fine-scale temporal and spatial variability in $SO_2$ and $PM_1$ peaks"): "The movement and sharp boundaries of the plume during the 28–30 May episode are shown in an animation in Supplementary Figure S1, based on a dispersion model used operationally for volcanic air quality advisories during the eruption by the IMO (Barsotti, 2020; Pfeffer et al., 2024). The model results are used here for qualitative purposes—as a binary yes/no indicator of potential plume presence at ground level. This is because the model has been shown to have a reasonable skill in predicting the general plume direction but relatively low accuracy in simulating ground-level $SO_2$ concentrations for the 2021 eruption (Pfeffer et al., 2024)."

Also in section 3.4: "Supplementary Figure S2 shows an animation of the simulated dispersion of volcanic $SO_2$ at ground level during the 18–19 July episode as simulated by the IMO model (Pfeffer et al., 2024). As discussed by Pfeffer et al. (2024), the dispersion model did not

accurately simulate all ground-level pollution events, including this one—the largest $SO_2$ pollution episode in Reykjavík during the eruption. This highlights the challenges of accurately simulating ground-level dispersion of volcanic emissions from eruptions like Fagradalsfjall 2021, as well as other small but highly dynamic natural and anthropogenic sources (Barsotti, 2020; Pfeffer et al., 2024; Sokhi et al., 2022). High-resolution observational datasets, including those presented here, can support improvements in dispersion model performance ."

In section 3.5.1: "From a nationwide public health perspective, it was fortunate that volcanic pollutants were predominantly transported to the north and northwest of the eruption site. This atmospheric transport pattern likely mitigated the frequency of $SO_2$ pollution episodes in the densely populated capital area, situated to the northeast of the eruption site. Supplementary Figure S3 illustrates the total probability of above-threshold $SO_2$ concentrations at ground level during the eruption, as simulated by the IMO dispersion model (Pfeffer et al., 2024). As outlined in Section 3.4, these simulations are used here solely to provide a qualitative indication of the broad plume direction at ground level. The modelled dispersion patterns are consistent with observational data, indicating that the plume most frequently grounded to the north and northwest of the eruption site, and more rarely in the capital area (Fig. S3)."

*RC1: And just to be clear, is it correct that the wind direction is shown as the direction the wind is COMING FROM, as is the standard convention? If so, why are there so few times when the wind comes from the west and northwest?*

We thank the reviewer for their observation. We have clarified in the caption of the wind rose figure that the wind direction refers to the direction *from which* the wind is blowing.

Regarding the distribution of wind directions: wind roses represent statistical summaries of wind patterns over a defined period. In this study, we analysed wind statistics over the duration of the eruption. While the jet stream generally flows from west to east, near-surface wind patterns are also influenced by local factors such as topography and proximity to the sea. These local effects can lead to more irregular wind distributions that deviate from the dominant synoptic-scale flow.

*RC1: I recommend major revisions to address these issues. And the authors need to respond to the 39 comments in the attached annotated manuscript.*

We have responded to the comments in the annotated manuscript and included it with this response as an attachment.

*RC1: Why is there an Appendix B and not an Appendix A? Anyway, the Appendix should be supplemental material. Otherwise the paper is too long, and supplemental material should not be needed to read the gist of the paper.*

We thank the reviewer for pointing out the numbering error in the Appendix. It has now been corrected to Appendix A throughout the manuscript. The figures in Appendix are in the appendix and not in the supplementary material based on the guidance from the journal:

**Supplements:** Supplementary material is reserved for items that cannot reasonably be included in the main text or as appendices. These may include short videos, very large images, maps, CIF files, as well as short computer codes such as matlab or python script.

*RC1: There are multiple problems with the figures and tables:*

*The supplemental table needs a caption that can be read. Right now it is buried in Excel spreadsheet cells. The column headings in row 9 are also hard to read. Make them wide enough so the text is easy to read.*

The caption for Table S1 was also in the main manuscript file, on the page after the Conclusions section, in agreement with the journal guidelines. Please note that according to the ACP journal procedures, "supplements will receive a title page added during the publication process including title". We have made the columns in the Excel file wider to make the headings in row 9 easier to read. Please note that other users of this spreadsheet may prefer narrower columns, e.g. for comparing the quantitative values side-by-side. The width of the columns can be adjusted further by the users to suit their individual needs.

*RC1: Table 1 should be all on one page. Why are you making it so hard for the reviewers to read it, with the column headers split over two pages? This is especially true, since there is empty space after the table on that page?*

We thank the reviewer for pointing out this formatting issue. We have amended the tables to make them more readable. In particular, we have made sure that tables and individual words are not split across different lines, and changed shortened words like "erupt." to full words like "eruptions", where possible. The font sized has reduced somewhat because of this but we believe it is still legible. We could not put tables into landscape page orientation by the copy-editing rules of the journal.

*RC1: For Fig. 3, the colors of the eruption and background plots are very similar. Make one red so they can be distinguished. And the diagrams show empty boxes, but I don't see any on the plots. However, the same type plots in Fig. 4 are easy to see. Why don't you make the ones in Fig. 3 the same?*

We have changed the dark blue colour of the background data in Fig. 3 to the same colour as on Fig. 4. This gives a high contrast to the black colour of the eruption data, more so than could be achieved with a red/black colour combination. Regarding the 'diagrams' we assume the reviewer is referring to the box-and-whiskers symbol in the figure legend. The boxes represent the interquartile ranges of the data and are not visible on the $SO_2$ plot because the interquartile range of $SO_2$ has very low values (a few $\mu g/m^3$). This is because the $SO_2$ concentrations in the local background atmosphere are virtually zero, and most of the time the background is 'clear' of volcanic air pollution; but when the volcanic plume is advected into the area, the concentrations become very high. The boxes are more easy to see on Figure 4 because the interquartile range of PM concentrations is relatively high. We have clarified this in the Figure 3 caption:

"The data are presented as box-and-whisker plots: boxes represent the interquartile range (IQR), the whiskers extend to +/–2.7σ from the mean, and crosses represent very high values (statistical outliers beyond +/–2.7σ from the mean). Note that the IQR is very low in most cases due to the negligible $SO_2$ concentrations in the clean local background; as a result, most of the $SO_2$ pollution episodes are statistical outliers". We have also added the crosses to the figure legend.

*RC1: Fig. 4: Why are some stars red and some orange?*

There were no red stars on Figure 4, but there were orange stars that were filled in with solid colour, and orange stars that were not filled in. We have clarified the meaning of each star type in the figure caption:

"Stars with solid orange fill represent the normalised number of times $PM_{10}$ and $PM_{2.5}$ concentrations at each station exceeded the Icelandic Directive (ID) air quality thresholds of 50 $\mu g/m^3$ and 15 $\mu g/m^3$ (24-hour mean), respectively. For $PM_1$, non-filled stars indicate the number of times concentrations during the eruption exceeded the Environmental Agency of Iceland (EAI) threshold of 13 $\mu g/m^3$ (24-hour mean). Different symbols (filled vs. non-filled stars) are used to distinguish between internationally accepted, evidence-based ID thresholds ($PM_{10}$ and $PM_{2.5}$) and the locally applied EAI threshold for $PM_1$, which is not internationally standardized."

*RC1: Table 2 is very poorly done. Because of the large cell margins, Erupt. goes over two lines, and does 0.2 on one line and 5 on the next line mean 0.25? Make the left and right cell margins smaller and use no indentation, and the table will be possible to understand.*

Please see the reply to the comment about Table 1.

*RC1: Shouldn't Table 2 be in the supplemental, and have it replaced with a figure? That would make it much easier to compare the numbers in each cell.*

The data in Table 2 are already visualised in Figures 4, 5 and 6. Table 2 is supporting the figures because the large dynamic range in PM concentrations makes it hard to read the exact values directly from the figures.

*RC1: Table 3 is also faulty. The caption mentions orange and green rows, but the text is all black. And the ratios written in the headers and the years on the side span two rows and are very hard to understand. This needs to be fixed.*

We thank the reviewer for pointing this out. The green and orange colours were in the originally submitted manuscript but were removed prior to review based on the copy-editing rules of the journal. The caption has been amended in the revised manuscript, copied here:

"Table 3. Concentration ratios of PM size fractions (hourly-means, $\mu g/m^3$) associated with different pollution sources in the Reykjavík capital area. Rows 1 and 2 represent periods considered typical of Reykjavík background conditions: the 'Summer period', when studded tyres are not in use (banned between April and November), and a period during the 2021 eruption when the volcanic plume was advected away from Reykjavík. Rows 3–6 show ratios during the 2021 eruption when the plume was advected toward Reykjavík. For definitions of 'fresh' and 'mature' plume, see Section 3.4. Rows 7 and 8, labelled 'Desert dust', correspond to pollution episodes caused by Icelandic highland desert storms (source area ~200 km from Reykjavík), confirmed by meteorological and visual observations from the Icelandic Meteorological Office (IMO). Station G3-G is listed first, as it is considered the most sensitive to the presence of volcanic plume due to its low background concentrations from local sources."

Please also see our reply related to Tables 1 and 2 regarding changes to the readability of the tables.

*RC1: Hint: If a table will not fit in a portrait orientation page, use landscape mode!*

We could not put tables into landscape page orientation due to the copy-editing rules of the journal. Please see the previous reply on how the readability of the tables has been improved.

*RC1: Fig. 9 should plot population density (by area), not by municipality.*

We have amended the figure to show the population density. Please note that it is now Figure 10 in the new manuscript version.

*RC1: Fig. 10a is so tiny with miniscule fonts that it cannot be read. Enlarge it and put it on its own separate page.*

We thank the reviewer for pointing this out. The low resolution was an error when the file was converted to a pdf. We have fixed it and increased the font size on the figure. Please note that it is now Figure 11 in the new manuscript version.

*RC1: Fig. 9 should plot population density (by area), not by municipality.*

We have amended the figure to show the population density. Please note that it is now Figure 10 in the new manuscript version.

*RC1: Fig. 10a is so tiny with miniscule fonts that it cannot be read. Enlarge it and put it on its own separate page.*

**Replies to Reviewer RC2**

*RC2: The article investigates the impact of the 2021 Fagradalsfjall volcanic eruption on air quality in the most densely populated region of Iceland. Using one of the densest reference-grade air quality monitoring networks in the world, the study finds statistically significant increases in $SO_2$ and particulate matter ($PM_1$, $PM_{2.5}$, $PM_{10}$) concentrations, particularly $PM_1$. These findings have important implications for assessing population exposure and public health risks associated with volcanic air pollution in inhabited areas.*

We thank the reviewer for their time and constructive comments.

*RC2: In the reviewers opinion, a major limitation of the study is the lack of calibration for the low-cost sensors.*

We wholeheartedly agree that, ideally, the low-cost sensors (LCS) would have been calibrated against reference-grade instruments. The reasons for the lack of calibration are outlined below and have also been added to the revised manuscript (Section 2.2). However, we respectfully disagree that this constitutes a major limitation of the overall study. The LCS dataset represents a relatively small component of the work. The primary focus of the study is on populated areas, where data were collected using regulatory-grade instruments. The LCS were deployed only at the eruption site to provide short-term guidance to visitors within a limited area (0–5 km from the vent).

**Reasons for the lack of calibration of LCS in this study (also detailed in Section 2.2):**

- **Crisis-response context:** The LCS were installed as part of an emergency response by the volcano observatory. The crisis was twofold: the eruption itself, and the unexpected and unprecedented influx of visitors to the eruption site. The primary purpose of the LCS deployment was to provide real-time alerts for very high $SO_2$ concentrations, not to generate a dataset with precise concentration values.

- **COVID-19 lockdown constraints:** The eruption occurred during national and international COVID-19 lockdowns, which significantly limited the capacity of observatory staff to conduct fieldwork and prevented international collaborators from assisting with sensor deployment and calibration.

While we fully acknowledge that the eruption-site data are not as robust as they could have been with proper calibration, we maintain that they provide valuable context and contribute meaningfully to the overall narrative of the study. Importantly, we use the LCS data only to identify whether $SO_2$ concentrations exceeded or remained below established thresholds. We do not use these data to draw conclusions about absolute concentration values. This limitation is clearly stated in the manuscript (Sections 2.2 and 3.5.2).

*RC2: Below you will find my comments regarding the text.*

*Major Comments:*

*Sensor models: Please provide the specific models of the PM and $SO_2$ sensors used. Were these commercial low-cost sensors (e.g., Alphasense), or custom-built?*

We thank the reviewer for this observation. The specific sensor models used at the eruption-site network (Alphasense $SO_2$-B4 and Crowcon Xgard) were already listed in Table S1, as noted in

the main text, which refers readers to the table for sensor specifications and operational details.

In response to the reviewer's suggestion, we have now added to the *Methodology* section that these sensors are commercially available. We also clarify that only $SO_2$ was monitored at the eruption-site network; particulate matter (PM) was not measured at these locations. We have clarified this in the Methodology Section 2.2:

"At the eruption site (0.6-3 km from the active craters), the IMO installed a network of five commercially available $SO_2$ LCS between April and July 2021 to monitor air quality in the near-field. PM was not monitored with this network due to cost-benefit considerations as PM does not pose as acute a hazard as $SO_2$ for short-term exposure. The sensor specifications and operational durations are detailed in Table S1."

*RC2: Air intake design: How was the air intake for the sensors constructed? Was it a prototype, experimental design, or a standardized market-available solution? This should be clearly described, ideally with a schematic or reference.*

The air intake was designed in-house at the volcano observatory (Iceland Met Office or IMO) taking into account local conditions, in particular the weather and dust resuspension. We have added photographs to the appendix showing the installation in the field, and the cover of the air intake (copied below).

[Figure]

**Figure A1 Lower-cost sensors used for the Fagradalsfjall 2021 eruption. Panel (a) shows the instrument installed in the field. The station was powered by a solar panel (triangular trellis at the back of the photo). The air intake was underneath the instrument (the white box at the front of the image). Panel (b) shows the air intake of the sensor. The air intake was designed in-house at the IMO taking into account local conditions, in particular the weather and dust resuspension. The cover was custom-made from Plexiglass with the sensors are recessed behind it to be protected from dust, precipitation, and other potentially damaging environmental factors.**

*RC2: Sensor sampling period: The manuscript lacks information on the temporal resolution or sampling interval of the sensors. This is crucial for interpreting short-term variability in pollutant concentrations and for reproducibility.*

The temporal resolution for each sensor has been added to Supplementary Table S1

*RC2: It may be helpful to add a dedicated supplementary table with detailed specifications for each sensor (manufacturer, model, detection range, resolution, operational dates).*

This information was already available in Supplementary Table S1, with the exception of temporal resolution; this has been added to the same table.

*RC2: Technical and Formatting Corrections:*

*Line 138 & 360: "PM2.5" → please use correct subscript formatting (e.g., $PM_{2.5}$) and apply this consistently throughout the manuscript.*

We thank the reviewer for pointing this out and have correct these errors throughout.

*RC2: Line 192: Correct the citation format – change "(Icelandic Directive, 2016)" to Icelandic Directive (2016).*

We apologise for this recurring formatting mistake, which was introduced by our referencing software and missed during our proofreading. We have done our best to find and correct all of these mistakes in the revised version.

*RC2: Line 193–194: Add space before parentheses: "15 µg/m3(World" → 15 µg/m$^3$ (World).*

Same reply as to previous comment.

*RC2: Line 119: please clarify which size fraction is being referenced here ($PM_1$, $PM_{2.5}$, or $PM_{10}$?).*

PM10, this has been clarified.

*RC2: Please revise the manuscript to correct improper or missing articles. Numerous instances throughout the manuscript require grammatical revision.*

The revised manuscript has been proofread for English grammar and clarity by the co-authors, with additional support from Microsoft Copilot (GPT-4) as stated in the Acknowledgements.

*RC2: Figures and Tables:*

*Figure 4: Please explain in the legend what is the meaning of full vs empty stars.*

We have rephrased the caption to clarify this:

"Stars with solid orange fill represent the normalised number of times $PM_{10}$ and $PM_{2.5}$ concentrations at each station exceeded the Icelandic Directive (ID) air quality thresholds of 50 µg/m$^3$ and 15 µg/m$^3$ (24-hour mean), respectively. For $PM_1$, non-filled stars indicate the number of times concentrations during the eruption exceeded the Environmental Agency of Iceland (EAI) threshold of 13 µg/m$^3$(24-hour mean). Different symbols (filled vs. non-filled stars) are used to distinguish between internationally accepted, evidence-based ID thresholds ($PM_{10}$ and $PM_{2.5}$) and the locally applied EAI threshold for $PM_1$, which is not internationally standardized."

*RC2: Table 1: The table should be kept on a single page to aid readability.*

We have amended the tables to make them more readable. In particular, we have made sure that tables and individual words are not split across different lines, and changed shortened words like "erupt." to full words like "eruptions", where possible. The font sized has reduced somewhat because of this but we believe it is still legible. We could not put tables into landscape page orientation by the copy-editing rules of the journal.

*RC2: Table 3: The manuscript refers to "Green-coloured" and "Orange-coloured" rows, but these distinctions are not visible in black-and-white print.*

We apologise for the wrong description in the caption – the green and orange colours were in the originally submitted manuscript but were removed prior to review based on the copy-editing rules of the journal. By mistake we did not update the caption accordingly. This has been amended in the revised manuscript:

"Table 3. Concentration ratios of PM size fractions (hourly-means, µg/m$^3$) associated with different pollution sources in the Reykjavík capital area. Rows 1 and 2 represent periods considered typical of Reykjavík background conditions: the 'Summer period', when studded tyres are not in use (banned between April and November), and a period during the 2021 eruption when the volcanic plume was advected away from Reykjavík. Rows 3–6 show ratios during the 2021 eruption when the plume was advected toward Reykjavík. For definitions of 'fresh' and 'mature' plume, see Section 3.4. Rows 7 and 8, labelled 'Desert dust', correspond to pollution episodes caused by Icelandic highland desert storms (source area ~200 km from Reykjavík), confirmed by meteorological and visual observations from the Icelandic Meteorological Office (IMO). Station G3-G is listed first, as it is considered the most sensitive to the presence of volcanic plume due to its low background concentrations from local sources."

*RC2: Please make the tables more readable.*

See previous reply regarding Table 1. The same reply applies to all tables.

*RC2: Consider using a logarithmic scale on selected figures. This would allow better visualization of both low and high values (e.g., Figure B3).*

We have introduced a logarithmic y-axis scale on Figures A3, A4 and A7 (formerly numbered B2, B3 and A6).

**Replies to Reviewer RC3**

*RC3: General comments:*

*This study reports measurements of SO2 and particulate matter concentrations at several Icelandic in-situ measurement stations before and during the 2021 Fagradalsfjall fissure eruption. The study contains some novel results and is of interest to the scientific community, in my opinion. It is generally well written and I don't have any major objections to the publication of the paper. I ask the authors to consider the following general and specific comments.*

We thank the reviewer for their time and constructive comments, which we have addressed as detailed below.

*RC3: One general comment addressed the term „reference-grade". It is used many times throughout the manuscript and I assume most readers will understand more or less what it means. It would be good, however, to state briefly at the beginning of the paper, what the exact meaning of the term is in this study.*

We have added an explanation to the Introduction (also copied below); and we changed the word 'reference-grade' to 'regulatory' throughout the manuscript because we think that a) it is more easily understood by the readers and b) it is consistent with the phrasing used in papers on similar topics, e.g. Crawford et al https://doi.org/10.1073/pnas.202554011.

Copied from the new version of the manuscript:

"Here, we use the term 'regulatory' to describe an air quality monitoring network operated by a national agency, employing certified commercial instrumentation with regulated setup and calibration protocols. These networks provide high-accuracy, high-precision measurements with high temporal resolution, but typically with low spatial resolution due to the high costs of installation (typically > € 100,000) and annual maintenance (typically > € 100,000 per annum)."

*RC3: Specific comments:*

*Line 90: "..., locally known as Reykjanes Fires."*

This is only a minor point, but from the way this is phrased, it is not entirely clear what the last part of the sentence refers to. I assume it is supposed to refer to the eruption, but linguistically it may also refer to "Reykjanes peninsula".

We have rephrased this part of the paragraph and added more explanation on what 'Reykjanes Fires' are, and why they are significant:

"This eruption is considered to mark the onset of a new period of frequent eruptions on the Reykjanes peninsula. Such periods, locally referred to as the 'Reykjanes Fires', have occurred roughly every 1000 years, each lasting for decades to centuries. The last period of Reykjanes Fires ended with an eruption in 1240 CE (Sigurgeirsson and Einarsson, 2019). Since the 2021 eruption, ten further eruptions have occurred on the Reykjanes peninsula: two within the Fagradalsfjall volcanic system (August 2022 and July 2023), and eight within the adjacent Reykjanes-Svartsengi system (December 2023 to April 2025). Volcanic unrest continues at the time of writing, and based on the eruption history of the Reykjanes peninsula, further eruptions may occur repeatedly over the coming decades or centuries."

*RC3: Line 98: "and over 70% of Iceland's total population (263,000 out of 369,000 people) lived within 50 km distance, including the capital area of Reykjavík."*

*This has been mentioned already two times before and I'm not sure, whether it has to be repeated again.*

We have removed the duplication of the number of people in Reykjavik and rephrased this to "This region is the most densely populated area of Iceland, with over 260,000 people—around 70% of the national population—residing within 50 km of the eruption site. The eruption site was 9 km from the town of Grindavík and approximately 25 km from the capital area of Reykjavík. Although the eruption took place in an uninhabited area, it attracted an estimated 300,000 visitors who observed the event at close range."

*RC3: Line 132: "The sensor accuracy limits during field deployment of (Whitty et al., 2022) were significantly poorer than the detection limits reported by the manufacturer."*

*Something is wrong here.*

Thank you for pointing out this phrasing mistake. Rephrased to "The sensor accuracy identified in the field study by Whitty et al. (2022) was significantly poorer than the detection limits reported by the manufacturer."

*RC3: Line 170: "We considered whether the year 2020 had lower PM and PM concentrations"*

*PM10 and PM2.5?*

Yes, we thank the reviewer for pointing out this unfortunate error. Corrected to $PM_{10}$ and $PM_{2.5}$

*RC3: Line 192: "by the (Icelandic Directive, 2016)."*

*Opening parenthesis in the wrong place.*

We apologise for this recurring formatting mistake, which was introduced by the referencing software and missed during the proofreading. We have done our best to find and correct all of these mistakes in the revised version.

*RC3: Lines 193 and 194: missing spaces before citations.*

Corrected, thank you.

*RC3: Figure 2, G3-A: an increase von 14% to 21% is not that large. It would be good to know what the standard deviation of the PM1 fraction before the eruption is.*

We have added text to explain that even a small increase in the $PM_1/PM_{10}$ mass ratio is important for potential health impacts:

"Emerging studies of the links between $PM_1$ and health impacts in urban air pollution have shown that even small increases in the $PM_1$ proportion within $PM_{10}$ can be associated with increasingly worse outcomes; e.g. liver cancer mortalities in China were found to increase for every 1% increase in the proportion of $PM_1$ within $PM_{10}$ (Gan et al., 2025). "

We have added standard deviation to Figure 2, as well as to Table S1. Please note that the % values shown on Figure 2 for the background period have changed slightly from the previous manuscript version because we discovered an error in our data processing script. The error was that the % contribution of the size fractions $PM_{2.5-1}$ and $PM_{10-2.5}$ were calculated based on a much longer timeseries than for $PM_1$, which has only been measured since 2020. We have updated the related discussion accordingly.

*RC3: Line 230: "This is a novel result showing that volcanic plumes contribute a significantly higher proportion of 230 PM1 relative to both PM10 and PM2.5"*

*Without the standard deviation etc. one cannot really tell, whether the PM1 proportion is significantly enhanced, right?*

See the reply to the previous comment, standard deviation has been added to Figure 2 and the related text.

*RC3: Table 1: It would be very useful for the reader to add the standard deviation of the SO2 hourly means for the background conditions, to judge how large the variability is.*

We have added standard deviation to Table 1 (and to Table S1 for individual air quality stations)

*RC3: Figure 3: I don't fully understand what the red asterisks show? The legend states: "The figure also shows whether the number of threshold exceedances at each station exceeded the recommended annual total (n=24, orange horizontal line" and I understand what the horizontal orange line means. But the red asterisks also appear below the line? Sorry, I think, I didn't get the point here.*

We have rephrased the figure caption to clarify this:

"The ID air quality threshold of 350 μg/m$^3$ (hourly-mean) is indicated by a black horizontal line in all panels. Red stars represent the number of times this threshold was exceeded at each station ('exceedance events'). The annual limit for cumulative hourly exceedance events is 24, shown by an orange horizontal line. Stations with red stars above the orange line exceeded the annual threshold."

*RC3: Also: because of the large dynamic range of the SO2 values a logarithmic y-axis would be useful here. At the moment it is impossible to tell, whether the values in Table 1 are consistent with the figures. And the boxes and whiskers are not really visible.*

We understand this comment and we have experimented with a logarithmic scale (figure version shown below); however, we believe that a linear scale is more suitable to display and discuss our data due to the following reasons:

1. The box and whiskers are not visible on the plot because the median value and the interquartile range are very low values (a few μg/m$^3$). This is because the SO$_2$ concentrations in the local background atmosphere are virtually zero, and most of the time the background is 'clear' of volcanic air pollution; but when the volcanic plume is advected into the area, the concentrations become very high. One of the key findings in our study is that the average values of SO$_2$ are not significantly affected by the volcanic eruption, but the SO$_2$ pollution peaks (i.e. the statistical 'outliers') are much higher during the eruption than during the background period. We have clarified this in the Figure 3 caption "Please note that the interquartile range is very low in most cases because SO$_2$ concentrations are virtually negligible in the clean local background; most of the SO$_2$ pollution episodes are therefore statistical outliers."

2. A logarithmic y-axis scale does not make it easier to see the median and the interquartile range as shown on the figure below (because, as explained in the previous paragraph, these values are very small compared to the statistical outliers). The logarithmic scale also has the well-known problem of visually inflating smaller values, meaning that in this case, the difference in the peak SO$_2$ values between the background

and the eruption periods is represented as being relatively small but in reality it was large. Given that the aim with this figure is to illustrate the large difference in the peak values we argue that using a logarithmic scale would be counterproductive for this reason.

[Figure]

Figure: Figure 3 with y-axis logarithmic scale

*RC3: Another question about the box and whisker plots: What percentiles do you assume for the whiskers? It cannot be the 0th and 100th percentiles, because you show outliers.*

*RC3: Another question about the box and whisker plots: What percentiles do you assume for the whiskers? It cannot be the $0^{th}$ and $100^{th}$ percentiles, because you show outliers.*

We have clarified this in the Figure 3 caption (as well as for other figures showing same type of plots):

"The data are presented as box-and-whisker plots: boxes represent the interquartile range (IQR), the whiskers extend to +/–2.7σ from the mean, and crosses represent very high values (statistical outliers beyond +/–2.7σ from the mean). Note that the IQR is very low in most cases due to the negligible $SO_2$ concentrations in the clean local background; as a result, most of the $SO_2$ pollution episodes are statistical outliers."

*RC3: Figure 4: What is the difference between solid and open asterisks? This doesn't seem to be explicitly mentioned?*

We have rephrased the caption to clarify this:

"Stars with solid orange fill represent the normalised number of times $PM_{10}$ and $PM_{2.5}$ concentrations at each station exceeded the Icelandic Directive (ID) air quality thresholds of 50 $\mu g/m^3$ and 15 $\mu g/m^3$ (24-hour mean), respectively. For $PM_1$, non-filled stars indicate the number of times concentrations during the eruption exceeded the Environmental Agency of Iceland (EAI) threshold of 13 $\mu g/m^3$ (24-hour mean). Different symbols (filled vs. non-filled stars) are used to distinguish between internationally accepted, evidence-based ID thresholds ($PM_{10}$ and $PM_{2.5}$) and the locally applied EAI threshold for $PM_1$, which is not internationally standardized."

*RC3: Figure 5: In some of the boxes the median seems to be missing? Why?*

In these cases the median is zero and therefore overlaps with the lower limit of the box. We have clarified this in the figure caption.

*RC3: Table 2: The standard deviation would also be useful here.*

We have added standard deviation to Table 2.

*RC3: Line 411: "Figures 7a-7d show the spatio-temporal resolution and ratios of SO2 and PM"*

*Why "resolution"? Perhaps "variation"?*

Variation is a good suggestion, we have amended the text accordingly. Please note that in the revised version the old Figure 7 is now Figure 8.

*RC3: And the plots don't really show ratios of SO2 and PM. Do you mean the scatter plots here?*

We have rephrased the text to clarify this: "Figure 8 illustrates the variation in $SO_2$ and $PM_1$ abundances during this episode, shown as time series (Figs. 8a–8b) and as concentration ratios (Figs. 8c–8d)." Please note that in the revised version the old Figure 7 is now Figure 8.

*RC3: Figure 10a: The text in figure and legend is barely legible.*

We apologise for the low resolution which was an unfortunate error introduced by the pdf conversion. We have improved the resolution and also increased the font size on the figure. Please note that in the revised version the old Figure 10 is now Figure 11.

*RC3: Section 3.5.2: This section only contains quite general and somewhat vague considerations with little quantitative results? What are the main conclusions of section 3.5.2?*

In the Introduction we have clarified the main objective of analysing the lower-cost sensor (LCS) data:

"We present and discuss the use of LCS in a crisis mitigation context, which has broader relevance for other high-concentration, rapid-onset air pollution events, such as wildfires."

Section 3.5.2 has also been rewritten to make the aims and the conclusions clear. The whole section is too long to be copied into this response (but is available in the revised manuscript file) so we copied just the main conclusion here:

"In conclusion, we suggest that the deployment of the LCS network contributed meaningfully to reducing the $SO_2$ hazard at the eruption site, given the high frequency of above-threshold $SO_2$ concentrations and the high number of people within a small area. Such networks are recommended in comparable crisis-response scenarios, provided that careful consideration is given to how the data and resulting alerts are interpreted and communicated. However, their applicability may be less suitable in contexts where chronic exposure among permanent residents is the primary concern."

*RC3: There is probably (?) no lower threshold for the harmful consequences of air pollution and it may also be of interest to consider periods with SO2 levels below the assumed threshold.*

We agree that lower concentrations of $SO_2$ may still be associated with adverse health impacts, and we acknowledge that this is an important area for future epidemiological research. In response, we have added a discussion of this point to Section 3.5.1 (copied below). Our full dataset includes measurements across a wide range of concentrations, including lower levels of $SO_2$, and is therefore well-suited to support follow-on epidemiological investigations.

"Based on the available evidence, it is likely that the 2021 eruption may have resulted in adverse health impacts among exposed populations. Epidemiological studies by Carlsen et al. (2021a, b) on the 2014–2015 Holuhraun eruption demonstrated a measurable increase in healthcare utilisation for respiratory conditions in the Reykjavík capital area, associated with the presence of the volcanic plume. Exposure to above-threshold $SO_2$ concentrations was linked to approximately 20% increase in asthma medication dispensations and primary care visits. Furthermore, even modest increases in $SO_2$ levels were associated with small but statistically significant rises in healthcare usage—approximately a 1% increase per 10 μg/m$^3$ $SO_2$— suggesting the absence of a safe lower threshold. During the Fagradalsfjall eruption, $SO_2$ concentrations in populated areas reached levels broadly comparable to those observed during the larger but more distal Holuhraun eruption. Consequently, similar health impacts may be expected, as inferred from the findings of Carlsen et al. (2021a, b). Holuhraun emissions led to 33 exceedances of the $SO_2$ air quality threshold in Reykjavík, with hourly-mean concentrations peaking at 1400 μg/m$^3$ (Ilyinskaya et al., 2017). In comparison, the Fagradalsfjall eruption caused 31 exceedances, with a maximum of 2400 μg/m$^3$ $SO_2$ recorded in the community of Vogar (station G2-F). Additionally, Fagradalsfjall caused $SO_2$ threshold exceedances across all monitored areas within approximately 50 km of the eruption site (areas G1–G5). By definition, there is no safe lower limit for the number of air quality exceedance events. Therefore, all areas that recorded above-threshold pollutant concentrations may have experienced adverse health effects. Furthermore, although the monitored regions in North and East Iceland (areas G6 and G7) did not register threshold exceedances, potential health impacts in these areas cannot be ruled out. As reported by Carlsen et al. (2021b), even relatively small, above-background

increases in $SO_2$ concentrations during the Holuhraun eruption were associated with measurable health effects.

Given the limited number and scope of health impact studies on previous volcanic eruptions, the potential health implications discussed here should be further investigated through dedicated epidemiological and/or clinical studies focused specifically on the Fagradalsfjall event. Moreover, existing health studies from volcanic regions have primarily concentrated on short-term exposure (hourly and daily), with a gap in research of potential long-term effects. Since the 2021 eruption, ten additional eruptions of similar style and in the same geographic area have occurred. Although each event has been relatively short-lived—ranging from several days to several months—their cumulative impact on air quality and public health may be chronic rather than acute, and thus warrants comprehensive investigation."

---

## Referee Report (RR1)

Thank you for asking me to review this paper. I was not one of the original reviewers so I have reviewed this paper as I would for any review (blind – I didn't read the reviews and responses first). The paper is well written and important, and should be published but, in my view, although the paper is clearly much improved, there are some additional issues that should be addressed. I have separated these into major and minor issues although I hope all can be easily and quickly addressed.

Major issues

1. Section 2.2 line 192-196 and other places throughout the paper – While I appreciate the effort taken to explain that the LCS data are not reliable, you then use the data to determine if SO2 concentrations are above or below the hourly AQ ID threshold. However, without co-location and calibration, your sensors could have been highly inaccurate. It is only reasonable to give a high/low concentration indication for SO2 concentrations, or to give relative values amongst the different sensors (but even this is challenging given that it doesn't seem like the LCS were first co-located together to check their precision). By comparing to a regulatory ('absolute') concentration, you are undertaking a quantitative assessment which cannot be verified. Stating that the assessment is 'qualitative' by just saying whether the exceedance occurred or not, does not make this an acceptable method. I have no doubt that the concentrations would be detectable (line 195) but this is not the point. In my view this analysis should be removed, throughout the paper. In relation to this, Figures 3 (panel a) and 11 (panel b), and section 3.3 are particularly problematic. Please see comments, below.

Minor issues

2. Abstract – I am uncomfortable with the phrasing of the sentence 'This suggests a possible increase in adverse health effects.' given that no research was conducted to confirm this. Instead, I would suggest: 'This suggests the potential for an increase in adverse health effects.'

3. Section 1.1, 1st sentence 'Much of the existing knowledge on the health impacts of volcanic air pollution comes from epidemiological and public health investigations of the eruptions at Holuhraun in Iceland and Kīlauea in Hawaii.' I do not agree with this sentence. What about all the studies related to ash? Modify the sentence to clarify that you are referring to gases and aerosols.

4. Section 1.1 line 104 – 'Epidemiological studies in volcanic regions further indicate that children (defined as ≤4 years old)' – please add in the word 'young' before 'children' since, clearly, children can be older than 4 years old.

5. Section 1.1. line 131-132 – typo - remove one of the 'been's.

6. Figure 1 – what are the red dots across the main map? Please label them in the caption.

7. Section 2.1 line 171 - 'pulsed fluorescence in the ultraviolet' – isn't this usually termed pulsed ultraviolet fluorescence?

8. Section 2.2 line 178 – 'PM was not monitored with this network due to costbenefit considerations as PM does not pose as acute a hazard as SO2 for short-term exposure.' I don't think I agree with this sentence. PM, especially acid coated PM, may cause acute respiratory issues just like SO2 can do. Maybe replace 'does not' with 'may not' unless you can provide a robust reference (e.g. meta-analysis) to evidence this point. Also, hyphenate cost-benefit.

9. Section 2.2 line 181 and section 3.5.2 lines 705-714 – 'The main purpose of the eruption-response network was to alert visitors when SO2 levels were high'. Please explain how this alerting was done.

10. Section 2.2. line 194 – What is 'ID'? It is actually explained later, in line 258. This needs to be moved to first mention.

11. Section 2.4 line 261 and Section 1.1 line 123 – In Section 1.1, it says 'This study contributes the first regulatory-grade time series and exposure dataset of PM1 from a volcanic source, as well as the first measurements of PM1 in Iceland.' Yet it becomes clear, later, that Iceland has been measuring PM1 since 2020 (line 227-8) and they have a regulatory threshold already in place. Therefore, this isn't the first measurement of PM1 in Iceland. The Introduction should be corrected/clarified.

12. Section 3.1 lines 288-289 – 'The proportion of PM1 mass within PM10 increased from 16-24% in the background (standard deviation 7-13%) to 24-32% during the eruption (standard deviation 16-19%);' Are the values ranges of the raw data or ranges of the means? It would make sense if they were ranges of the means at the different stations. If they are ranges of the raw datasets, it is strange to give standard deviation without the means. Also, I presume this is 1 SD? Figure 2 indicates that the data presented in the main text are likely mean + 1 SD but this needs clarifying.

13. Table 1 (& Table 2). 'The number of AQ exceedances is the maximum number of exceedances recorded by an individual station within a geographic area.' Is this sentence necessary given that the sentence before already explains what 'ID exceedances' means? Or is the 2[nd] sentence referring to columns labelled 'Number of AQ exceedances' that no longer exist that are different from the 'ID exceedances' columns? Looking at Table 2, I now think that this is an issue with the column labelling. In Table 2, there are similar sentences, but both refer to 'AQ exceedances' rather than 'ID/AQ exceedances'. As with Table 1, I don't think that the final sentence is required. Maybe, to cover the point you are making and to avoid confusion, you could combine the sentences as follows (example for Table 2): 'AQ exceedances' denotes the number of times PM concentrations (at any single station within a geographic area) exceeded the following thresholds: PM10 - 50 µg/m3; PM2.5 - 15 µg/m3; PM1 - 13 µg/m3.'

14. Figure 3 (& Figures 4-6). Firstly, it's not at all clear that Figure 3 shows box and whisker plots. I can't see the boxes or 'whiskers' except the crosses for statistical outliers. Figures 4-6 are much clearer, especially as the box is wider than the crosses. Could you present the data on a logarithmic scale to allow visualisation of the boxes? Each panel is also quite small, so it is hard to see the detail.

Secondly, it took me a long time to work out that the stars in Figure 3 relate to the number of exceedances (although it does say this in the caption) and, therefore, the right-hand axis. I think it would really help if the stars were orange rather than red, to match the exceedance line and the right-hand axis. It would also really help if the star, line and axis were in the same shade of orange. Same issue for Figures 4-6 – make the filled star the same shade of orange as the unfilled star and right-hand axis (but at least these ones are orange, not red … or at least that's what it says in the captions … it's actually hard to tell!).

Note my (major) issue with the validity of comparing LCS data to regulatory thresholds, and therefore the inclusion of panel a of Figure 3. If this is to remain in the paper, i) please explain what the error bars refer to, in the caption; and ii) in the main text be absolutely clear that

these values are extremely indicative and should not be taken as definite exceedances. Currently, panel a isn't discussed in the text at all (on p12).

15. Section 3.2 lines 345-361 – one thing that is not discussed is that PM in Iceland can also be influenced by dust mobilising events from the island's interior. Did you check wind conditions, and other evidence, to ensure that none of the increased PM during eruption periods was, coincidentally, from different crustal sources? Or potentially other acute sources (fires/construction etc.)? The way it is written at the moment, it is assumed that there is a causation in the correlation! However, in section 3.3, this is briefly discussed – in general terms – but whether the data collected solely relate to increases due to volcanic pollution is not discussed.

16. In section 3.3, it's not clear if the number of exceedances (line 420) includes those at the eruption site (G1) from LCS. If so, I would remove these values (or discuss them separately). Given that the following sentence says there were 16,000 exceedances at a G1 station, my guess is that the 0-31 value does not include G1 stations (!) but this should be clarified. At this point, there should be a clear discussion to highlight the reliability of the data indicating 16,000 exceedances, if you are going to keep these data in the paper.

17. In general, section 3.3 seems repetitive of earlier sections and those parts that are not could have been incorporated into earlier discussions, and Table 3 would be much better visualised as a figure, with the main text being more explanatory (line 456: 'suggests distinct 'fingerprint' ratios' – so, what are these? It's hard to work this out from the table).

18. Section 3.4 – repetition re. Pfeffer et al. 2024 model limitations in lines 486-491 and 516-521.

19. Figure 9 – there is an issue in the caption. Instead of panels c and d, the caption says panels g and h.

20. Section 3.5.1 line 638 – do you think there is a cumulative impact on air quality from different eruptive events? I do not think so. There is more than sufficient time for the air quality to return to background concentrations between episodes. I would remove this (but keep in the cumulative effect on public health). And I would say that the health effects could be chronic 'as well as' acute (instead of 'rather than').

21. Figure 11 – I strongly disagree with using 'hours above SO2 ID threshold' for the reasons stated in the major comments, above. It would be much better to use the raw data to give indicative air quality concentrations (time series) at each station, allowing qualitative comparison among stations with no reference to air quality thresholds.
Additionally, the graph is very hard to read with bars being used both for daily visitors and SO2 concentrations (for multiple stations – I can only differentiate 2 or 3 because the greyscale is similar and complicated by the overlayed blue bars). Also, to the left of the orange dashed line (note, the O in SO2 label looks subscripted), the bars look both blue and grey. I don't understand this if sensor installation hadn't happened yet. Time series line graphs for SO2 concentrations would improve graph visualisation and interpretation.

22. Section 3.5.2 lines 670-680 – another point is that you don't know if individuals visited multiple times, therefore increasing their exposure.

23. Section 3.5.2 lines 697-714 – this section is weak. You have not conducted any evaluation of the efficacy of the LCS network because neither the precision or accuracy have been measured. Therefore, drawing conclusions of its meaningfulness and utility is overstepping the remit of this paper, especially given that there is also no information on how alerts were

disseminated and whether they were responded to or not. Please remove this section. This aligns with my other concerns about the LCS data reported in this paper.

24. Conclusions line 720 – 'These results suggest that the Fagradalsfjall eruption may have contributed to measurable adverse health effects, warranting further public health investigations.' As already discussed, health effects were not measured, so such wording needs to be carefully chosen. I would suggest something like: 'These results suggest that the Fagradalsfjall eruption generated sufficient air pollution that it could have triggered negative health responses, which should be investigated retrospectively or during future events.'

25. Figure A1 caption – 'The cover was custom-made from Plexiglass with the sensors are recessed…' Remove 'are'.

26. Figure A2 and A5 captions – 'The stations were not in operation before the eruption an therefore' Replace 'an' with 'and'.

27. Figure A4 – panel 3A has multiple horizontal dashed lines. I think this is because the Y axis scale has more tick marks than the other panels despite using the same scale as the other panels. Please remove. Figure A7 has the same issue but at least it is in all the panels.

---

## Author Response (AR2)

**Summary of Revisions**

We thank the reviewers and the editors for their time and comments that have significantly improved the manuscript.

**Major changes include:**

- **LCS Data Handling:**

  - Removed all comparisons of LCS data to regulatory thresholds and reframed interpretation as qualitative ("elevated concentrations") rather than quantitative.

  - Separated LCS data from regulatory-grade data (now in Section 3.4.2 and Figure 12) and added clear disclaimers in text and captions.

  - Clarified sensor co-location details and uncertainty estimation.

- **PM Source Attribution:**

  - Added $SO_2$-based plume filtering method and presented both filtered and unfiltered PM datasets.

  - Incorporated identification and discussion of two summer dust storms (24–29 May and 3–4 June 2021) with supporting HYSPLIT back-trajectory and observations.

  - Expanded discussion on variability in PM ratios and cited relevant Icelandic dust storm studies.

- **Figures and Tables:**

  - Converted Table 3 into Figure 9 for better visualization of PM source "fingerprint" ratios; moved original table to Appendix (Table A1).

  - Updated Figures 3–6 for colour consistency and clarified captions; added explanation for small IQR values.

  - Revised Figure 12 for clarity (visitor data as line graph; LCS data shown as daily max–min range).

- **Textual Revisions:**

  - Corrected introduction claims about $PM_1$ measurements; clarified novelty as first volcanic $PM_1$ time series.

  - Rephrased statements on health impacts to avoid implying measured outcomes.

  - Added explanation of alerting process during eruption and clarified visitor exposure limitations.

  - Combined sections 3.2 and 3.3 to avoid repetition and improve flow.

  - Removed most of Section 3.5.2 (3.4.2 in the revised version) per reviewer request, retaining only a short lessons-learned paragraph.

  - Added missing references and corrected typographical errors.

Below we list our detailed responses to the reviewers' comments. The reviewers' comments are in *blue italics* and our responses are in normal font.

**Replies to Anonymous Referee #4**

*AR#4: Thank you for asking me to review this paper. I was not one of the original reviewers so I have reviewed this paper as I would for any review (blind – I didn't read the reviews and responses first). The paper is well written and important, and should be published but, in my view, although the paper is clearly much improved, there are some additional issues that should be addressed. I have separated these into major and minor issues although I hope all can be easily and quickly addressed..*

We thank the reviewer for their time and thoughtful and constructive comments, which have improved the manuscript.

*Major issues*

*1.Section 2.2 line 192-196 and other places throughout the paper – While I appreciate the effort taken to explain that the LCS data are not reliable, you then use the data to determine if SO2concentrations are above or below the hourly AQ ID threshold. However, without co-location and calibration, your sensors could have been highly inaccurate. It is only reasonable to give ahigh/low concentration indication for SO2 concentrations, or to give relative values amongst the different sensors (but even this is challenging given that it doesn't seem like the LCS were first co-located together to check their precision). By comparing to a regulatory ('absolute')concentration, you are undertaking a quantitative assessment which cannot be verified. Stating that the assessment is 'qualitative' by just saying whether the exceedance occurred or not, does not make this an acceptable method. I have no doubt that the concentrations would be detectable (line 195) but this is not the point. In my view this analysis should be removed, throughout the paper. In relation to this, Figures 3 (panel a) and 11 (panel b), and section 3.3are particularly problematic. Please see comments, below.*

We appreciate the reviewer's detailed comments and have carefully considered their concerns. In the revised manuscript, we have substantially revised the analysis and discussion of the LCS data to further clarify their limitations and ensure that our interpretation is consistent with the capabilities of these sensors. Specifically, we have:

- Removed any comparisons to regulatory thresholds and reframed the discussion to emphasize only qualitative trends and relative differences among sensors.
- In Section 2.2. we clarified that the LCS were co-located in the field during the eruption period and the co-location data were used to estimate the sensor uncertainty, presented as error bars (formerly Figure 3, now Figure 12).
- Removed LCS data from Figure 3 and from Section 3.3. in order to clearly separate its presentation and discussion from regulatory-grade data.
- The figures and sections where LCS are presented have been revised accordingly (Figure 12 and Section 3.4.2 in the revised version.)

We believe that presenting the LCS data remains important because these sensors provided the only near-real-time information available during the 2021 eruption and continue to play a role in the operational hazard assessment for ongoing activity in the region. The revised manuscript

presents these data in an appropriate and transparent way, without implying quantitative accuracy.

*Minor issues*

*2.Abstract – I am uncomfortable with the phrasing of the sentence 'This suggests a possible increase in adverse health effects.' given that no research was conducted to confirm this. Instead, I would suggest: 'This suggests the potential for an increase in adverse health effects.'*

We agree with the suggested rephrasing and have amended the text accordingly.

*3.Section 1.1, 1st sentence 'Much of the existing knowledge on the health impacts of volcanic air pollution comes from epidemiological and public health investigations of the eruptions at Holuhraun in Iceland and Kīlauea in Hawaii.' I do not agree with this sentence. What about all the studies related to ash? Modify the sentence to clarify that you are referring to gases and aerosols.*

Thank you, the text has been amended.

*4.Section 1.1 line 104 – 'Epidemiological studies in volcanic regions further indicate that children (defined as ⩽4 years old)' – please add in the word 'young' before 'children' since, clearly,children can be older than 4 years old.*

Thank you, the text has been amended.

*5.Section 1.1. line 131-132 – typo - remove one of the 'been's.*

Thank you, the text has been amended.

*6.Figure 1 – what are the red dots across the main map? Please label them in the caption.*

Explanation added to the caption: "Red circles on the main map show the location of populated areas, including the capital area Reykjavík which is represented with a comparatively larger circle. The stations were organised in seven geographic clusters (each shown on the enlarged insets)."

*7.Section 2.1 line 171 - 'pulsed fluorescence in the ultraviolet' – isn't this usually termed pulsed ultraviolet fluorescence?*

Thank you, the text has been amended.

*8. Section 2.2 line 178 – 'PM was not monitored with this network due to cost-benefit considerations as PM does not pose as acute a hazard as SO2 for short-term exposure.' I don't think I agree with this sentence. PM, especially acid coated PM, may cause acute respiratory issues just like SO2 can do. Maybe replace 'does not' with 'may not' unless you can provide a robust reference (e.g. meta-analysis) to evidence this point. Also, hyphenate cost-benefit.*

Thank you for this suggestion. We agree that the original phrasing could be misleading. In the revised manuscript, we have simplified the sentence to: *"PM was not monitored with this network due to cost-benefit considerations."*. This reflects the primary reason for excluding PM measurements without making assumptions about its relative health impact.

*9. Section 2.2 line 181 and section 3.5.2 lines 705-714 – 'The main purpose of the eruption-response network was to alert visitors when SO2 levels were high'. Please explain how this alerting was done.*

We have added the following explanations to sections 2.2 and 3.4.2 (formerly 3.5.2):

"The measurements from the sensor network were publicly available in real-time on the EAI air quality monitoring website (airquality.is). The eruption site was staffed by members of the rescue services and/or rangers, who carried handheld $SO_2$ LCS to supplement the installed network. When any of the LCS reported $SO_2$ concentrations as elevated (potentially-above 350 µg/m³) visitors were urged to relocate to areas with cleaner air. During the course of the 2021 eruption and subsequent events (2022–2025), $SO_2$ measurements from the LCS stations were also used by the IMO to produce hazard maps around the active and potential eruption sites, with hazard zones defined by the distances at which elevated $SO_2$ was detected (Icelandic Meteorological office, 2025)".

Reference: Icelandic Meteorological office: https://en.vedur.is/volcanoes/fagradalsfjall-eruption/hazard-map/, last access: 14 December 2025.

*10. Section 2.2. line 194 – What is 'ID'? It is actually explained later, in line 258. This needs to be moved to first mention.*

Thank you for pointing this out. The acronym ID (Icelandic Directive) was originally defined in Section 2.1 (line 168). In the revised manuscript, we have removed the reference to ID from Section 2.2 to align with the reviewer's recommendation that LCS data should not be compared to air quality thresholds. Additionally, we have ensured that the acronym is explained again in Section 2.4 where it is mentioned, to avoid confusion.

*11. Section 2.4 line 261 and Section 1.1 line 123 – In Section 1.1, it says 'This study contributes the first regulatory-grade time series and exposure dataset of PM1 from a volcanic source, as well as the first measurements of PM1 in Iceland.' Yet it becomes clear, later, that Iceland has been measuring PM1 since 2020 (line 227-8) and they have a regulatory threshold already in place. Therefore, this isn't the first measurement of PM1 in Iceland. The Introduction should be corrected/clarified.*

We have rephrased the sentence in section 1.1 to clarify our intended meaning: "This study reports on the first three years of regulatory-grade $PM_1$ measurements in Iceland (2020-2022) and represents the first regulatory-grade time series of $PM_1$ from a volcanic source.". We have also removed a similar statement from Section 2.4 to avoid repetition.

*12. Section 3.1 lines 288-289 – 'The proportion of PM1 mass within PM10 increased from 16-24% in the background (standard deviation 7-13%) to 24-32% during the eruption (standard deviation 16-19%);' Are the values ranges of the raw data or ranges of the means? It would make sense if they were ranges of the means at the different stations. If they are ranges of the raw datasets, it is strange to give standard deviation without the means. Also, I presume this is 1 SD? Figure 2 indicates that the data presented in the main text are likely mean + 1 SD but this needs clarifying.*

We have clarified this as suggested by the reviewer: "The proportion of $PM_1$ mass within $PM_{10}$ increased from the average of 16-24% in the background (one standard deviation ±7-13%) to 24-32% during the eruption (±16-19%) …"

*13. Table 1 (& Table 2). 'The number of AQ exceedances is the maximum number of exceedances recorded by an individual station within a geographic area.' Is this sentence necessary given that the sentence before already explains what 'ID exceedances' means? Or is the 2nd sentence referring to columns labelled 'Number of AQ exceedances' that no longer exist that are different from the 'ID exceedances' columns? Looking at Table 2, I now think that this is*

*an issue with the column labelling. In Table 2, there are similar sentences, but both refer to 'AQ exceedances' rather than 'ID/AQ exceedances'. As with Table 1, I don't think that the final sentence is required. Maybe, to cover the point you are making and to avoid confusion, you could combine the sentences as follows (example for Table 2): 'AQ exceedances' denotes the number of times PM concentrations (at any single station within a geographic area) exceeded the following thresholds: PM10 - 50 µg/m3; PM2.5 - 15 µg/m3; PM1 - 13 µg/m3.'*

We thank the reviewer for a good suggestion for how to combine the two sentences and clarify the  meaning:

Revised caption Table 1: $SO_2$ concentrations (hourly-mean, $µg/m^3$) in populated areas around Iceland during both the non-eruptive background and the Fagradalsfjall 2021 eruption. 'Average' is the long-term mean of all stations within a geographic area ± 1σ standard deviation. 'Peak' is the maximum hourly-mean recorded by an individual station within the geographic area. 'ID exceedances' denotes the maximum number of times $SO_2$ concentrations (at any single station within a geographic area) exceeded the Icelandic Directive (ID) air quality threshold of 350 $µg/m^3$.

Revised caption Table 2: $PM_{10}$, $PM_{2.5}$ and $PM_1$ concentrations ($µg/m^3$, 24-h mean) in populated areas around Iceland during both the non-eruptive background ('BG'), the whole eruption period ('Eruption'), and on 'plume present' days only (see Methods for the definition of plume-present days). 'Average' refers to the long-term mean of 24-hour values of all stations within a geographic area ± 1σ standard deviation.  'Peak' is the maximum 24 h-mean recorded by an individual station within the geographic area. 'AQ exceedances' denotes the maximum number of times PM concentrations (at any single station within a geographic area) exceeded the following thresholds: $PM_{10}$ - 50 $µg/m^3$; $PM_{2.5}$ - 15 $µg/m^3$; $PM_1$ - 13 $µg/m^3$.

*14. Figure 3 (& Figures 4-6). Firstly, it's not at all clear that Figure 3 shows box and whisker plots. I can't see the boxes or 'whiskers' except the crosses for statistical outliers. Figures 4-6 are much clearer, especially as the box is wider than the crosses. Could you present the data on a logarithmic scale to allow visualisation of the boxes? Each panel is also quite small, so it is hard to see the detail.*

We understand this comment and we have experimented with a logarithmic scale as shown below on Figure A; however, we believe that a linear scale is more suitable to display and discuss our data due to the following reasons:

1. The box and whiskers are not visible on Figure 3 because the median value and the interquartile range are very low values (a few $µg/m^3$). This is because the $SO_2$ concentrations in the local background atmosphere are virtually zero, and most of the time the background is 'clear' of volcanic air pollution; but when the volcanic plume is advected into the area, the concentrations become very high. One of the key findings in our study is that the average values of $SO_2$ are not significantly affected by the volcanic eruption, but the $SO_2$ pollution peaks (i.e. the statistical 'outliers') are much higher during the eruption than during the background period. We have clarified this in the Figure 3 caption "*Note that the IQR is very low in most cases due to the negligible $SO_2$ concentrations in the local background; as a result, most of the $SO_2$ pollution episodes are statistical outliers*". On Figures 4-6, the range of PM concentrations in the background are non-negligible, resulting in a visually clearer statistical distribution.

2. A logarithmic y-axis scale does not make it easier to see the median and the interquartile range as shown on Figure A below (because, as explained in the previous paragraph, these values are very small compared to the statistical outliers). The logarithmic scale also has the well-known problem of visually inflating smaller values, meaning that in this case, the difference in the peak $SO_2$ values between the background and the eruption periods is represented as being relatively small but in reality it was large. Given that the aim with this figure is to illustrate the large difference in the peak values we argue that using a logarithmic scale would be counterproductive for this reason.

[Figure]

Figure A: Figure 3 with y-axis logarithmic scale

*Secondly, it took me a long time to work out that the stars in Figure 3 relate to the number of exceedances (although it does say this in the caption) and, therefore, the right-hand axis. I think it would really help if the stars were orange rather than red, to match the exceedance line and the right-hand axis. It would also really help if the star, line and axis were in the same shade of orange. Same issue for Figures 4-6 – make the filled star the same shade of orange as the unfilled star and right-hand axis (but at least these ones are orange, not red ... or at least that's what it says in the captions ... it's actually hard to tell!).*

We thank the reviewer for pointing out the colour inconsistency in the figure and have fixed it on all relevant figures (Figures 3-6).

*Note my (major) issue with the validity of comparing LCS data to regulatory thresholds, and therefore the inclusion of panel a of Figure 3. If this is to remain in the paper, i) please explain what the error bars refer to, in the caption; and ii) in the main text be absolutely clear that these values are extremely indicative and should not be taken as definite exceedances. Currently, panel a isn't discussed in the text at all (on p12).*

We have made the following changes to the manuscript to address the reviewer's comments and suggestions:

i) The error bars on panel A of Figure 3 in the original manuscript represented the sensor uncertainty based data from sensor co-location in the field. We have clarified in section 2.2. what the error bars on LCS data represent (now presented on Figure 12 in the revised manuscript):

Section 2.2: "The absence of a regulatory-grade field calibration significantly limits the accuracy of LCS dataset, particularly at lower concentration levels. To partially mitigate this, two LCS units were co-located at station G1-B between 6 and 22 June 2021 to quantify inter-sensor uncertainty. The co-located sensors were of two types used in this study: Crowcon XGuard (deployed at G1-A throughout the monitoring period and at G1-B until 22 June) and Alphasense $SO_2$-B4 (deployed at G1-B from 22 June and at G1-C, D, and E for the entire period). The measured concentrations showed a strong linear correlation ($r^2$ = 0.70), but Alphasense reported lower values relative to Crowcon, with a correlation coefficient of 0.38 (Fig. A2). This coefficient was used to estimate the measurement uncertainty for the two sensor types, represented here as error bars on relevant figures. While the colocation experiment was useful for identifying uncertainty between sensor brands, it did not quantify variability among sensors of the same brand.

[Figure]

Figure A2: SO₂ concentrations measured by two types of lower-cost sensors (LCS) used in this study—Alphasense SO₂-B4 and Crowcon XGuard—during a field colocation at the eruption site (6–22 June 2021). Measurements from the two sensors showed a strong linear correlation ($r^2 = 0.70$), but Alphasense reported lower values relative to Crowcon, with a correlation coefficient of 0.38. Panel (a) Correlation of raw data points from the two sensors. Panel (b) Correlation after Crowcon data were adjusted using the correlation coefficient

ii) We have made the following changes to ensure clarity:
   a. **Separated the datasets visually** by presenting LCS data in a new figure (Figure 12), distinct from the regulatory-grade measurements, to avoid any implication of equivalence. The revised Figure 3 includes only the regulatory-grade measurements.
   b. **Removed references to air quality thresholds** when discussing LCS concentrations in Sections 2.2 and 3.4.2 (previously 3.5.2). We now refer only to "elevated concentrations" rather than exceedances.
   c. **Revised text in Section 2.2** to explicitly state the indicative nature of the LCS data:
   "Given the calibration and co-location limitations, we do not report quantitative SO₂ concentrations from the LCS network. Instead, the data are presented as a qualitative indicator of whether concentrations were likely elevated—defined as exceeding 350 µg m⁻³ hourly mean—within the uncertainty of the sensors. This threshold is approximately two orders of magnitude above the manufacturer-reported detection limit, making it reasonable to assume that such levels were detectable. However, these values should be interpreted only as indicative; 'elevated levels' do not represent confirmed air quality exceedances."
   d. **Added a clear statement in the caption of Figure 12**:
   "The LCS data should be interpreted only as indicative; 'elevated SO₂' levels do not represent confirmed air quality exceedances."

*15. Section 3.2 lines 345-361 – one thing that is not discussed is that PM in Iceland can also be influenced by dust mobilising events from the island's interior. Did you check wind conditions, and other evidence, to ensure that none of the increased PM during eruption periods was,*

*coincidentally, from different crustal sources? Or potentially other acute sources (fires/construction etc.)? The way it is written at the moment, it is assumed that there is a causation in the correlation! However, in section 3.3, this is briefly discussed – in general terms – but whether the data collected solely relate to increases due to volcanic pollution is not discussed.*

To address the reviewer's concern about non-volcanic PM sources, we revised the manuscript as follows:

- **Section 2.3 (Data Processing):** Added a detailed explanation of how volcanic PM was distinguished from other sources using $SO_2$-based plume identification, while acknowledging limitations (e.g., dust storms, smelter emissions, mixed-source days). Both filtered (plume-present) and unfiltered PM datasets are now presented for transparency.

  Section 2.3: "The importance of non-volcanic sources of PM in Iceland meant that PM concentrations during the eruption period may have been elevated independently of volcanic activity. To identify the volcanic contribution to PM levels, we processed the data following a similar approach to Ilyinskaya et al. (2017). PM data were filtered to include only periods when $SO_2$ concentrations exceeded the non-eruptive background average; these periods are hereafter referred to as 'plume-present days'. Stations G3-G and G3-H did not monitor $SO_2$ and were filtered using $SO_2$ data from stations located within 2 km distance (G3-A and G3-E, respectively). This plume-identification approach has inherent strengths and limitations. First, it is effective at sites with negligible non-volcanic $SO_2$ sources, which applies to most of the monitored locations in Iceland; however, its reliability decreases near aluminium smelters, which represented a minor yet locally important $SO_2$ source at stations G5-all, G6-C, and G7-all. Second, it may exclude periods when the volcanic plume was present with low $SO_2$ but elevated PM, as can occur when the plume is chemically mature (Ilyinskaya et al., 2017). Third, it cannot distinguish between days when PM is predominantly sourced from an eruption and days when volcanic PM is strongly mixed with another PM source, such as dust storms. To address these uncertainties, we present both filtered and unfiltered PM datasets and compare them in our discussion"

- **Section 3.2:** Included analysis showing that some of the major $PM_{10}$ and $PM_{2.5}$ peaks during the eruption period were linked to dust storms, confirmed via HYSPLIT back-trajectory and crowd-sourced observations (Figure A14). We have discussed confidence levels for plume-day identification and highlighted that $PM_1$ is strongly correlated with $SO_2$, suggesting that the volcanic eruption was the most important source.

- **Figures and Tables:** Updated Figures 2, 4–6 and Table 2 to show plume-present days, background, and whole-eruption data for comparison.

*16. In section 3.3, it's not clear if the number of exceedances (line 420) includes those at the eruption site (G1) from LCS. If so, I would remove these values (or discuss them separately). Given that the following sentence says there were 16,000 exceedances at a G1 station, my guess is that the 0-31 value does not include G1 stations (!) but this should be clarified. At this point, there should be a clear discussion to highlight the reliability of the data indicating 16,000 exceedances, if you are going to keep these data in the paper.*

We have clarified the separation between datasets to address the reviewer's concern. In the revised manuscript:

- **Regulatory-grade data** are discussed exclusively in Sections 3.1–3.3.

- **LCS data** are discussed separately in Section 3.4.2.

*17. In general, section 3.3 seems repetitive of earlier sections and those parts that are not could have been incorporated into earlier discussions, and Table 3 would be much better visualised as a figure, with the main text being more explanatory (line 456: 'suggests distinct 'fingerprint' ratios' – so, what are these? It's hard to work this out from the table).*

To address the reviewer's suggestion, we have:

- **Merged Sections 3.2 and 3.3** (now section 3.2) to reduce repetition and improve flow.

- **Converted Table 3 into a figure (Figure 9)** for clearer visualization of the 'fingerprint' ratios.

- **Stated the fingerprint ratios in the text of section 3.2:** "This comparison suggests distinct 'fingerprint' ratios for the different PM sources: volcanic plume periods show the highest $PM_1/PM_{10}$ ratios (mean range 0.3–0.9), dust storms the lowest (mean range during storm peaks 0.04–0.05, mean range during the whole storm 0.1–0.3), and background conditions intermediate (mean ~0.2)."

- **Moved the original Table 3 to the Appendix (Table A1)** to support Figure 9 with detailed timings of the visualised pollution events, and station-specific mean pollutant concentrations.

*18. Section 3.4 – repetition re. Pfeffer et al. 2024 model limitations in lines 486-491 and 516-521.*

We have rephrased the relevant parts of this section to avoid repetition and make the discussion of the model limitations clearer:

"Supplementary Figures S1 and S2 show animations of the simulated dispersion of volcanic $SO_2$ at ground level during the two pollution episodes discussed in this section, 28-30 May and 18-19 July 2021. The simulations were produced by a dispersion model used operationally for volcanic air quality advisories during the eruption by the Icelandic Meteorological Office (IMO) (Barsotti, 2020; Pfeffer et al., 2024). As discussed by Pfeffer et al. (2024), the model had a reasonable skill in predicting the general plume direction but relatively low accuracy in simulating ground-level $SO_2$ concentrations for the 2021 eruption (Pfeffer et al., 2024).The model results are included here for qualitative purposes— as a binary yes/no indicator of potential plume presence at ground level. The sharp ground-level movement and boundaries of the plume during the 28–30 May episode were captured reasonably well by the model (Supplementary Figure S1), but the larger episode on 18-19 July was not reproduced by the model. This highlights the challenges of accurately simulating ground-level dispersion of volcanic emissions from eruptions like Fagradalsfjall 2021, as well as other small but highly dynamic natural and anthropogenic sources (Barsotti, 2020; Pfeffer et al., 2024; Sokhi et al., 2022). High-resolution observational datasets, including those presented here, can support improvements in dispersion model performance."

*19. Figure 9 – there is an issue in the caption. Instead of panels c and d, the caption says panels g and h.*

Thank you for pointing this out, corrected to c) and d). Please note that this figure is now Figure 8.

*20. Section 3.5.1 line 638 – do you think there is a cumulative impact on air quality from different eruptive events? I do not think so. There is more than sufficient time for the air quality to return to background concentrations between episodes. I would remove this (but keep in the cumulative effect on public health). And I would say that the health effects could be chronic 'as well as' acute (instead of 'rather than').*

We agree and have amended the text accordingly: "Although each event has been relatively short-lived—ranging from several days to several months—their cumulative impact on public health may be chronic as well as acute, and thus warrants comprehensive investigation.".

*21. Figure 11 – I strongly disagree with using 'hours above SO2 ID threshold' for the reasons stated in the major comments, above. It would be much better to use the raw data to give indicative air quality concentrations (time series) at each station, allowing qualitative comparison among stations with no reference to air quality thresholds.*

*Additionally, the graph is very hard to read with bars being used both for daily visitors and SO2 concentrations (for multiple stations – I can only differentiate 2 or 3 because the greyscale is similar and complicated by the overlayed blue bars). Also, to the left of the orange dashed line (note, the O in SO2 label looks subscripted), the bars look both blue and grey. I don't understand this if sensor installation hadn't happened yet. Time series line graphs for SO2 concentrations would improve graph visualisation and interpretation.*

Please see our response to the reviewer's major concerns above. In the revised manuscript, we have removed all comparisons of LCS data to regulatory thresholds. We maintain that reporting actual $SO_2$ concentrations from the LCS network would be misleading given the high sensor uncertainty. Instead, we present the LCS data in the most appropriate format—as a qualitative indicator of whether the $SO_2$ concentrations were likely elevated.

Revised section 2.2: "Given the calibration and co-location limitations, we do not report quantitative $SO_2$ concentrations from the LCS network. Instead, the data are presented as a qualitative indicator of whether concentrations were likely elevated—defined as exceeding 350 µg m$^{-3}$ hourly mean—within the uncertainty of the sensors. This threshold is approximately two orders of magnitude above the manufacturer-reported detection limit, making it reasonable to assume that such levels were detectable. However, these values should be interpreted only as indicative; 'elevated levels' do not represent confirmed air quality exceedances". Please also see the revised caption of Figure 12 (formerly Figure 11): "The LCS data should be interpreted only as indicative; 'elevated SO2' levels do not represent confirmed air quality exceedances."

We have revised Figure 12 (formerly Figure 11) to improve clarity and address the reviewer's comments:

- Visitor data is now shown as a line graph rather than bars.

- The LCS data from all five stations are presented as a single bar type, representing the daily max–min range across all five stations, rather than multiple overlapping bars.

- This approach removes the confusing greyscale scheme and ensures that the figure is easier to interpret.

- We also corrected the $SO_2$ label formatting and ensured that no bars appear before sensor installation.

*22. Section 3.5.2 lines 670-680 – another point is that you don't know if individuals visited multiple times, therefore increasing their exposure.*

We have added a sentence on this "In addition, there is no data on whether people visited the eruption multiple times and were therefore potentially cumulatively more exposed."

*23. Section 3.5.2 lines 697-714 – this section is weak. You have not conducted any evaluation of the efficacy of the LCS network because neither the precision or accuracy have been measured. Therefore, drawing conclusions of its meaningfulness and utility is overstepping the remit of this paper, especially given that there is also no information on how alerts were disseminated and whether they were responded to or not. Please remove this section. This aligns with my other concerns about the LCS data reported in this paper.*

We have removed most of Section 3.5.2 (3.4.2 in the revised version) in accordance with the reviewer's recommendation. However, we retained a short reflective paragraph because it is intended as a lessons-learned statement and a call for action in future emergency monitoring campaigns, rather than an evaluation of the LCS network's performance. The retained text reads:

"In conclusion, the deployment of the LCS network at the eruption site for the purposes of alerting people to potentially-high $SO_2$ concentrations was likely valuable given the high frequency of elevated $SO_2$ concentrations and the large number of visitors in a confined area. However, the absence of regulatory-grade calibration prevented any quantitative assessment of individual exposure to hazardous pollutants. To obtain high-quality datasets with LCS, regular and frequent field calibration against regulatory instruments is essential. However, such calibration is typically feasible only during short-term campaigns at reasonably accessible locations. In this crisis-response scenario, the challenging terrain and limited accessibility of the eruption site precluded field calibration. The primary concerns associated with uncalibrated LCS in emergency contexts are false negatives—where the sensor underreports concentrations that exceed health thresholds—and false positives—where the sensor overreports concentrations that are actually below threshold. False negatives pose a problem by failing to alert individuals to hazardous conditions, while repeated false positives may undermine public trust and reduce compliance with safety advisories. "

*24. Conclusions line 720 – 'These results suggest that the Fagradalsfjall eruption may have contributed to measurable adverse health effects, warranting further public health investigations.' As already discussed, health effects were not measured, so such wording needs to be carefully chosen. I would suggest something like: 'These results suggest that the Fagradalsfjall eruption generated sufficient air pollution that it could have triggered negative health responses, which should be investigated retrospectively or during future events.'*

We agree and have amended the text according to the reviewer's suggestion.

*25. Figure A1 caption – 'The cover was custom-made from Plexiglass with the sensors are recessed...' Remove 'are'.*

Amended.

*26. Figure A2 and A5 captions – 'The stations were not in operation before the eruption an therefore' Replace 'an' with 'and'.*

Amended – thank you for the careful proofreading.

*27. Figure A4 – panel 3A has multiple horizontal dashed lines. I think this is because the Y axis scale has more tick marks than the other panels despite using the same scale as the other panels. Please remove. Figure A7 has the same issue but at least it is in all the panels.*

Figure A4 (A5 in the revised version) has been amended. We did not revise Figure A7 (now A8) as the 'extra' grid lines make it easier to read the values on the log-scaled y-axis in this particular case.

**Replies to Referee #5: Pavla Dagsson Waldhauserova**

*Comments to the Editor and Authors*
*This study presents a valuable investigation into the impacts of recent volcanic eruptions on the Reykjanes Peninsula, Iceland, on air quality and the long-range transport of volcanic emissions across the country, particularly to densely populated areas such as Reykjavík. It is an important contribution that combines real-time measurements with data on resident exposure and visitor numbers at the eruption sites. The study also provides evidence that fissure eruptions are one of, or potentially the most, important sources of $PM_1$ in Iceland. However, more comparisons with dust storms and biomass burning events are needed.*
*The paper is clearly written, and the figures effectively support the analyses. I highly recommend this work for publication in EGUsphere, pending minor revisions as outlined below.*

We would like to thank Pavla for the review, which has greatly improved the manuscript.

*General comments:*
*Fissure eruptions are emphasized here as an important source of $PM_1$ in Iceland. Could you provide a more detailed description of all the potential $PM_1$ sources associated with an effusive eruption? You mention ash and sulphate particles directly emitted from the eruption, but it is clear that burning mosses around the eruption sites significantly increased $PM_x$ concentrations, for example, in August 2023, with a clear signature of black carbon.*
We have added text to the Introduction and Conclusions to explain that eruptions can trigger wildfires, but also to clarify that this was not the case in the 2021 eruption and is therefore outside of the scope of this particular study.

In section 1.1 "Some eruptions (e.g. at Kīlauea, Cumbre Vieja, and several recent Reykjanes episodes) cause extensive lava-ignited wildfires, which are also a source of PM.".

In section 1.2 "The 2021 eruption did not trigger significant wildfires; however, several subsequent episodes have caused extensive fires (primarily of vegetation but also some urban structures), warranting a dedicated investigation into their effects on air quality and related health outcomes.".

In Conclusions: "the high frequency of eruptions, and eruption-ignited wildfires in this region since 2021 raises the possibility of chronic exposure, which should also be examined, particularly given that the ongoing Reykjanes Fires'eruptions may continue for several generations."

*It might also be worthwhile to include long-range transport of biomass smoke from Canadian and U.S. fires, which have significantly increased $PM_1$ levels in Iceland, in Table 3. Table 3 is a particularly strong part of the study.*

We thank the reviewer for recognising the importance of Table 3 and for suggesting the inclusion of North American wildfires. However, we argue this falls outside the scope of the present study, which is focused on PM sources that are very frequent and local to Iceland. Our group is preparing a separate paper dedicated to wildfire air quality impacts, including lava-ignited wildfires from the Reykjanes eruptions. As both this manuscript and the forthcoming paper are led by PhD students, merging these topics would compromise the independence and integrity of their respective research efforts. We appreciate the reviewer's and editor's understanding.

*Long-range transport of SO$_2$–PM$_x$ plumes outside Iceland is not addressed. Existing studies have documented such transport to Ireland (e.g., Ovadnevaite et al., 2009). Moreover, emissions from the Reykjanes Fires have been detected in Svalbard and elsewhere, but this paper does not include information on that.*

*Ovadnevaite J., Ceburnis D., Plauskaite-Sukiene K., Modini R., Dupuy R., Rimselyte I., Ramonet R., Kvietkus K., Ristovski Z., Berresheim H., O'Dowd C.D., 2009. Volcanic sulphate and arctic dust plumes over the North Atlantic Ocean. Atmospheric Environment 43, 4968-4974.*

We have added the following discussion to section 3.2:

"Historically, larger Icelandic fissure eruptions (>1 km$^3$ of erupted magma) have caused volcanic air pollution episodes far beyond Iceland—across mainland Europe during the 2014–2015 Holuhraun eruption (Schmidt et al., 2015; Twigg et al., 2016) and potentially even farther during the 1783–1784 Laki eruption (Grattan, 1998; Trigo et al., 2009). Simulations indicate that associated health impacts in Europe could have been substantial (Heaviside et al., 2021; Schmidt et al., 2011; Sonnek et al., 2017). During the recent Reykjanes eruptions (2021–2025), elevated volcanic SO$_2$ was detected at ground level by UK regulatory-grade stations on at least one occasion, in May 2024, exceeding previously documented levels at this distance (UKCEH, 2024). This suggests that PM concentrations may also have been elevated beyond Iceland during these events. Assessing the impacts of recent eruptions on air quality and public health in European and potentially more distant communities is therefore an important priority for future research."

We did not include Ovadnevaite et al 2009 in this discussion because it does not report on volcanic plume transport during an eruption, but rather on diffuse emissions in Iceland during a period of no eruptions. The dynamics of plume transport will likely be very different and not directly comparable. Instead, we have referenced studies which focus on emissions during Icelandic fissure eruptions and report direct observations from regulatory-grade stations in Europe: Schmidt et al., 2015; and Twigg et al., 2016.

Regarding the potential detection in Svalbard: We were unable to confirm which publication the reviewer was referring to, as no further details were provided. The study that most closely matches this description appears to be:

Wu, K., Luo, Y., Li, Q., Zhou, H., Xi, L., and Si, F. (2025). Arctic haze induced by an Icelandic volcanic eruption: Evidence from China's highest-resolution trace gas monitoring. The Innovation Geoscience, 3(2), p.100131.

This is a potentially interesting report based on satellite observations of the 2024 eruption; however, we had concerns about its scientific authenticity that precluded citing it. Specifically, Figure 1 appears to include an unrealistic, AI-generated image (although use of GenAI is unreferenced), and the statement that *"On August 23rd and 24th, the SO$_2$-rich air mass spread to the south and the east under the influence of prevailing winds, affecting Scotland and Ireland for a time"* seems unsupported by data or references.

*The reference list is incomplete, as several cited works are missing. The discussion should include more references to in-situ studies conducted in Iceland that are currently not cited.*

*Specific comments:*
*L121 – Tomášková et al. (2024) is missing from the reference list.*

Thank you for noticing this, this has been amended.

*L446 – Dust from local Icelandic deserts does not need to be "resuspended," similarly to desert dust elsewhere. These are primary dust sources, often originating from beneath glaciers, meaning the material cannot be considered resuspended.*

Amended to: "The dominant non-volcanic PM source in Iceland is natural dust from highland deserts, with dust storms occurring frequently throughout the year with significant regional and seasonal variability (Butwin et al., 2019; Dagsson-Waldhauserova et al., 2014; Nakashima and Dagsson-Waldhauserová, 2019)."

*L447 – "..elevated levels typically occurring during the drier summer months." Other studies show different long-term patterns in the frequency of dust storms in Iceland. In the southern half of Iceland, the highest frequency of dust storms occurs in late winter and spring, with the lowest in summer. Winter dust storms occur frequently in Iceland. Please consider the following works:*

*Nakashima, M. and Dagsson-Waldhauserová, P., 2019. A 60 Year Examination of Dust Day Activity and Its Contributing Factors From Ten Icelandic Weather Stations From 1950 to 2009. Frontiers in Earth Science 6, 245-252. DOI:10.3389/feart.2018.00245*

*Dagsson-Waldhauserova, P., Arnalds, O., Olafsson, H., 2014. Long-term variability of dust events in Iceland. Atmospheric Chemistry and Physics 14, 13411-13422. DOI:10.5194/acp-14-13411-2014.*

We have rephrased this section to state the importance of dust storms as a year-round PM source, with significant seasonal and regional variability: "The dominant non-volcanic PM source in Iceland is natural dust from highland deserts, with dust storms occurring frequently throughout the year with significant regional and seasonal variability (Butwin et al., 2019; Dagsson-Waldhauserova et al., 2014; Nakashima and Dagsson-Waldhauserová, 2019).

*L 452 – There are already an existing study dealing with PM1 proportions during Icelandic dust storms and the reported values differ substantially from yours. See here:*
*Dagsson-Waldhauserova P, Magnusdottir AÖ, Olafsson H, Arnalds O 2016. The spatial variation of dust particulate matter concentrations during two Icelandic dust storms in 2015. Atmosphere 2016 7, 77.*
*Dupont, S., Klose, M., Irvine, M., González-Flórez, C., Alastuey, A. Bonnefond, J.-M., Dagsson-Waldhauserova, P., Gonzalez-Romero, A., Hussein, T., Lamaud, E., Meyer, H., Panta, A., Querol, X. Schepanski, S. Vergara Palacio, Wieser, A., Diez, J., Kandler, K., and Pérez García-Pando, C., 2024. Impact of dust source patchiness on the existence of a constant dust flux layer during aeolian erosion events. Journal of Geophysical Research: Atmospheres 129(12), e2023JD040657*

We thank the reviewer for pointing this out and have revised section 3.2: "Different $PM_1/PM_{10}$ ratios (~0.4–0.5) were reported for two dust storms affecting Reykjavík in 2015 (Dagsson-Waldhauserova et al., 2016), suggesting variability among these events and the need for further research.".

The Dupont et al. (2024) study is a very important contribution to our understanding of Icelandic dust storms; however, we believe a direct comparison between PM size fraction ratios is not appropriate because their measurements were taken at dust-source locations, whereas our data were collected hundreds of kilometers away in populated areas.

*L 452 – It is excellent that you compare your ratios to Icelandic dust storms, which are an extremely important source of PM in Iceland, but there are no permanent measurements around the local deserts of an area of 44000 km2. However, you state here you used the summer period due to eruption timing but then you chose two storms in November? These storms are not even visible on satellite images. Why not use clearly detectable dust storms instead? The ratios would likely differ.*

We thank the reviewer for this valuable suggestion. In the revised manuscript, we have included analysis of two dust storms that occurred during the summer of 2021 (24–29 May and 3–4 June), aligning with the eruption period. Data from these events are now incorporated into Figure 9 (which supplements Table 3 in the revised manuscript) and Table A1 (formerly Table 3).

We have also expanded the discussion in Section 3.2 to include these storms:

"Some of the highest $PM_{10}$ and $PM_{2.5}$ peaks in Reykjavík capital area (G3) during the eruption occurred on non-plume days (Fig. 4), notably in the periods 24–29 May and 3–4 June 2021. These two events accounted for most threshold exceedances— for example, five of seven for $PM_{10}$ and four of six for $PM_{2.5}$ at station G3-A—and were recorded across all G3 stations, suggesting a diffuse distal source. The dominant non-volcanic PM source in Iceland is natural dust from highland deserts, with dust storms occurring frequently throughout the year with significant regional and seasonal variability (Butwin et al., 2019; Dagsson-Waldhauserova et al., 2014; Nakashima and Dagsson-Waldhauserová, 2019). We used back-trajectory analysis (HYSPLIT) and crowd-sourced observations to confirm that the $PM_{10}$ and $PM_{2.5}$ peaks in Reykjavík on 24–29 May and 3–4 June were consistent with dust storms (Fig. A14)."

*L 485 – These 'sharp edges´ are also often identified during dust storm measurements in Iceland. Great that you are emphasizing it here.*

Very interesting observations; more research into the dispersion dynamics is clearly warranted.

*L 672 – Is this (PYRO-Box, Eco Counter, 2021) also a reference?*

We have improved the formatting to clarify that this is an instrument reference rather than a literature citation. The revised sentence reads: "These counters (PYRO-Box by Eco Counter) have a reported accuracy of 95% and a sensing range of 4 meters."

*L 724 - 'Iceland's exceptionally dense network'. I would argue here that the Icelandic AQ network is insufficient outside urban and industrialized areas. Thus, Iceland may be incorrectly described as one of the cleanest-air countries in the world. Please clarify by specifying: "…in the Capital Area and Reykjanes Peninsula," so it does not imply coverage of the entire country.*

We have changed the phrasing according to the reviewer's comment.

*Reference list missing references: Janssen et al., 2013; McDonnell et al., 2000; Tomášková et al., 2024*

We apologise for this error. The missing references have been included.